# On the Adversarial Robustness of Out-of-distribution Generalization Models

**Xin Zou**    **Weiwei Liu**[*]
School of Computer Science, Wuhan University
National Engineering Research Center for Multimedia Software, Wuhan University
Institute of Artificial Intelligence, Wuhan University
Hubei Key Laboratory of Multimedia and Network Communication Engineering, Wuhan University

## Abstract

Out-of-distribution (OOD) generalization has attracted increasing research attention in recent years, due to its promising experimental results in real-world applications. Interestingly, we find that existing OOD generalization methods are vulnerable to adversarial attacks. This motivates us to study OOD adversarial robustness. We first present theoretical analyses of OOD adversarial robustness in two different complementary settings. Motivated by the theoretical results, we design two algorithms to improve the OOD adversarial robustness. Finally, we conduct experiments to validate the effectiveness of our proposed algorithms. Our code is available at https://github.com/ZouXinn/OOD-Adv.

## 1  Introduction

Recent years have witnessed the remarkable success of modern machine learning techniques in many applications. A fundamental assumption of most machine learning algorithms is that the training and test data are drawn from the same underlying distribution. However, this assumption is always violated in many practical applications. The test environment is influenced by a range of factors, such as the distributional shifts across the photos caused by different cameras in image classification tasks, the voices of different persons in voice recognition tasks, and the variations between scenes in self-driving tasks [48]. Therefore, there is now a rapidly growing body of research with a focus on generalizing to unseen distributions, namely **out-of-distribution (OOD)** generalization [56].

Deep neural networks (DNNs) have achieved state-of-the-art performance in many fields. However, several prior works [59, 25] have demonstrated that DNNs may be vulnerable to imperceptibly changed adversarial examples, which has increased focus on the adversarial robustness of the models. **Adversarial robustness** refers to the invariance of a model to small perturbations of its input [54], while **adversarial accuracy** refers to a model's prediction performance on adversarial examples generated by an attacker. However, the adversarial robustness of OOD generalization models (OOD adversarial robustness) is less explored, despite its importance in many systems requiring high security such as self-driving cars. We evaluate the adversarial robustness of the models trained with the current OOD generalization algorithms (the detailed experimental settings can be found in Appendix C.3), and present the results in Table 1. Surprisingly, under the PGD-20 [44] attack, the algorithms achieve **nearly** $0\%$ adversarial accuracy on RotatedMNIST [21], VLCS [15], PACS [38], and OfficeHome [62], and **no more than** $10\%$ adversarial accuracy on ColoredMNIST [4]. These results show that even if the OOD generalization algorithms generalize well in different scenes, they remain highly vulnerable to adversarial attacks.

Motivated by these limitations of existing algorithms, we provide theoretical analyses for OOD adversarial robustness in two different but complementary OOD settings. We then design two

---

[*]Corresponding author: Weiwei Liu (liuweiwei863@gmail.com).

37th Conference on Neural Information Processing Systems (NeurIPS 2023).

Table 1: The results (%) for some of the current OOD generalization algorithms. We present the results in the form of **a/b**: here, **a** is the OOD adversarial accuracy under PGD-20 attack [44]; and **b** is the OOD adversarial accuracy under AutoAttack [12]. We conduct the $\ell_\infty$-norm attack. We use the perturbation radius $\epsilon = 0.1$ for RotatedMNIST and ColoredMNIST, and $\epsilon = \frac{4}{255}$ for VLCS. For the architecture, following [26], we use a small CNN-architecture for RotatedMNIST and ColoredMNIST, and ResNet-50 [27] for VLCS, PACS and OfficeHome. Since [64] do not realize MAT and LDAT for RotatedMNIST, we use X to denote the unrealized results. For more details about the algorithms, please refer to Appendix C.1. We use RM, CM, and OH as the abbreviation of RotatedMNIST, ColoredMNIST, and OfficeHome, respectively.

| Algorithm | RM | CM | VLCS | PACS | OH | Avg |
|---|---|---|---|---|---|---|
| ERM [61] | 0.6/0.0 | 5.8/X | 0.0/0.0 | 0.3/0.6 | 0.4/0.0 | 1.4/X |
| MLDG [39] | 0.2/0.0 | 4.8/X | 0.0/0.0 | 0.1/0.3 | 0.6/0.1 | 1.2/X |
| CDANN [42] | 0.9/0.0 | 8.2/X | 3.0/0.0 | 1.5/0.3 | 0.1/0.0 | 2.7/X |
| VREx [33] | 0.2/0.0 | 6.4/X | 0.0/0.0 | 0.0/0.3 | 0.4/0.1 | 1.4/X |
| RSC [28] | 1.0/0.0 | 3.9/X | 0.0/0.0 | 0.1/0.4 | 0.7/0.0 | 1.1/X |
| MAT [64] | X/X | 10.7/X | 0.0/0.0 | 0.7/1.4 | 0.8/0.1 | X/X |
| LDAT [64] | X/X | 7.9/X | 0.0/0.0 | 0.1/0.3 | 0.4/0.1 | X/X |

baseline algorithms to improve the OOD adversarial robustness, based on the implications of our theory, and validate the effectiveness of our proposed algorithms through experiments.

Our **contributions** can be summarized as follows:

1. We evaluate the adversarial robustness of current OOD generalization algorithms and experimentally verify that the current OOD generalization algorithms are vulnerable to adversarial attacks.

2. We present theoretical analyses for the adversarial OOD generalization error bounds in the average case and the limited training environments case. Specifically, our bounds in limited training environments involve a "distance" term between the training and the test environments. We further use a toy example to illustrate how the "distance" term affects the OOD adversarial robustness, which is verified by the experimental results in Section 5.

3. Inspired by our theory, we propose two algorithms to improve OOD adversarial robustness. Extensive experiments show that our algorithms are able to achieve **more than** 53% average adversarial accuracy over the datasets.

The remainder of this article is structured as follows: §2 introduces related works. §3 presents our main theory. §4 shows our two theory-driven algorithms. §5 provides our experimental results. Finally, the conclusions are presented in the last section.

## 2 Related Work

### 2.1 Adversarial Robustness

[59] show that DNNs are fragile to imperceptible distortions in the input space. One of the most popular methods used to improve adversarial robustness is **adversarial training (AT)**. The seminal AT work, the fast gradient sign method [25, FGSM], perturbs a sample towards its gradient direction to increase the loss, then uses the generated sample to train the model. Following this line of research, [44, 47, 34, 11] propose iterative variants of the gradient attack with improved AT frameworks. [70, 30, 41] investigates the adversarial robustness from the perspective of ordinary differential equations. Recently, [50, 66] utilize the data from generative models as data augmentation to improve adversarial robustness. Besides, [43] analyze the trade-off between robustness and fairness, [37] study the worst-class adversarial robustness in adversarial training.

For the theoretical perspective, [46] study the PAC learnability of adversarial robust learning, [69] extend the work of [46] to multiclass case, [72, 31, 5, 18, 67] give theoretical analyses to adversarial training by standard uniform convergence argumentation and giving a bound of the Rademacher complexity, and [65, 78] study adversarial robustness under self-supervised learning.

## 2.2 Out-of-distribution generalization

OOD generalization aims to train a model with data from the training environments so that it is capable of generalizing to an unseen environment. A large number of algorithms have been developed that aim to improve OOD generalization. One series of works focuses on minimizing the discrepancies between the training environments [40, 17, 42, 58, 1]. The most related work among them is [1], which measures the discrepancy between the domains by $d_{\mathcal{H}}(S, T)$, while we focus on adversarial robustness and use $d_{\ell(\mathcal{H})}^{\mathcal{B}}(S, T)$. Meta-learning domain generalization [39, MLDG] leverages the meta-learning approach and simulates train/test distributional shift during training by synthesizing virtual testing domains within each mini-batch. [53, GroupDRO] studies applying distributionally robust optimization (DRO) [19, 36, 77, 13] to learn models that instead minimize the worst-case training loss over a set of pre-defined groups. Another line of works [68, 64] conducts adversarial training to improve the OOD generalization performance. In this work, we focus on improving the OOD adversarial robustness.

From the theoretical perspective, [10] introduce a formal framework and argue that OOD generalization can be viewed as a kind of supervised learning problem by augmenting the original feature space with the marginal distribution of feature vectors. In [51], OOD generalization is cast into an online game where a player (model) minimizes the risk for a "new" distribution presented by an adversary at each time-step. [14] propose a probabilistic framework for domain generalization called Probable Domain Generalization, wherein the key idea is that distribution shifts seen during training should inform us of probable shifts at test time. Notably, all these works focus on OOD generalization performance, while we present theoretical results for OOD adversarial robustness.

## 2.3 Works relating domain shifts and adversrial robustness

[8] focuses on improving the adversarial robustness of the models by regarding the adversarial distribution as the target domain and then applying the domain adaptation methods, while we focus on improving the model's adversarial robustness on the OOD distribution. [71] studies the relationship between adversarial robustness and OOD generalization, it shows that good adversarial robustness implies good OOD performance when the target domain lies in a Wasserstein ball. While we study the OOD adversarial robustness and propose algorithms to improve OOD adversarial robustness. [2] empirically analyzes the transferability of models' adversarial/certified robustness under distributional shifts. It shows that adversarially trained models do not generalize better without fine-tuning and that the accuracy-robustness trade-off generalizes to the unseen domain. Its results for adversarial robustness can also be found in our experimental results. [29] investigates how to improve the adversarial robustness of a model against ensemble attacks or unseen attacks. [29] regards the adversarial examples for each type of attack (such as FGSM, PGD, CW, and so on) as a domain, and utilizes the OOD generalization methods to improve the models generalization performance under different (maybe unseen) attacks.

[45, 22] study the relationship between the dependence on spurious correlations and the adversarial robustness of the models and show that adversarial training increases the model's reliance on spurious features. [60] studies the relationship between fairness and adversarial robustness and shows that models that are fairer will be less adversarially robust. However, they do not consider the adversarial robustness of the model on the unseen target domain, which is the topic of this paper.

## 3 Theoretical Analysis for OOD Adversarial Robustness

In this section, we present two theorems in two different settings, each of which inspires an algorithm designed to improve the OOD adversarial robustness. The proofs of all results in this section can be found in Appendix A. We first introduce some notations and basic setups.

**Notations.** We define $[n] := \{1, 2, \cdots, n\}$. We denote scalars and vectors with lowercase letters and lowercase bold letters respectively. We use uppercase letters to denote matrices or random variables, and uppercase bold letters to denote random vectors or random matrices. For a vector $\boldsymbol{x} \in \mathbb{R}^n$, we define the $\ell_p$-norm of $\boldsymbol{x}$ as $\|\boldsymbol{x}\|_p := \left(\sum_{i=1}^n |x_i|^p\right)^{1/p}$ for $p \in [1, \infty)$, where $x_i$ is the $i$-th element of $\boldsymbol{x}$; for $p = \infty$, we define $\|\boldsymbol{x}\|_\infty := \max_{1 \le i \le n} |x_i|$. For a matrix $A \in \mathbb{R}^{m \times n}$, the Frobenius norm of $A$

is defined as $\|A\|_F := \left( \sum_{i=1}^{m} \sum_{j=1}^{n} A_{ij}^2 \right)^{\frac{1}{2}}$, where $A_{ij}$ is the entry of $A$ at the $i$-th row and $j$-th column. We define the determinant of $A$ as $\det(A)$. $\mathcal{N}(\boldsymbol{\mu}, \Sigma)$ represents the multivariable Gaussian distribution with mean vector $\boldsymbol{\mu}$ and covariance matrix $\Sigma$. Given $f, g : \mathbb{R} \to \mathbb{R}_+$, we write $f = \mathcal{O}(g)$ if there exist $x_0, \alpha \in \mathbb{R}_+$ such that for all $x > x_0$, we have $f(x) \leq \alpha g(x)$. We use $sign(\cdot)$ to denote the sign function [57].

**Setups.** Let $\mathcal{X} \in \mathbb{R}^m$ be the input space and $\mathcal{Y}$ be the label space. We set $\mathcal{Y} = \{\pm 1\}$, $\mathcal{Y} = \{1, 2, \cdots, K\}$ (where $K$ is the number of classes), and $\mathcal{Y} = \mathbb{R}$ for the binary classification problem, the multi-class classification problem, and the regression problem, respectively. We use $\ell : \mathcal{Y} \times \mathcal{Y} \to \mathbb{R}_+$ as the loss function. We consider learning with the hypothesis class $\mathcal{H} \subseteq \{h : \mathcal{X} \to \mathcal{Y}\}$. Given a distribution $\mathcal{D}$ on $\mathcal{X} \times \mathcal{Y}$, the error of $h \in \mathcal{H}$ with respect to the loss function $\ell$ under the distribution $\mathcal{D}$ is $\mathcal{R}_{\mathcal{D}}(\ell, h) = \mathbb{E}_{(\boldsymbol{x}, y) \sim \mathcal{D}}[\ell(h(\boldsymbol{x}), y)]$, where $\boldsymbol{x} \in \mathcal{X}$ and $y \in \mathcal{Y}$. We further define $\mathcal{B}(\cdot) : \mathcal{X} \to 2^{\mathcal{X}}$ as a perturbation function that maps an input $\boldsymbol{x}$ to a subset $\mathcal{B}(\boldsymbol{x}) \subseteq \mathcal{X}$, where $2^{\mathcal{X}}$ is the power set of $\mathcal{X}$. The adversarial error of the predictor $h$ under the perturbation function $\mathcal{B}(\cdot)$ is defined as

$$\mathcal{R}_{\mathcal{D}}^{\mathcal{B}}(\ell, h) = \mathbb{E}_{(\boldsymbol{x}, y) \sim \mathcal{D}} \left[ \sup_{\boldsymbol{x}' \in \mathcal{B}(\boldsymbol{x})} \ell(h(\boldsymbol{x}'), y) \right].$$

## 3.1 The Average Case

In this section, following [20, 14, 52], we consider the average case, i.e., the case in which the target environment follows a distribution. In this case, we aim to minimize the average target adversarial error of the hypothesis $h \in \mathcal{H}$, where the average is taken over the distribution of the target environment.

Suppose that $\mathcal{P}(\mathcal{X} \times \mathcal{Y})$ is the set of all possible probability measures (environments) on $\mathcal{X} \times \mathcal{Y}$ for the task of interest. Assume there is a prior $p$ on $\mathcal{P}(\mathcal{X} \times \mathcal{Y})$, which is the distribution of the environments. Moreover, suppose the process of sampling training and test data is as follows:

(1). We generate the training data according to the following two steps: (i) we sample $t$ training environments from $p$, i.e., $\mathcal{D}_1, \cdots, \mathcal{D}_t \sim p$; (ii) the examples $\widehat{\mathcal{D}}_i = \{(\boldsymbol{x}_{i1}, y_{i1}), \cdots, (\boldsymbol{x}_{in_i}, y_{in_i})\} \sim \mathcal{D}_i^{n_i}$ are drawn independent and identically distributed (i.i.d.) from $\mathcal{D}_i$, where $n_i$ is the size of $\widehat{\mathcal{D}}_i$ and $i \in [t]$. To simplify the notations, we also use $\widehat{\mathcal{D}}_i$ to denote the empirical distribution of the dataset $\{(\boldsymbol{x}_{i1}, y_{i1}), \cdots, (\boldsymbol{x}_{in_i}, y_{in_i})\}$. Let $\widehat{\mathcal{D}} = \frac{1}{t} \sum_{i=1}^{t} \widehat{\mathcal{D}}_i$ and $\hat{p} = \frac{1}{t} \sum_{i=1}^{t} \mathcal{D}_i$. To simplify the problem, we assume $n_1 = n_2 = \cdots = n_t = n$, but note that with a more careful analysis, our result in the average case can be extended to the case in which $n_1, \cdots, n_t$ are not necessarily the same.

(2). For each test example, we first sample an environment from $p$, i.e., $\mathcal{D} \sim p$, then sample an example $(\boldsymbol{x}, y) \sim \mathcal{D}$.

Let $\mathcal{L}_p(\ell, h) = \mathbb{E}_{\mathcal{D} \sim p}[\mathcal{R}_{\mathcal{D}}(\ell, h)]$ be the average risk of $h$ on prior $p$. For the perturbation function $\mathcal{B}(\cdot)$, we define $\mathcal{L}_p^{\mathcal{B}}(\ell, h) = \mathbb{E}_{\mathcal{D} \sim p}[\mathcal{R}_{\mathcal{D}}^{\mathcal{B}}(\ell, h)]$ as the average adversarial risk of $h$. The empirical Rademacher complexity [55, Chapter 26] of the hypothesis class $\mathcal{H}$ is defined as follows:

$$\mathfrak{R}_n(\mathcal{H}) = \mathbb{E}_{\boldsymbol{\sigma}} \left[ \sup_{h \in \mathcal{H}} \frac{1}{n} \sum_{i=1}^{n} \sigma_i h(\boldsymbol{x}_i) \right],$$

where $\boldsymbol{\sigma}$ is a Rademacher random vector with i.i.d. entries, and $\{\boldsymbol{x}_1, \cdots, \boldsymbol{x}_n\}$ is a set of data points. The following theorem presents an upper bound for the average adversarial risk of $h$.

**Theorem 3.1.** *Suppose the loss function $\ell$ is bounded, i.e., $\ell \in [0, U]$. Then with probability of at least $1 - \delta$ over the sampling of $\widehat{\mathcal{D}}_1, \cdots, \widehat{\mathcal{D}}_t$, the following bound holds for all $h \in \mathcal{H}$:*

$$\mathcal{L}_p^{\mathcal{B}}(\ell, h) \leq \mathcal{L}_{\widehat{\mathcal{D}}}^{\mathcal{B}}(\ell, h) + 2\mathfrak{R}_t(\widetilde{\mathcal{G}}) + 2\mathfrak{R}_{tn}(\widetilde{\mathcal{G}}) + 3U\sqrt{\frac{ln4/\delta}{2t}} + 3U\sqrt{\frac{ln4/\delta}{2tn}},$$

*where*

$$\widetilde{\mathcal{G}} = \left\{ g_h : \mathcal{X} \times \mathcal{Y} \to \mathbb{R}_+ \Big| g_h(\boldsymbol{x}, y) = \sup_{\boldsymbol{x}' \in \mathcal{B}(\boldsymbol{x})} \ell(h(\boldsymbol{x}', y)), h \in \mathcal{H} \right\}.$$

Theorem 3.1 presents a bound for all hypotheses $h \in \mathcal{H}$. We now consider the convergence property of the adversarial empirical risk minimization (**AERM**) algorithm in the average case. We define the output of the AERM algorithm as $\hat{h} \in \arg\inf_{h \in \mathcal{H}} \mathcal{L}_{\widehat{\mathcal{D}}}^{\mathcal{B}}(\ell, h)$. We then define the hypothesis with the best adversarial generalization performance as $h^{\star} \in \arg\inf_{h \in \mathcal{H}} \mathcal{L}_p^{\mathcal{B}}(\ell, h)$.

The next corollary shows an upper bound for the excess risk [63, Chapter 4] of AERM:

**Corollary 3.2.** *Suppose the loss function $\ell$ is bounded, i.e., $\ell \in [0, U]$. Then with probability of at least $1 - \delta$ over the sampling of $\widehat{\mathcal{D}}_1, \cdots, \widehat{\mathcal{D}}_t$, the following bound holds:*

$$\mathcal{L}_p^{\mathcal{B}}(\ell, \hat{h}) \leq \mathcal{L}_p^{\mathcal{B}}(\ell, h^{\star}) + 4\mathfrak{R}_t(\widetilde{\mathcal{G}}) + 4\mathfrak{R}_{tn}(\widetilde{\mathcal{G}}) + 3U\sqrt{\frac{ln8/\delta}{2t}} + 3U\sqrt{\frac{ln8/\delta}{2tn}}.$$

**Remark 1.** *The convergence rate for both Theorem 3.1 and Corollary 3.2 is $\mathcal{O}\left(t^{-\frac{1}{2}}\right)$, which prompts us to ask: can we derive a tighter bound that has a faster convergence rate than $\mathcal{O}\left(t^{-\frac{1}{2}}\right)$? The next section provides an affirmative answer to this question.*

## 3.2  Theory with Limited Environments

In this section, we provide the theoretical analysis for the limited training environment case, i.e., in the case in which $t = \mathcal{O}(1)$. Suppose that we have $t$ training environments $\mathcal{D}_1, \cdots, \mathcal{D}_t$ and one unknown target environment $\mathcal{T}$. Our goal is to find a predictor that obtains good adversarial robustness on the target environment $\mathcal{T}$. Assume we obtain $n$ examples from each training distribution $\mathcal{D}_i$.

First, we introduce a discrepancy between the distributions. Based on the hypothesis class $\mathcal{H}$, the seminal work for unsupervised domain adaptation (UDA), [9], defines the discrepancy between two distributions as follows:

$$d_{\mathcal{H}}(\mathcal{D}, \mathcal{D}') = 2 \sup_{h \in \mathcal{H}} \left| \mathbb{P}_{\boldsymbol{x} \sim \mathcal{D}}[h(\boldsymbol{x}) = 1] - \mathbb{P}_{\boldsymbol{x} \sim \mathcal{D}'}[h(\boldsymbol{x}) = 1] \right|.$$

[9] analyze the generalization error bound of the target domain for UDA by $d_{\mathcal{H}\Delta\mathcal{H}}(\cdot, \cdot)$, where $g \in \mathcal{H}\Delta\mathcal{H} \iff g = h \oplus h'$ for some $h, h' \in \mathcal{H}$; here, $\oplus$ is the XOR operator. However, the above definition is limited to the binary classification. This paper extends $d_{\mathcal{H}}$ to the multi-class adversarial case. Given the loss function $\ell$, distributions $P, Q$ on $\mathcal{X} \times \mathcal{Y}$, and hypothesis class $\mathcal{H}$, we define the adversarial discrepancy as follows:

$$d_{\ell(\mathcal{H})}^{\mathcal{B}}(P, Q) = \sup_{h \in \mathcal{H}} \left| \mathbb{E}_{(\boldsymbol{x}, y) \sim P}\left[ \sup_{\boldsymbol{x}' \in \mathcal{B}(\boldsymbol{x})} \ell(h(\boldsymbol{x}'), y) \right] - \mathbb{E}_{(\boldsymbol{x}, y) \sim Q}\left[ \sup_{\boldsymbol{x}' \in \mathcal{B}(\boldsymbol{x})} \ell(h(\boldsymbol{x}'), y) \right] \right|.$$

When $\mathcal{B}(\boldsymbol{x}) = \{\boldsymbol{x}\}$, $d_{\ell(\mathcal{H})}^{\mathcal{B}}(P, Q)$ becomes the standard case. It can be easily verified that $d_{\ell(\mathcal{H})}^{\mathcal{B}}(\cdot, \cdot)$ is symmetric and satisfies the triangle inequality (the proof can be found in Appendix A.3); thus, $d_{\ell(\mathcal{H})}^{\mathcal{B}}(\cdot, \cdot)$ is a pseudometric.

**Comparison of $d_{\mathcal{H}}(\cdot, \cdot)$ and $d_{\ell(\mathcal{H})}^{\mathcal{B}}(\cdot, \cdot)$.** The theory outlined in [9, Theorem 2] presents an upper bound with a $d_{\mathcal{H}\Delta\mathcal{H}}(\cdot, \cdot)$ term. To align the feature distributions of the source and target domain, we need to calculate $d_{\mathcal{H}\Delta\mathcal{H}}(P, Q)$, which takes the supremum over two hypotheses $h, h' \in \mathcal{H}$:

$$d_{\mathcal{H}\Delta\mathcal{H}}(P, Q) = 2 \sup_{h, h' \in \mathcal{H}} \left| \mathbb{P}_{\boldsymbol{x} \sim P}[h(\boldsymbol{x}) \neq h'(\boldsymbol{x})] - \mathbb{P}_{\boldsymbol{x} \sim Q}[h(\boldsymbol{x}) \neq h'(\boldsymbol{x})] \right|.$$

However, the definition of $d_{\ell(\mathcal{H})}^{\mathcal{B}}(\cdot, \cdot)$ shows that: $d_{\ell(\mathcal{H})}^{\mathcal{B}}(P, Q)$ takes the supremum over one hypothesis $h \in \mathcal{H}$, which is easier to optimize [76] and can thus significantly ease the minimax optimization in Section 4.2.

**Theorem 3.3.** *For a given but unknown target distribution $\mathcal{T}$, let $\Delta^{t-1} := \{(\lambda_1, \cdots, \lambda_t) | \lambda_i \geq 0, \sum_{i=1}^{t} \lambda_i = 1\}$ be the $(t-1)$-dimensional simplex, and $Conv(\mathfrak{D}) := \left\{ \sum_{i=1}^{t} \lambda_i \mathcal{D}_i | \boldsymbol{\lambda} \in \Delta^{t-1} \right\}$ be the convex hull of $\mathfrak{D} = \{\mathcal{D}_1, \ldots, \mathcal{D}_t\}$. Let $\mathcal{T}_P \in \arg\inf_{\mathcal{D} \in Conv(\mathfrak{D})} d_{\ell(\mathcal{H})}^{\mathcal{B}}(\mathcal{D}, \mathcal{T})$ be the "projection" of $\mathcal{T}$*

onto $Conv(\mathfrak{D})$, and $\boldsymbol{\lambda}^\star$ be the weight vector where $\mathcal{T}_P = \sum_{i=1}^t \lambda_i^\star \mathcal{D}_i$. Assume $\ell \in [0, U]$. Then: with probability of at least $1 - \delta$, for all $h \in \mathcal{H}$,

$$\mathcal{R}_{\mathcal{T}}^{\mathcal{B}}(\ell, h) \leq \frac{1}{t} \sum_{i=1}^t \mathcal{R}_{\widehat{\mathcal{D}}_i}^{\mathcal{B}}(\ell, h) + \frac{1}{t} \sum_i \sum_j \lambda_j^\star d_{\ell(\mathcal{H})}^{\mathcal{B}}(\widehat{\mathcal{D}}_i, \widehat{\mathcal{D}}_j) + d_{\ell(\mathcal{H})}^{\mathcal{B}}(\mathcal{T}, \mathcal{T}_P)$$

$$+ 4\mathfrak{R}_{tn}(\widetilde{\mathcal{G}}) + 2\mathfrak{R}_n(\widetilde{\mathcal{G}}) + 6U\sqrt{\frac{ln8/\delta}{2tn}} + 3U\sqrt{\frac{ln(16t/\delta)}{2n}}.$$

**Remark 2.** *The first term of the bound is the average empirical adversarial robust error of the model on training environments, and the second term is the weighted average discrepancy on the empirical training distributions $\widehat{\mathfrak{D}} = \{\widehat{\mathcal{D}}_1, \cdots, \widehat{\mathcal{D}}_t\}$. The third term can be viewed as the **distance** between $\mathcal{T}$ and the convex hull $Conv(\mathfrak{D})$, which is fixed once the task, $\ell, \widehat{\mathfrak{D}}$ and $\mathcal{T}$ are given. Thus, minimizing the first two terms of the bound is able to improve the OOD adversarial robustness.*

*Moreover, $\mathfrak{R}_{tn}(\widetilde{\mathcal{G}})$ and $\mathfrak{R}_n(\widetilde{\mathcal{G}})$ measure the capacity of the model, and can be regarded as an implicit regularizer on the model. There are many existing works that present the upper bounds of the empirical Rademacher complexity for neural networks [49, 6, 7, 23] and the adversarial Rademacher complexity [31, 73, 18, 5, 67]. Their results can be summarized as follows: for bounded $\mathcal{X}$, with proper weak assumptions on the loss function $\ell$, $\mathfrak{R}_n(\widetilde{\mathcal{G}})$ can be upper-bounded by $\mathcal{O}(\frac{C}{\sqrt{n}})$, where $C$ is a constant that is related to some norm of the parameters of $h$ and increases with the norm. Thus, we consider constraints on the norm of the parameters of $h$ in the algorithm designing part of Section 4.*

*Last but not least, different from the results in Theorem 3.1, the convergence rate is $\mathcal{O}\left(\sqrt{\frac{ln1/\delta}{tn}} + \sqrt{\frac{lnt/\delta}{n}}\right)$. When $t = \mathcal{O}(1)$, $\mathcal{O}\left(\sqrt{\frac{lnt/\delta}{n}}\right)$ in Theorem 3.3 converges much faster than $\mathcal{O}\left(\sqrt{\frac{ln1/\delta}{t}}\right)$ in Theorem 3.1, since $n \gg t$.*

The $d_{\ell(\mathcal{H})}^{\mathcal{B}}(\mathcal{T}, \mathcal{T}_P)$ term in Theorem 3.3 is determined by the distance between the source and target environments. A larger $d_{\ell(\mathcal{H})}^{\mathcal{B}}(\mathcal{T}, \mathcal{T}_P)$ may lead to worse adversarial robustness of the model on the target domain. Next, we present a toy example to illustrate this case.

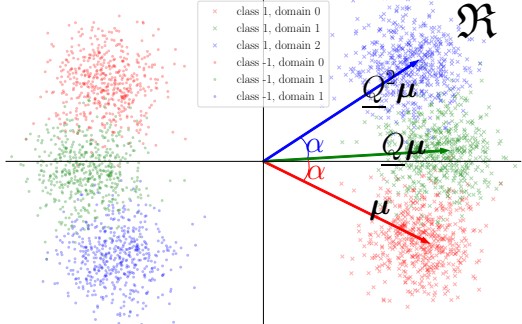

Figure 1: An intuitive visualization of the distributions $\mathcal{D}^{(0)}, \mathcal{D}^{(1)}, \mathcal{D}^{(2)}$ (**red, green, blue** respectively). We also represent the three mean vectors $\boldsymbol{\mu}, Q\boldsymbol{\mu}, Q^2\boldsymbol{\mu}$ using red, green, and blue arrows. The angle between $\boldsymbol{\mu}, Q\boldsymbol{\mu}$ is $\alpha$, which is the same as the angle between $Q\boldsymbol{\mu}, Q^2\boldsymbol{\mu}$. More details about the relationship between $Q$ and $\alpha$ can be found in the proof of Theorem A.4 in Appendix A.1.

**Example 1.** We consider the input space $\mathcal{X} = \mathbb{R}^d \subseteq \mathbb{R}^m$ and label space $\mathcal{Y} = \{+1, -1\}$. For simplicity, we consider only three distributions on $\mathcal{X} \times \mathcal{Y}$, i.e., $\mathcal{D}^{(0)}, \mathcal{D}^{(1)}$ and $\mathcal{D}^{(2)}$. The marginal distributions for $Y$ satisfy $\mathcal{D}_Y^{(0)}(\{-1\}) = \mathcal{D}_Y^{(0)}(\{+1\}) = \mathcal{D}_Y^{(1)}(\{-1\}) = \mathcal{D}_Y^{(1)}(\{+1\}) = \mathcal{D}_Y^{(2)}(\{-1\}) = \mathcal{D}_Y^{(2)}(\{+1\}) = \frac{1}{2}$. For $\mathcal{D}^{(i)}$, let the conditional distribution of $\boldsymbol{X}^{(i)}$ given $Y$ be $\boldsymbol{X}^{(i)}|Y = y \sim \mathcal{N}(y\boldsymbol{\mu}^{(i)}, \sigma^2 I)$, where $\boldsymbol{\mu}^{(i)} \in \mathbb{R}^d$ is a non-zero vector and $\sigma^2 I \in \mathbb{R}^{d \times d}$ is the covariance matrix, i.e., the elements of $\boldsymbol{X}^{(i)}$ are independent. Let $\underline{Q}$ be a rotation transformation on $\mathbb{R}^d$, which is a special orthogonal transformation. Suppose that $\underline{Q} \neq \underline{I}$, where $\underline{I}$ is the identity transformation. Let $Q \in \mathbb{R}^{d \times d}$ be the matrix of $\underline{Q}$ under some orthonormal basis. We then know that $Q$ is a rotation matrix and $Q^T Q = QQ^T = \overline{I}, \det(Q) = 1, Q \neq I$. To model the distributional shift among the environments, we apply the transformation $\underline{Q}$ to $\boldsymbol{\mu}^{(0)}$ and use $\boldsymbol{\mu}^{(1)} = \underline{Q}\boldsymbol{\mu}^{(0)}, \boldsymbol{\mu}^{(2)} = \underline{Q}^2\boldsymbol{\mu}^{(0)}$ as the mean vectors of $\mathcal{D}^{(1)}, \mathcal{D}^{(2)}$ respectively. Here, we consider the $\ell_2$-norm adversarial attack with radius $\epsilon$, i.e., $\mathcal{B}(\boldsymbol{x}) = \{\boldsymbol{x}' : \|\boldsymbol{x} - \boldsymbol{x}'\|_2 \leq \epsilon\}$. We use two environments as the training environments and the remainder as the test environment. Theorem 3.4 shows that a larger distance between the training and test environments leads to worse OOD adversarial robustness.

**Theorem 3.4.** *Consider the setting in Example 1, and suppose that $\boldsymbol{\mu}$ lies in the 2-dimensional subspace $\mathfrak{R}$ in Theorem A.4 (see Appendix A.1 for details about $\mathfrak{R}$, and see Figure 1 for an intuitive visualization). Let $\ell_{01}(x,y) = \mathbb{1}[x \neq y]$, where $\mathbb{1}[\cdot]$ is the indicator function and $B_p^r(\boldsymbol{x}) = \{\boldsymbol{x}' : \|\boldsymbol{x} - \boldsymbol{x}'\|_p \leq r\}$. Consider training with hypothesis class $\mathcal{H} = \{h_{\boldsymbol{w}} : h_{\boldsymbol{w}}(x) = sign(\boldsymbol{w}^T x), \boldsymbol{w} \in \mathbb{R}^d\}$, and denote $\mathcal{D}^{(ij)} = \frac{1}{2}\mathcal{D}^{(i)} + \frac{1}{2}\mathcal{D}^{(j)}, i, j \in \{0, 1, 2\}, i \neq j$. Consider the $\ell_2$-norm adversarial attack with radius $\epsilon$; for notation convenience, let $\widetilde{\mathcal{R}}_{ij}(\boldsymbol{w}) = \mathcal{R}_{\mathcal{D}^{(ij)}}^{B_2^\epsilon}(\ell_{01}, h_{\boldsymbol{w}})$ and $\widetilde{\mathcal{R}}_i(\boldsymbol{w}) = \mathcal{R}_{\mathcal{D}^{(i)}}^{B_2^\epsilon}(\ell_{01}, h_{\boldsymbol{w}})$. Let $\Phi(\cdot)$ be the distribution function of the standard normal distribution. Let $\alpha$ denote the angle between $\boldsymbol{\mu}$ and $\underline{Q}\boldsymbol{\mu}$, which is the rotation angle of $\underline{Q}$ in the subspace $\mathfrak{R}$. Furthermore, suppose that $0 < \alpha \leq arccos\frac{\epsilon}{\|\boldsymbol{\mu}_1\|_2}$. We then have:*

*(1) If we train with $\mathcal{D}^{(0)}$ and $\mathcal{D}^{(1)}$, let $\boldsymbol{w}_{(01)} = \boldsymbol{\mu} + \underline{Q}\boldsymbol{\mu}$, which achieves the minimum of $\widetilde{\mathcal{R}}_{01}(\boldsymbol{w})$. Then the adversarial accuracy of $\boldsymbol{w}_{(01)}$ on the test distribution $\mathcal{D}^{(2)}$ is $\widetilde{\mathcal{R}}_2(\boldsymbol{w}_{(01)}) = \Phi\left(\frac{\epsilon}{\sigma} - \frac{(\boldsymbol{\mu}+\underline{Q}\boldsymbol{\mu})^T \underline{Q}^2\boldsymbol{\mu}}{\sigma\|\boldsymbol{\mu}+\underline{Q}\boldsymbol{\mu}\|_2}\right)$.*

*(2) If we train with $\mathcal{D}^{(0)}$ and $\mathcal{D}^{(2)}$, let $\boldsymbol{w}_{(02)} = \boldsymbol{\mu} + \underline{Q}^2\boldsymbol{\mu}$, which achieves the minimum of $\widetilde{\mathcal{R}}_{02}(\boldsymbol{w})$. Then the adversarial accuracy of $\boldsymbol{w}_{(02)}$ on the test distribution $\mathcal{D}^{(1)}$ is $\widetilde{\mathcal{R}}_1(\boldsymbol{w}_{(02)}) = \Phi\left(\frac{\epsilon}{\sigma} - \frac{(\boldsymbol{\mu}+\underline{Q}^2\boldsymbol{\mu})^T \underline{Q}\boldsymbol{\mu}}{\sigma\|\boldsymbol{\mu}+\underline{Q}^2\boldsymbol{\mu}\|_2}\right)$.*

*(3) If we train with $\mathcal{D}^{(1)}$ and $\mathcal{D}^{(2)}$, let $\boldsymbol{w}_{(12)} = \underline{Q}\boldsymbol{\mu} + \underline{Q}^2\boldsymbol{\mu}$, which achieves the minimum of $\widetilde{\mathcal{R}}_{12}(\boldsymbol{w})$. Then the adversarial accuracy of $\boldsymbol{w}_{(12)}$ on the test distribution $\mathcal{D}^{(0)}$ is $\widetilde{\mathcal{R}}_0(\boldsymbol{w}_{(12)}) = \Phi\left(\frac{\epsilon}{\sigma} - \frac{(\underline{Q}\boldsymbol{\mu}+\underline{Q}^2\boldsymbol{\mu})^T \boldsymbol{\mu}}{\sigma\|\underline{Q}\boldsymbol{\mu}+\underline{Q}^2\boldsymbol{\mu}\|_2}\right)$.*

*(4) $\widetilde{\mathcal{R}}_1(\boldsymbol{w}_{(02)}) < \widetilde{\mathcal{R}}_2(\boldsymbol{w}_{(01)}) = \widetilde{\mathcal{R}}_0(\boldsymbol{w}_{(12)})$.*

**Remark 3.** *Let $\{i, j, k\} = \{0, 1, 2\}$. For task $i$, we use $\mathcal{D}^{(j)}, \mathcal{D}^{(k)}$ as the training environments and $\mathcal{D}^{(i)}$ as the test environment. $\widetilde{\mathcal{R}}_i(\boldsymbol{w}_{(jk)})$ is the target adversarial error of the learned classifier $\boldsymbol{w}_{(jk)}$ in task $i$.*

*We now consider the distance between the training and test environments for each task. Intuitively, we regard the angle as the "distance" between two distributions. We define $angle(i, j)$ as the angle between the mean vector of $\mathcal{D}^{(i)}$ and $\mathcal{D}^{(j)}$, $i \neq j$. For task $i$, we define the average "distance" between the test environment and each training environment as $d_{avg}(i) := \frac{angle(i,j)+angle(i,k)}{2}$. We use $d_{avg}(i)$ as a measure of the distance between the training environments and the test environment for task $i$.*

*From the settings in Example 1, it can be clearly seen that $d_{avg}(0) = \frac{angle(0,1)+angle(0,2)}{2} = \frac{\alpha+2\alpha}{2} = \frac{3\alpha}{2} = d_{avg}(2)$ and $d_{avg}(1) = \frac{angle(1,0)+angle(1,2)}{2} = \frac{\alpha+\alpha}{2} = \alpha$, which implies that $d_{avg}(1) < d_{avg}(0) = d_{avg}(2)$. Theorem 3.4 tells us that $\widetilde{\mathcal{R}}_1(\boldsymbol{w}_{(02)}) < \widetilde{\mathcal{R}}_0(\boldsymbol{w}_{(12)}) = \widetilde{\mathcal{R}}_2(\boldsymbol{w}_{(01)})$, and thus implies that **a smaller distance between the training and test environments leads to better OOD adversarial robustness**.*

*Moreover, the assumption of the angle between $\boldsymbol{\mu}$ and $\underline{Q}\boldsymbol{\mu}$ in Theorem 3.4 is reasonable. Consider $\epsilon = \frac{\|\boldsymbol{\mu}\|_2}{2}$; in this case, the attack is strong and perceptible to human eyes. Then, $\alpha \leq arccos\frac{\epsilon}{\|\boldsymbol{\mu}_1\|_2} = arccos\frac{1}{2} = \frac{\pi}{3}$. In this case, the maximal angle between the environments is $\frac{2\pi}{3}$, which leads to a strong distribution shift. When $\epsilon < \frac{\|\boldsymbol{\mu}\|_2}{2}$, the allowed rotation angle can be further enlarged, and when $\epsilon = 0$, it becomes the standard case.*

*Furthermore, the data distribution here can be regarded as a simplified data model for RotatedMNIST. Moreover, our experimental results in Section 5 are consistent with our analysis here in both the standard and adversarial cases. Please refer to the observation part of Section 5.2 and the table in Appendix D.1 for further details.*

# 4   Algorithms

In this section, based on our theory, we present two algorithms that can improve the adversarial robustness of the model on the target environment.

## 4.1 Adversarial Empirical Risk Minimization (AERM or AT)

Based on Theorem 3.1, which shows an upper bound of the average adversarial risk over the environments, we propose our first algorithm: adversarial empirical risk minimization (AERM, which corresponds to applying AT to multiple source domains). The bound in Theorem 3.1 consists of the average adversarial empirical risk $\mathcal{L}_{\widehat{\mathcal{D}}}^{\mathcal{B}}(\ell, h) = \frac{1}{t}\sum_{i=1}^{t} \mathcal{R}_{\widehat{\mathcal{D}}_i}^{\mathcal{B}}(\ell, h)$ and two empirical Rademacher complexity terms $\mathfrak{R}_t(\widetilde{\mathcal{G}}) + \mathfrak{R}_{tn}(\widetilde{\mathcal{G}})$. As outlined in Remark 2, the empirical Rademacher complexity implies an implicit regularizer on the model capacity. We choose $\|\cdot\|_F$ as a regularizer on the model parameters. The optimization objective of AT is as follows:

$$L_{\text{AT}}(h) = \frac{1}{t}\sum_{i=1}^{t} \mathcal{R}_{\widehat{\mathcal{D}}_i}^{\mathcal{B}}(\ell, h) + \lambda\|W\|_F,$$

where $\lambda$ is a trade-off hyper-parameter, and $W$ is the parameter matrix of the model. From Remark 1, we know that AT may not generalize well in the case where the training environments are limited. Next, we propose another algorithm for the limited environment case.

## 4.2 Robust DANN (RDANN)

Theorem 3.3 shows that $\mathfrak{R}_n(\widetilde{\mathcal{G}})$, $\frac{1}{t}\sum_{i=1}^{t} \mathcal{R}_{\widehat{\mathcal{D}}_i}^{\mathcal{B}}(\ell, h)$, and $\frac{1}{t}\sum_i\sum_j \lambda_j^\star d_{\ell(\mathcal{H})}^{\mathcal{B}}(\widehat{\mathcal{D}}_i, \widehat{\mathcal{D}}_j)$ play the key roles in designing the OOD adversarial training methods. However, the weights $\lambda_1^\star, \cdots, \lambda_t^\star$ are unknown, since we have no information about the target distribution $\mathcal{T}$. Since $\lambda_j^\star \in [0, 1], \forall j$, it is evident that $\frac{1}{t}\sum_i\sum_j \lambda_j^\star d_{\ell(\mathcal{H})}^{\mathcal{B}}(\widehat{\mathcal{D}}_i, \widehat{\mathcal{D}}_j) \leq \frac{1}{t}\sum_i\sum_j d_{\ell(\mathcal{H})}^{\mathcal{B}}(\widehat{\mathcal{D}}_i, \widehat{\mathcal{D}}_j)$. We therefore turn to optimize the average discrepancy $\frac{1}{t^2}\sum_i\sum_j d_{\ell(\mathcal{H})}^{\mathcal{B}}(\widehat{\mathcal{D}}_i, \widehat{\mathcal{D}}_j)$. To improve the OOD adversarial robustness, we minimize the following:

$$\frac{1}{t}\sum_{i=1}^{t} \mathcal{R}_{\widehat{\mathcal{D}}_i}^{\mathcal{B}}(\ell, h) + \lambda_1\frac{1}{t^2}\sum_{i=1}^{t}\sum_{j=1}^{t} d_{\ell(\mathcal{H})}^{\mathcal{B}}(\widehat{\mathcal{D}}_i, \widehat{\mathcal{D}}_j) + \lambda_2\|W\|_F,$$

where $\lambda_1, \lambda_2$ are two hyper-parameters, and $W$ is the parameter matrix of the model. However, the term $\frac{1}{t^2}\sum_{i=1}^{t}\sum_{j=1}^{t} d_{\ell(\mathcal{H})}^{\mathcal{B}}(\widehat{\mathcal{D}}_i, \widehat{\mathcal{D}}_j)$ is a constant. Motivated by [17], we minimize the discrepancy of the training environments in the feature space of a feature extractor $f$.

Specifically, we consider $\mathcal{H} = \{c \circ f | f \in \mathcal{F}, c \in \mathcal{C}\}$; this means that the predictor $h$ consists of a classifier $c$ and a feature extractor $f$, where $\mathcal{F} = \{f : \mathcal{X} \to \mathcal{Z}\}, \mathcal{C} = \{c : \mathcal{Z} \to \mathcal{Y}\}$ and $\mathcal{Z} \subseteq \mathbb{R}^l$ is the feature space. Any feature extractor $f \in \mathcal{F}$ determines a hypothesis class $\mathcal{H}_f = \{c \circ f | c \in \mathcal{C}\}$. Given a feature extractor $f$, we apply Theorem 3.3 to the hypothesis class $\mathcal{H}_f$. Then with high probability, for any $c \in \mathcal{C}$, the target error of $h = c \circ f$ can be controlled mainly by $\frac{1}{t}\sum_{i=1}^{t}\mathcal{R}_{\widehat{\mathcal{D}}_i}^{\mathcal{B}}(\ell, c \circ f)$ and $\frac{1}{t^2}\sum_{i=1}^{t}\sum_{j=1}^{t} d_{\ell(\mathcal{H}_f)}^{\mathcal{B}}(\widehat{\mathcal{D}}_i, \widehat{\mathcal{D}}_j)$. We then aim to find $c$ and $f$ such that $h = c \circ f$ has good OOD adversarial robustness. Thus, we aim to minimize the following:

$$\underbrace{\underbrace{\frac{1}{t}\sum_{i=1}^{t} \mathcal{R}_{\widehat{\mathcal{D}}_i}^{\mathcal{B}}(\ell, c \circ f)}_{L_{\text{cls}}(\ell, c \circ f)} + \lambda_1\underbrace{\frac{1}{t^2}\sum_{i=1}^{t}\sum_{j=1}^{t} d_{\ell(\mathcal{H}_f)}^{\mathcal{B}}(\widehat{\mathcal{D}}_i, \widehat{\mathcal{D}}_j)}_{L_{\text{disc}}(\ell, f)} + \lambda_2\|W\|_F}_{L(c \circ f)}.$$

To minimize $L(c \circ f)$, we need to solve a minimax problem. For simplicity, we fix $\mathcal{B}(\cdot), \ell$ and define the following:

$$D(h, P, Q) = \left| \mathbb{E}_{(\boldsymbol{x},y)\sim P}\left[\sup_{\boldsymbol{x}'\in\mathcal{B}(\boldsymbol{x})} \ell(h(\boldsymbol{x}'), y)\right] - \mathbb{E}_{(\boldsymbol{x},y)\sim Q}\left[\sup_{\boldsymbol{x}'\in\mathcal{B}(\boldsymbol{x})} \ell(h(\boldsymbol{x}'), y)\right] \right|.$$

Since $D(h, P, P) = 0$ and $D(h, P, Q) = D(h, Q, P)$ for any $P, Q, h$, our optimization problem can be formulated as follows:

$$\min_{c\in\mathcal{C}, f\in\mathcal{F}} \max_{c_{ij}\in\mathcal{C}:1\leq i<j\leq t} L_{\text{cls}}(\ell, c \circ f) + \lambda_1\frac{2}{t(t-1)}\sum_{1\leq i<j\leq t} D(c_{ij} \circ f, \widehat{\mathcal{D}}_i, \widehat{\mathcal{D}}_j) + \lambda_2\|W\|_F.$$

Table 2: The results (%) of the algorithms. Results are presented in the form of **a/b**: here, **a** is the OOD adversarial accuracy under PGD-20 attack [44]; and **b** is the OOD adversarial accuracy under AutoAttack [12]. We conduct the $\ell_\infty$-norm attack. Since [64] do not realize MAT and LDAT for RotatedMNIST, we use X to denote the unrealized results. Best results for PGD-20 attack are shown in **bold**. For more details about the algorithms, please refer to Appendix C.1. We use RM, CM, and OH as the abbreviation of RotatedMNIST, ColoredMNIST, and OfficeHome, respectively.

| Algorithm | RM | CM | VLCS | PACS | OH | Avg |
|---|---|---|---|---|---|---|
| ERM | 0.6/0.0 | 5.8/X | 0.0/0.0 | 0.3/0.6 | 0.4/0.0 | 1.4/X |
| MLDG | 0.2/0.0 | 4.8/X | 0.0/0.0 | 0.1/0.3 | 0.6/0.1 | 1.2/X |
| CDANN | 0.9/0.0 | 8.2/X | 3.0/0.0 | 1.5/0.3 | 0.1/0.0 | 2.7/X |
| VREx | 0.2/0.0 | 6.4/X | 0.0/0.0 | 0.0/0.3 | 0.4/0.1 | 1.4/X |
| RSC | 1.0/0.0 | 3.9/X | 0.0/0.0 | 0.1/0.4 | 0.7/0.0 | 1.1/X |
| MAT | X/X | 10.7/X | 0.0/0.0 | 0.7/1.4 | 0.8/0.1 | X/X |
| LDAT | X/X | 7.9/X | 0.0/0.0 | 0.1/0.3 | 0.4/0.1 | X/X |
| AT (ours) | 93.4/**93.3** | **51.6**/X | 42.6/41.8 | **48.1**/47.6 | **30.2/29.8** | 53.2/X |
| RDANN (ours) | **93.5**/93.3 | 51.1/X | **44.9/43.9** | **48.1**/48.0 | 28.6/27.4 | **53.3**/X |

To solve the minimax optimization problem, we adopt an idea similar to that of adversarial neural networks [24] and refer to $c_{ij}$ as the discriminator. However, there are $\frac{t(t-1)}{2}$ discriminators in this case, and training with many discriminators may make the optimization process unstable [3]. We therefore opt to use the same discriminator for all $(\widehat{\mathcal{D}}_i, \widehat{\mathcal{D}}_j)$ pairs. Note that:

$$\max_{c' \in \mathcal{C}} \sum_{i<j} D(c' \circ f, \widehat{\mathcal{D}}_i, \widehat{\mathcal{D}}_j) \leq \max_{c_{ij} \in \mathcal{C}} \sum_{i<j} D(c_{ij} \circ f, \widehat{\mathcal{D}}_i, \widehat{\mathcal{D}}_j),$$

such that optimizing over a shared discriminator $c'$ is equivalent to optimizing a lower bound of the original objective. Our final optimization problem then becomes:

$$\min_{c \in \mathcal{C}, f \in \mathcal{F}} \max_{c' \in \mathcal{C}} \underbrace{L_{\text{cls}}(\ell, c \circ f)}_{L_c(w, \theta)} + \lambda_1 \underbrace{\frac{2}{t(t-1)} \sum_{1 \leq i < j \leq t} D(c' \circ f, \widehat{\mathcal{D}}_i, \widehat{\mathcal{D}}_j)}_{L_d(w', \theta)} + \lambda_2 \underbrace{\|W\|_F}_{L_{\text{reg}}(w, \theta)}.$$

where $w, w', \theta$ are parameters for $c, c', f$ respectively. We call our method robust adversarial-domain neural network (**RDANN**), the pseudo-code of which is presented in Algorithm 1.

## 5 Experiments

To verify the adversarial robustness of our algorithms, we conduct experiments on the DomainBed benchmark [26], a testbed for OOD generalization that implements consistent experimental protocols across various approaches to ensure fair comparisons. Our code is attached in the supplementary material.

### 5.1 Experimental Setup

**Datasets.** The datasets we use are RotatedMNIST [21], ColoredMNIST [4], VLCS [15], PACS [38], and OfficeHome [62]. There are several different environments in each dataset. For the DomainBed benchmark, we choose one environment as the test environment and use the others as training environments. We report the average accuracy over different choices of the test environment. Further details about the selected datasets can be found in Appendix C.2.

---

**Algorithm 1** RDANN

**Input:** the training data $\widehat{\mathcal{D}}_1, \ldots, \widehat{\mathcal{D}}_t$, the number of iterations $T$, the number of iterations for the inner maximization problem $k$, the learning rate $\eta$, the inner max learning rate $\alpha$, the tradeoff hyper-parameters $\lambda_1, \lambda_2$.

**Output:** the parameters $w^T, \theta^T$ for $c, f$.

1: initialize $w^0, \theta^0, w'^k$ randomly
2: **for** $i \leftarrow 0$ to $T - 1$ **do**
3:     set $w'^0 \leftarrow w'^k$
4:     **for** $j \leftarrow 0$ to $k - 1$ **do**
5:        $w'^{(j+1)} \leftarrow w'^j + \alpha \nabla_{w'} L_d(w'^j, \theta^i)$
6:     **end for**
7:     $L_1^i \leftarrow L_c(w^i, \theta^i) + \lambda_2 L_{\text{reg}}(w^i, \theta^i)$
8:     $L_2^i \leftarrow L_1^i + \lambda_1 L_d(w'^k, \theta^i)$
9:     $\theta^{i+1} \leftarrow \theta^i - \eta \nabla_\theta L_2^i$
10:     $w^{i+1} \leftarrow w^i - \eta \nabla_w L_1^i$
11: **end for**

---

**Backbone network.** Following [26], we use a small CNN architecture for RotatedMNIST, ColoredMNIST, and ResNet-50 [27] for VLCS, PACS, and OfficeHome.

**Hyper-parameters for adversarial attack and adversarial training.** We use $\ell_\infty$-norm attack for both adversarial attack and adversarial training. We use $\epsilon = 0.1$ for ColoredMNIST and RotatedMNIST, and $\epsilon = 4/255$ for VLCS, PACS, and OfficeHome; moreover, following [74, 75], we use PGD-10 to generate adversarial examples at the training stage and PGD-20 at the evaluation stage to avoid overfitting. The step size used to generate adversarial examples is set to be $\epsilon/4$.

More details of the experimental settings can be found in Appendix C.4.

## 5.2 Results

Table 2 presents the results of our experiments. As is clear from the table, our proposed algorithms significantly improve the OOD adversarial robustness of the model. For example, under the attack PGD-20, all other algorithms achieve **no more than** $3\%$ average adversarial accuracy, while our proposed algorithms achieve **more than** $53\%$ average adversarial accuracy. Moreover, in our datasets, the number of environments ($t$) is small. From Table 2, we can see that the overall performance of RDANN is superior to that of AT. RDANN achieves better or comparable adversarial accuracy on most datasets (except OfficeHome). The results are consistent with our claim in Remark 2: when $t = \mathcal{O}(1)$, the bound in Theorem 3.3 converges faster than that in Theorem 3.1. The detailed results for each test environment are attached in Appendix D.

**Observations.** According to the detailed results for RotatedMNIST and ColoredMNIST in Appendix D.1 and Appendix D.2, we can make the following observations (these phenomena occur in both the adversarial training setting and the standard training setting):

- For RotatedMNIST, we have six environments, each of which corresponds to a rotation angle of the original image. The rotation angle of the $i$-th environment is $i \times 15°$, $i \in \{0, 1, 2, 3, 4, 5\}$. For task $i$, we use the $i$-th environment as the test environment and the remaining environments as the training environments. Following Remark 3, we define $d_{\text{avg}}(i) := \frac{1}{5} \sum_{j \neq i} \text{angle}(i, j)$ as the average distance between the test environment and each training environment for task $i$. Then, $d_{\text{avg}}(0) = 45°$, $d_{\text{avg}}(1) = 33°$, $d_{\text{avg}}(2) = 27°$, $d_{\text{avg}}(3) = 27°$, $d_{\text{avg}}(4) = 33°$ and $d_{\text{avg}}(5) = 45°$. We define the PGD-20 adversarial accuracy of the model trained by RDANN for task $i$ as $a(i)$; then, $a(0) = 90.8$, $a(1) = 94.7$, $a(2) = 95.3$, $a(3) = 95.6$, $a(4) = 95.5$ and $a(5) = 89.0$. As $i$ increases, $d_{\text{avg}}(i)$ first decreases and then increases, while $a(i)$ first increases and then decreases. The result indicates that $d_{\text{avg}}(i)$ is anticorrelated with $a(i)$, which is consistent with the analysis in Remark 3. Note that this phenomenon occurs in all algorithms.

- For ColoredMNIST, we have three environments, each of which corresponds to a correlation between the additional channel and the label. For task $i$, $i \in \{0, 1, 2\}$, we define the correlation as $\text{cor}(i)$ and $\text{cor}(1) = 0.9$, $\text{cor}(2) = 0.8$, $\text{cor}(3) = -0.9$. Here, we define $d_{\text{avg}}(i) = \left| \frac{1}{2} \sum_{j \neq i} \text{cor}(j) - \text{cor}(i) \right|$, then $d_{\text{avg}}(1) = 0.95$, $d_{\text{avg}}(2) = 0.8$, $d_{\text{avg}}(3) = 1.75$. Similarly, we can define $a(i)$ for a given algorithm. The detailed results in Appendix D.2 imply that $d_{\text{avg}}(i)$ is anticorrelated with $a(i)$. To further understand this phenomenon, we present another toy example for ColoredMNIST with a different data model in Appendix B.

## 6 Conclusion

In this paper, we focus specifically on out-of-distribution adversarial robustness. First, we show that existing OOD generalization algorithms are easily fooled by adversarial attacks. Motivated by this, we then study the theory of the adversarial robustness of models in two different but complementary OOD settings. Based on our theory, we propose two algorithms, AT and RDANN. Extensive experiments show that our proposed algorithms can significantly improve the OOD adversarial robustness of the model.

## Acknowledgements

This work is supported by the National Natural Science Foundation of China under Grant 61976161, the Fundamental Research Funds for the Central Universities under Grant 2042022rc0016.

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

# A Proofs

In this section, we show the proofs of the results in the mainscript.

## A.1 Proofs of the Toy Example

**Lemma A.1.** *If $X \sim \mathcal{N}(\boldsymbol{\mu}, \Sigma)$ where $X \sim \mathbb{R}^n$, then for any $A \in \mathbb{R}^{m \times n}$, we have:*

$$A\boldsymbol{X} \sim \mathcal{N}(A\boldsymbol{\mu}, A\Sigma A^T)$$

*Proof of Lemma A.1.* We prove the lemma by a powerful tool, the characteristic function of random variables. The characteristic function of a random vector $\boldsymbol{X}$ is defined as:

$$\phi_{\boldsymbol{X}}(\boldsymbol{\omega}) = \mathbb{E}\left[e^{i\boldsymbol{\omega}^T \boldsymbol{X}}\right]$$

Let $\boldsymbol{Y} = A\boldsymbol{X}$, we have:

$$\phi_{\boldsymbol{Y}}(\boldsymbol{\omega}) = \mathbb{E}\left[e^{i\boldsymbol{\omega}^T \boldsymbol{Y}}\right] = \mathbb{E}\left[e^{i\boldsymbol{\omega}^T (A\boldsymbol{X})}\right] = \mathbb{E}\left[e^{i(A^T\boldsymbol{\omega})^T \boldsymbol{X}}\right] = \phi_{\boldsymbol{X}}(A^T\boldsymbol{\omega}).$$

Since $\boldsymbol{X} \sim \mathcal{N}(\boldsymbol{\mu}, \Sigma)$, we have:

$$\phi_{\boldsymbol{X}}(\boldsymbol{\omega}) = \mathbb{E}\left[e^{i\boldsymbol{\omega}^T \boldsymbol{X}}\right] = \mathbb{E}\left[e^{i\boldsymbol{\omega}^T \boldsymbol{\mu} - \frac{1}{2}\boldsymbol{\omega}^T \Sigma \boldsymbol{\omega}}\right],$$

so we have:

$$\phi_{\boldsymbol{Y}}(\boldsymbol{\omega}) = \mathbb{E}\left[e^{i(A^T\boldsymbol{\omega})^T \boldsymbol{\mu} - \frac{1}{2}(A^T\boldsymbol{\omega})^T \Sigma (A^T\boldsymbol{\omega})}\right] = \mathbb{E}\left[e^{i\boldsymbol{\omega}^T (A\boldsymbol{\mu}) - \frac{1}{2}\boldsymbol{\omega}^T (A\Sigma A^T)\boldsymbol{\omega}}\right].$$

Since the characteristic function and the distribution is one-to-one, we have: $\boldsymbol{Y}$ obeys the multi-variable Gaussian distribution and $\boldsymbol{Y} \sim \mathcal{N}(A\boldsymbol{\mu}, A\Sigma A^T)$. $\qquad\square$

**Lemma A.2.** *Let $\mathcal{D}_Y(Y = 1) = \mathcal{D}_Y(Y = -1) = \frac{1}{2}, X \in \mathbb{R}^d$ and $\mathcal{D}_{X|Y=y} = \mathcal{N}(y\boldsymbol{\mu}, \Sigma)$ where $\boldsymbol{\mu} \in \mathbb{R}^d$ and $\Sigma \in \mathbb{R}^{d \times d}$, then for any linear classifier $h_{\boldsymbol{w}} \in \mathcal{H}$:*

$$\mathcal{R}_{\mathcal{D}}^{B_p^\epsilon}(\ell_{01}, h_{\boldsymbol{w}}) = \Phi\left(\frac{\epsilon \|\boldsymbol{w}\|_{p^*} - \boldsymbol{w}^T \boldsymbol{\mu}}{\sqrt{\boldsymbol{w}^T \Sigma \boldsymbol{w}}}\right),$$

*where $\frac{1}{p} + \frac{1}{p^*} = 1$. Further more, if $p = 2$ and $\Sigma = \sigma^2 I$, then $\boldsymbol{w}^\star = \arg\min_{\boldsymbol{w}:h_{\boldsymbol{w}} \in \mathcal{H}} \mathcal{R}_{\mathcal{D}}^{B_2^\epsilon}(\ell_{01}, h_{\boldsymbol{w}})$, the weight of the most robust $h_{\boldsymbol{w}} \in \mathcal{H}$ under adversarial attack $B_2^\epsilon(\cdot)$ is:*

$$\boldsymbol{w}^\star = \boldsymbol{\mu}.$$

*Proof of Lemma A.2.* Let $h_{\boldsymbol{w}}(\boldsymbol{x}) = \text{sign}(\boldsymbol{w}^T \boldsymbol{x})$, then we have:

$$\mathcal{R}_{\mathcal{D}}^{B_p^\epsilon}(\ell_{01}, h_{\boldsymbol{w}}) = \mathop{\mathbb{E}}_{(\boldsymbol{x},y)\sim\mathcal{D}}\left\{\sup_{\boldsymbol{x}' \in B_p^\epsilon(\boldsymbol{x})} \mathbb{1}\left[h_{\boldsymbol{w}}(\boldsymbol{x}) \neq y\right]\right\}$$

$$= \frac{1}{2}\mathop{\mathbb{E}}_{\boldsymbol{x}\sim\mathcal{D}_{X|Y=1}}\left\{\sup_{\boldsymbol{x}' \in B_p^\epsilon(\boldsymbol{x})} \mathbb{1}\left[\boldsymbol{w}^T \boldsymbol{x}' < 0\right]\right\} + \frac{1}{2}\mathop{\mathbb{E}}_{\boldsymbol{x}\sim\mathcal{D}_{X|Y=-1}}\left\{\sup_{\boldsymbol{x}' \in B_p^\epsilon(\boldsymbol{x})} \mathbb{1}\left[\boldsymbol{w}^T \boldsymbol{x}' > 0\right]\right\}$$

$$= \frac{1}{2}\mathop{\mathbb{P}}_{\boldsymbol{x}\sim\mathcal{D}_{X|Y=1}}\left\{\inf_{\boldsymbol{x}' \in B_p^\epsilon(\boldsymbol{x})} \boldsymbol{w}^T \boldsymbol{x}' < 0\right\} + \frac{1}{2}\mathop{\mathbb{P}}_{\boldsymbol{x}\sim\mathcal{D}_{X|Y=-1}}\left\{\sup_{\boldsymbol{x}' \in B_p^\epsilon(\boldsymbol{x})} \boldsymbol{w}^T \boldsymbol{x}' > 0\right\}$$

$$= \frac{1}{2}\mathop{\mathbb{P}}_{\boldsymbol{x}\sim\mathcal{D}_{X|Y=1}}\left\{\boldsymbol{w}^T \boldsymbol{x} - \epsilon\|\boldsymbol{w}\|_{p^*} < 0\right\} + \frac{1}{2}\mathop{\mathbb{P}}_{\boldsymbol{x}\sim\mathcal{D}_{X|Y=-1}}\left\{\boldsymbol{w}^T \boldsymbol{x} + \epsilon\|\boldsymbol{w}\|_{p^*} > 0\right\}$$

$$\overset{a}{=} \frac{1}{2}\mathbb{P}\left[\mathcal{N}(\boldsymbol{w}^T \boldsymbol{\mu}, \boldsymbol{w}^T \Sigma \boldsymbol{w}) < \epsilon\|\boldsymbol{w}\|_{p^*}\right] + \frac{1}{2}\mathbb{P}\left[\mathcal{N}(-\boldsymbol{w}^T \boldsymbol{\mu}, \boldsymbol{w}^T \Sigma \boldsymbol{w}) > -\epsilon\|\boldsymbol{w}\|_{p^*}\right]$$

$$= \mathbb{P}\left[\mathcal{N}(\boldsymbol{w}^T \boldsymbol{\mu}, \boldsymbol{w}^T \Sigma \boldsymbol{w}) < \epsilon\|\boldsymbol{w}\|_{p^*}\right] = \Phi\left(\frac{\epsilon\|\boldsymbol{w}\|_{p^*} - \boldsymbol{w}^T \boldsymbol{\mu}}{\sqrt{\boldsymbol{w}^T \Sigma \boldsymbol{w}}}\right),$$

where $a$ is the result of Lemma A.1 when we regard $\boldsymbol{w}^T$ as $A$ in Lemma A.1.

If $p = 2$ and $\Sigma = \sigma^2 I$, then we have:

$$\mathcal{R}_{\mathcal{D}}^{B_2^\epsilon}(\ell_{01}, h_{\boldsymbol{w}}) = \Phi\left(\frac{\epsilon\|\boldsymbol{w}\|_2 - \boldsymbol{w}^T\boldsymbol{\mu}}{\sigma\|\boldsymbol{w}\|_2}\right) = \Phi\left(\frac{\epsilon}{\sigma} - \frac{\boldsymbol{w}^T\boldsymbol{\mu}}{\sigma\|\boldsymbol{w}\|_2}\right).$$

Since $\Phi(\cdot)$ is increasing, we need to minimize $\frac{\epsilon}{\sigma} - \frac{\boldsymbol{w}^T\boldsymbol{\mu}}{\sigma\|\boldsymbol{w}\|_2}$, i.e., to maximize $\frac{\boldsymbol{w}^T\boldsymbol{\mu}}{\sigma\|\boldsymbol{w}\|_2}$, it's easy to see that the maximum is achieved if we choose $\boldsymbol{w} = \boldsymbol{\mu}$, so we have $\boldsymbol{w}^\star = \boldsymbol{\mu}$. $\qquad\square$

**Lemma A.3.** *Let $\mathcal{D}_1, \mathcal{D}_2$ to be defined as: $\mathcal{D}_{1,Y}(Y = 1) = \mathcal{D}_{1,Y}(Y = -1) = \mathcal{D}_{2,Y}(Y = -1) = \mathcal{D}_{2,Y}(Y = -1)\frac{1}{2}, X \in \mathbb{R}^d$ and $\mathcal{D}_{1,X|Y=y} = \mathcal{N}(y\boldsymbol{\mu}_1, \sigma^2 I), \mathcal{D}_{2,X|Y=y} = \mathcal{N}(y\boldsymbol{\mu}_2, \sigma^2 I)$ where $\boldsymbol{\mu}_1, \boldsymbol{\mu}_2 \in \mathbb{R}^d, \|\boldsymbol{\mu}_1\|_2 = \|\boldsymbol{\mu}_2\|_2$ and $\sigma \in \mathbb{R}_+$, let $\mathcal{D} = \frac{1}{2}\mathcal{D}_1 + \frac{1}{2}\mathcal{D}_2$ and suppose $\alpha$ is the angle between $\boldsymbol{\mu}_1$ and $\boldsymbol{\mu}_2$, if $0 < \alpha \le 2\arccos\frac{\epsilon}{\|\boldsymbol{\mu}_1\|_2}$, then $\boldsymbol{w}^\star = \boldsymbol{\mu}_1 + \boldsymbol{\mu}_2$ achieves the minimal robust risk.*

*Proof of Lemma A.3.* To simplify the notation, we denote $\mathcal{R}_{\mathcal{D}}^{B_2^\epsilon}(\ell_{01}, h_{\boldsymbol{w}})$ by $\mathcal{R}(\boldsymbol{w})$. Then by Lemma A.2 we have:

$$\mathcal{R}(\boldsymbol{w}) = \frac{1}{2}\mathcal{R}_{\mathcal{D}_1}^{B_2^\epsilon}(\ell_{01}, h_{\boldsymbol{w}}) + \frac{1}{2}\mathcal{R}_{\mathcal{D}_2}^{B_2^\epsilon}(\ell_{01}, h_{\boldsymbol{w}}) = \frac{1}{2}\Phi\left(\frac{\epsilon\|\boldsymbol{w}\|_2 - \boldsymbol{w}^T\boldsymbol{\mu}_1}{\sigma\|\boldsymbol{w}\|_2}\right) + \frac{1}{2}\Phi\left(\frac{\epsilon\|\boldsymbol{w}\|_2 - \boldsymbol{w}^T\boldsymbol{\mu}_2}{\sigma\|\boldsymbol{w}\|_2}\right),$$

where $\Phi(u) = \frac{1}{\sqrt{2\pi}}\int_{-\infty}^{u} e^{-\frac{y^2}{2}}\, dy$. The proof is divided into two steps. In the first step, we prove that for any weight $\boldsymbol{w}$, we have $\mathcal{R}(\boldsymbol{\mu}_P) \le \mathcal{R}(\boldsymbol{\mu})$, where $\boldsymbol{\mu}_P$ is the project of $\boldsymbol{\mu}$ onto the space span $\langle \boldsymbol{\mu}_1, \boldsymbol{\mu}_2 \rangle$. Then we can reduce our analysis to the weights in the space span $\langle \boldsymbol{\mu}_1, \boldsymbol{\mu}_2 \rangle$.

**Step 1.** For any $\boldsymbol{w} \in \mathbb{R}^d$, we take a direct sum decomposition as $\boldsymbol{w} = \boldsymbol{w}_s + \boldsymbol{w}_t$ where $\boldsymbol{w}_s \in$ span $\langle \boldsymbol{\mu}_1, \boldsymbol{\mu}_2 \rangle$ and $\boldsymbol{w}_t \in$ span $\langle \boldsymbol{\mu}_1, \boldsymbol{\mu}_2 \rangle^\perp$, the orthogonal complement space of span $\langle \boldsymbol{\mu}_1, \boldsymbol{\mu}_2 \rangle$, then we have:

$$\boldsymbol{w}^T\boldsymbol{\mu}_1 = \boldsymbol{w}_s^T\boldsymbol{\mu}_1 + \boldsymbol{w}_t^T\boldsymbol{\mu}_1 = \boldsymbol{w}_s^T\boldsymbol{\mu}_1$$

Similarly, we have $\boldsymbol{w}^T\boldsymbol{\mu}_2 = \boldsymbol{w}_s^T\boldsymbol{\mu}_2$. Since $\boldsymbol{w}_s \perp \boldsymbol{w}_t$, then we have $\|\boldsymbol{w}\|_2^2 = \|\boldsymbol{w}_s\|_2^2 + \|\boldsymbol{w}_t\|_2^2$, so for any $\boldsymbol{w} \in \mathbb{R}^d$, we have:

$$\begin{aligned}
\mathcal{R}(\boldsymbol{w}) &= \frac{1}{2}\Phi\left(\frac{\epsilon}{\sigma} - \frac{\boldsymbol{w}^T\boldsymbol{\mu}_1}{\sigma\|\boldsymbol{w}\|_2}\right) + \frac{1}{2}\Phi\left(\frac{\epsilon}{\sigma} - \frac{\boldsymbol{w}^T\boldsymbol{\mu}_2}{\sigma\|\boldsymbol{w}\|_2}\right)\\
&= \frac{1}{2}\Phi\left(\frac{\epsilon}{\sigma} - \frac{\boldsymbol{w}_s^T\boldsymbol{\mu}_1}{\sigma\sqrt{\|\boldsymbol{w}_s\|_2^2 + \|\boldsymbol{w}_t\|_2^2}}\right) + \frac{1}{2}\Phi\left(\frac{\epsilon}{\sigma} - \frac{\boldsymbol{w}_s^T\boldsymbol{\mu}_2}{\sigma\sqrt{\|\boldsymbol{w}_s\|_2^2 + \|\boldsymbol{w}_t\|_2^2}}\right)
\end{aligned}$$

So it is obvious that $\mathcal{R}(\boldsymbol{w}_s) \le \mathcal{R}(\boldsymbol{w})$.

**Step 2.** According to the above results, we consider $\boldsymbol{w} \in$ span $\langle \boldsymbol{\mu}_1, \boldsymbol{\mu}_2 \rangle$. Suppose the angle between $\boldsymbol{\mu}_1, \boldsymbol{\mu}_2$ is $\alpha$, note that $\mathcal{R}(\boldsymbol{w})$ does not depend on the length of $\boldsymbol{w}$, so without loss of generality, we assume $\|\boldsymbol{w}\|_2 = 1$ here. Suppose the angle between $\boldsymbol{w}$ and $\boldsymbol{\mu}_1$ is $\theta$, then the angle between $\boldsymbol{w}$ and $\boldsymbol{\mu}_2$ is $\alpha - \theta$, then $\boldsymbol{w}$ is uniquely determined by $\theta$, we denote $\mathcal{R}(\theta) = \mathcal{R}(\boldsymbol{w})$, then we have:

$$\mathcal{R}(\theta) = \frac{1}{2}\Phi\left(\frac{\epsilon}{\sigma} - \frac{\|\boldsymbol{\mu}_1\|_2\cos\theta}{\sigma}\right) + \frac{1}{2}\Phi\left(\frac{\epsilon}{\sigma} - \frac{\|\boldsymbol{\mu}_1\|_2\cos(\alpha - \theta)}{\sigma}\right)$$

By the rule of derivation for composition function, we have:

$$\begin{aligned}
\mathcal{R}'(\theta) &= \frac{1}{2\sqrt{2\pi}}\exp\left\{-\frac{(\epsilon - \|\boldsymbol{\mu}_1\|_2\cos\theta)^2}{2\sigma^2}\right\}\frac{\|\boldsymbol{\mu}_1\|_2}{\sigma}\sin\theta\\
&\quad - \frac{1}{2\sqrt{2\pi}}\exp\left\{-\frac{(\epsilon - \|\boldsymbol{\mu}_1\|_2\cos(\alpha-\theta))^2}{2\sigma^2}\right\}\frac{\|\boldsymbol{\mu}_1\|_2}{\sigma}\sin(\alpha - \theta)\\
&= \frac{\|\boldsymbol{\mu}_1\|_2}{2\sqrt{2\pi}\sigma}\left[\exp\left\{-\frac{(\epsilon - \|\boldsymbol{\mu}_1\|_2\cos\theta)^2}{2\sigma^2}\right\}\sin\theta\right.\\
&\quad \left. - \exp\left\{-\frac{(\epsilon - \|\boldsymbol{\mu}_1\|_2\cos(\alpha-\theta))^2}{2\sigma^2}\right\}\sin(\alpha - \theta)\right].
\end{aligned}$$

It is easy to see that when $\theta = \frac{\alpha}{2}$ or $\theta = \frac{\alpha}{2} + \pi$, $\mathcal{R}'(\theta) = 0$, since:

$$\mathcal{R}'(\frac{\alpha}{2}) = \frac{\|\boldsymbol{\mu}_1\|_2}{2\sqrt{2\pi}\sigma}\left[\exp\left\{-\frac{(\epsilon - \|\boldsymbol{\mu}_1\|_2 \cos\frac{\alpha}{2})^2}{2\sigma^2}\right\}\sin\frac{\alpha}{2} - \exp\left\{-\frac{(\epsilon - \|\boldsymbol{\mu}_1\|_2 \cos\frac{\alpha}{2})^2}{2\sigma^2}\right\}\sin\frac{\alpha}{2}\right]$$
$$= 0$$

Similarly we have $\mathcal{R}'(\frac{\alpha}{2} + \pi) = 0$. Next we prove $\mathcal{R}(\theta)$ achieves the minimum at $\theta = \frac{\alpha}{2}$, taking the second order derivation of $\theta$, we have:

$$\frac{2\sqrt{2\pi}\sigma}{\|\boldsymbol{\mu}_1\|_2}\mathcal{R}''(\theta) = \exp\left\{-\frac{(\epsilon - \|\boldsymbol{\mu}_1\|_2 \cos\theta)^2}{2\sigma^2}\right\}\cdot\left(-\frac{\|\boldsymbol{\mu}_1\|_2\,(\epsilon - \|\boldsymbol{\mu}_1\|_2 \cos\theta)\sin^2\theta}{\sigma^2}\right)$$
$$+ \exp\left\{-\frac{(\epsilon - \|\boldsymbol{\mu}_1\|_2 \cos\theta)^2}{2\sigma^2}\right\}\cos\theta$$
$$- \exp\left\{-\frac{(\epsilon - \|\boldsymbol{\mu}_1\|_2 \cos(\alpha - \theta))^2}{2\sigma^2}\right\}\cdot\left(\frac{\|\boldsymbol{\mu}_1\|_2(\epsilon - \|\boldsymbol{\mu}_1\|_2 \cos(\alpha - \theta))\sin^2(\alpha - \theta)}{\sigma^2}\right)$$
$$+ \exp\left\{-\frac{(\epsilon - \|\boldsymbol{\mu}_1\|_2 \cos(\alpha - \theta))^2}{2\sigma^2}\right\}\cos(\alpha - \theta)$$
$$= \exp\left\{-\frac{(\epsilon - \|\boldsymbol{\mu}_1\|_2 \cos\theta)^2}{2\sigma^2}\right\}\left(\cos\theta - \frac{\|\boldsymbol{\mu}_1\|_2\,(\epsilon - \|\boldsymbol{\mu}_1\|_2 \cos\theta)\sin^2\theta}{\sigma^2}\right)$$
$$+ \exp\left\{-\frac{(\epsilon - \|\boldsymbol{\mu}_1\|_2 \cos(\alpha - \theta))^2}{2\sigma^2}\right\}\left(\cos(\alpha - \theta) - \frac{\|\boldsymbol{\mu}_1\|_2(\epsilon - \|\boldsymbol{\mu}_1\|_2 \cos(\alpha - \theta))\sin^2(\alpha - \theta)}{\sigma^2}\right)$$

Then we have:

$$\frac{2\sqrt{2\pi}\sigma}{\|\boldsymbol{\mu}_1\|_2}\mathcal{R}''(\frac{\alpha}{2}) = \exp\left\{-\frac{(\epsilon - \|\boldsymbol{\mu}_1\|_2 \cos\frac{\alpha}{2})^2}{2\sigma^2}\right\}\left(\cos\frac{\alpha}{2} - \frac{\|\boldsymbol{\mu}_1\|_2\,(\epsilon - \|\boldsymbol{\mu}_1\|_2 \cos\frac{\alpha}{2})\sin^2\frac{\alpha}{2}}{\sigma^2}\right)$$
$$+ \exp\left\{-\frac{(\epsilon - \|\boldsymbol{\mu}_1\|_2 \cos\frac{\alpha}{2})^2}{2\sigma^2}\right\}\left(\cos\frac{\alpha}{2} - \frac{\|\boldsymbol{\mu}_1\|_2(\epsilon - \|\boldsymbol{\mu}_1\|_2 \cos\frac{\alpha}{2})\sin^2\frac{\alpha}{2}}{\sigma^2}\right)$$
$$= \frac{2}{\sigma^2}\exp\left\{-\frac{(\epsilon - \|\boldsymbol{\mu}_1\|_2 \cos\frac{\alpha}{2})^2}{2\sigma^2}\right\}\left(\sigma^2\cos\frac{\alpha}{2} - \|\boldsymbol{\mu}_1\|_2(\epsilon - \|\boldsymbol{\mu}_1\|_2 \cos\frac{\alpha}{2})\sin^2\frac{\alpha}{2}\right)$$

Now, for simplicity, let $k = \|\boldsymbol{\mu}_1\|_2, x = \cos\frac{\alpha}{2}$, then $\sin^2\frac{\alpha}{2} = 1 - x^2$, then we consider $f(x) = \sigma^2 x - k(\epsilon - kx)(1 - x^2) = \sigma^2\cos\frac{\alpha}{2} - \|\boldsymbol{\mu}_1\|_2(\epsilon - \|\boldsymbol{\mu}_1\|_2 \cos\frac{\alpha}{2})\sin^2\frac{\alpha}{2}$, where $\alpha \in (0, \pi)$, so $x \in (0, 1)$. We have:

$$f(x) = \sigma^2 x - k(\epsilon - kx)(1 - x^2) = \sigma^2 x - k\epsilon + k^2 x + k\epsilon x^2 - k^2 x^3$$
$$= k^2(x - x^3) - k\epsilon(1 - x^2) + \sigma^2 x = k^2 x(1 - x^2) - k\epsilon(1 - x^2) + \sigma^2 x$$
$$= k(1 - x^2)(kx - \epsilon) + \sigma^2 x.$$

Since $x \in (0, 1)$, then $\sigma^2 x > 0$ and $k(1 - x^2) > 0$, by the assumption, $0 < \alpha \leq 2\arccos\frac{\epsilon}{\|\boldsymbol{\mu}_1\|_2} = 2\arccos\frac{\epsilon}{k}$, so $x = \cos\frac{\alpha}{2} \geq \frac{\epsilon}{k}$, so $kx - \epsilon \geq 0$, so we have $f(x) > 0$, which means that $\mathcal{R}''(\frac{\alpha}{2}) > 0$, so we know that $\theta = \frac{\alpha}{2}$ achieves the minimum of $\mathcal{R}(\theta)$, which means that the direction $\boldsymbol{w}^\star = \boldsymbol{\mu}_1 + \boldsymbol{\mu}_2$ achieves the minimum of $\mathcal{R}(\boldsymbol{w})$. Similarly we can show that $\mathcal{R}(\theta)$ achieves its maximum at $\theta = \frac{\alpha}{2} + \pi$, which means that the direction $-(\boldsymbol{\mu}_1 + \boldsymbol{\mu}_2)$ achieves the maximum of $\mathcal{R}(\boldsymbol{w})$ in span $\langle\boldsymbol{\mu}_1, \boldsymbol{\mu}_2\rangle$.

$\square$

**Theorem 3.4.** *Consider the setting in Example 1 and suppose that $\boldsymbol{\mu}$ lies in the 2-dimensional subspace $\mathfrak{R}$ in Theorem A.4 (see Appendix A.1 for details about $\mathfrak{R}$ and see Figure 1 for intuitive illustration). Let $\ell_{01}(x, y) = \mathbb{1}[x \neq y]$, where $\mathbb{1}[\cdot]$ is the indicator function and $B_p^r(\boldsymbol{x}) = \{\boldsymbol{x}' : \|\boldsymbol{x} - \boldsymbol{x}'\|_p \leq r\}$, consider training with hypothesis class $\mathcal{H} = \{h_{\boldsymbol{w}} : h_{\boldsymbol{w}}(x) = \text{sign}(\boldsymbol{w}^T x), \boldsymbol{w} \in \mathbb{R}^d\}$, and denote $\mathcal{D}^{(ij)} = \frac{1}{2}\mathcal{D}^{(i)} + \frac{1}{2}\mathcal{D}^{(j)}, i, j \in \{0, 1, 2\}, i \neq j$. Consider the $\ell_2$-norm adversarial attack with radius $\epsilon$, for notation convenience, let $\widetilde{\mathcal{R}}_{ij}(\boldsymbol{w}) = \mathcal{R}_{\mathcal{D}^{(ij)}}^{B_2^\epsilon}(\ell_{01}, h_{\boldsymbol{w}})$ and $\widetilde{\mathcal{R}}_i(\boldsymbol{w}) = \mathcal{R}_{\mathcal{D}^{(i)}}^{B_2^\epsilon}(\ell_{01}, h_{\boldsymbol{w}})$, let $\Phi(\cdot)$ be the distribution function of the standard normal distribution, let $\alpha$ denote the angle between $\boldsymbol{\mu}$ and $\underline{Q}\boldsymbol{\mu}$, which is the rotation angle of $\underline{Q}$ in subspace $\mathfrak{R}$, suppose that $0 < \alpha \leq \arccos\frac{\epsilon}{\|\boldsymbol{\mu}_1\|_2}$, then we have:*

1. *If we train with $\mathcal{D}^{(0)}$ and $\mathcal{D}^{(1)}$ under $\mathcal{H}$, let $\boldsymbol{w}_{(01)} = \boldsymbol{\mu} + \underline{Q}\boldsymbol{\mu}$, which achieves the minimum of $\widetilde{\mathcal{R}}_{01}(\boldsymbol{w})$, then the robust accuracy of $\boldsymbol{w}_{(01)}$ on the test distribution $\mathcal{D}^{(2)}$ is:*

$$\widetilde{\mathcal{R}}_2(\boldsymbol{w}_{(01)}) = \Phi\left(\frac{\epsilon}{\sigma} - \frac{(\boldsymbol{\mu} + \underline{Q}\boldsymbol{\mu})^T \underline{Q}^2 \boldsymbol{\mu}}{\sigma \|\boldsymbol{\mu} + \underline{Q}\boldsymbol{\mu}\|_2}\right).$$

2. *If we train with $\mathcal{D}^{(0)}$ and $\mathcal{D}^{(2)}$ under $\mathcal{H}$, let $\boldsymbol{w}_{(02)} = \boldsymbol{\mu} + \underline{Q}^2\boldsymbol{\mu}$, which achieves the minimum of $\widetilde{\mathcal{R}}_{02}(\boldsymbol{w})$, then the robust accuracy of $\boldsymbol{w}_{(02)}$ on the test distribution $\mathcal{D}^{(1)}$ is:*

$$\widetilde{\mathcal{R}}_1(\boldsymbol{w}_{(02)}) = \Phi\left(\frac{\epsilon}{\sigma} - \frac{(\boldsymbol{\mu} + \underline{Q}^2\boldsymbol{\mu})^T \underline{Q}\boldsymbol{\mu}}{\sigma \|\boldsymbol{\mu} + \underline{Q}^2\boldsymbol{\mu}\|_2}\right).$$

3. *If we train with $\mathcal{D}^{(1)}$ and $\mathcal{D}^{(2)}$ under $\mathcal{H}$, then $\boldsymbol{w}_{(12)} = \underline{Q}\boldsymbol{\mu} + \underline{Q}^2\boldsymbol{\mu}$, which achieves the minimum of $\widetilde{\mathcal{R}}_{12}(\boldsymbol{w})$, then the robust accuracy of $\boldsymbol{w}_{(12)}$ on the test distribution $\mathcal{D}^{(0)}$ is:*

$$\widetilde{\mathcal{R}}_0(\boldsymbol{w}_{(12)}) = \Phi\left(\frac{\epsilon}{\sigma} - \frac{(\underline{Q}\boldsymbol{\mu} + \underline{Q}^2\boldsymbol{\mu})^T \boldsymbol{\mu}}{\sigma \|\underline{Q}\boldsymbol{\mu} + \underline{Q}^2\boldsymbol{\mu}\|_2}\right).$$

*What's more, we have $\widetilde{\mathcal{R}}_1(\boldsymbol{w}_{(02)}) < \widetilde{\mathcal{R}}_2(\boldsymbol{w}_{(01)}) = \widetilde{\mathcal{R}}_0(\boldsymbol{w}_{(12)})$.*

*Proof of Theorem 3.4.* The values of $\widetilde{\mathcal{R}}_0(\boldsymbol{w}_{(12)}), \widetilde{\mathcal{R}}_1(\boldsymbol{w}_{(02)}), \widetilde{\mathcal{R}}_2(\boldsymbol{w}_{(01)})$ is a direct result of Lemma A.2 and Lemma A.3. Next we compare the three adversarial risks.

Let $C_{(01)} = \frac{(\boldsymbol{\mu}+\underline{Q}\boldsymbol{\mu})^T\underline{Q}^2\boldsymbol{\mu}}{\sigma\|\boldsymbol{\mu}+\underline{Q}\boldsymbol{\mu}\|_2}, C_{(02)} = \frac{(\boldsymbol{\mu}+\underline{Q}^2\boldsymbol{\mu})^T\underline{Q}\boldsymbol{\mu}}{\sigma\|\boldsymbol{\mu}+\underline{Q}^2\boldsymbol{\mu}\|_2}, C_{(12)} = \frac{(\underline{Q}\boldsymbol{\mu}+\underline{Q}^2\boldsymbol{\mu})^T\boldsymbol{\mu}}{\sigma\|\underline{Q}\boldsymbol{\mu}+\underline{Q}^2\boldsymbol{\mu}\|_2}$, i.e., the subtractor in $\widetilde{\mathcal{R}}_2(\boldsymbol{w}_{(01)}), \widetilde{\mathcal{R}}_1(\boldsymbol{w}_{(02)}), \widetilde{\mathcal{R}}_0(\boldsymbol{w}_{(12)})$ respectively. Since $\Phi(u) = \frac{1}{\sqrt{2\pi}}\int_{-\infty}^u e^{-\frac{y^2}{2}} du$ is monotonely increasing, to judge which of the three robust errors in Theorem 3.4 is the smallest, we need only to judge which of the corresponding $C_{(ij)}$ is the largest.

Firstly, let $Q_e$ be the matrix of $\underline{Q}$ under the orthonormal basis $\boldsymbol{e}_1, \cdots, \boldsymbol{e}_d$ where $(\boldsymbol{e}_i)_j = \delta_{ij}$ and $\delta_{ij} = 1$ if $i = j$, $\delta_{ij} = 0$ otherwise. Then we have $\underline{Q}\boldsymbol{\mu} = Q_e\boldsymbol{\mu}$ and we can see that $C_{(12)} = \frac{(Q_e\boldsymbol{\mu}+Q_e^2\boldsymbol{\mu})^T\boldsymbol{\mu}}{\sigma\|Q_e\boldsymbol{\mu}+Q_e^2\boldsymbol{\mu}\|_2} = \frac{\boldsymbol{\mu}^TQ_e^T\boldsymbol{\mu}+\boldsymbol{\mu}^TQ_e^TQ_e^T\boldsymbol{\mu}}{\sigma\|Q_e(\boldsymbol{\mu}+Q_e\boldsymbol{\mu})\|_2} \overset{a}{=} \frac{\boldsymbol{\mu}^TQ_e\boldsymbol{\mu}+\boldsymbol{\mu}^TQ_e^2\boldsymbol{\mu}}{\sigma\|\boldsymbol{\mu}+Q_e\boldsymbol{\mu}\|_2} = \frac{\boldsymbol{\mu}^TQ_e^TQ_eQ_e\boldsymbol{\mu}+\boldsymbol{\mu}^TQ_e^2\boldsymbol{\mu}}{\sigma\|\boldsymbol{\mu}+Q_e\boldsymbol{\mu}\|_2} = \frac{(\boldsymbol{\mu}+Q_e\boldsymbol{\mu})^TQ_e^2\boldsymbol{\mu}}{\sigma\|\boldsymbol{\mu}+Q_e\boldsymbol{\mu}\|_2} = C_{(01)}$ (where $a$ is from the fact that $\|Q_e\boldsymbol{x}\|_2 = \|\boldsymbol{x}\|_2, \forall\boldsymbol{x}$ and $\boldsymbol{x}^TA\boldsymbol{x} = \boldsymbol{x}^TA^T\boldsymbol{x}, \forall\boldsymbol{x}, A$). So $\widetilde{\mathcal{R}}_2(w_{(01)}) = \widetilde{\mathcal{R}}_0(w_{(12)})$.

To compare which one of $\widetilde{\mathcal{R}}_1(\boldsymbol{w}_{(02)})$ and $\widetilde{\mathcal{R}}_2(\boldsymbol{w}_{(01)})$ is smaller, we note that $\boldsymbol{\mu}$ lies in the 2-dimensional subspace $\mathfrak{R} \in \mathbb{R}^d$ in Theorem A.4 as specified in Theorem 3.4, so there exists a scalar $k \in \mathbb{R}$ such that $\boldsymbol{\mu} + \underline{Q}^2\boldsymbol{\mu} = k\,\underline{Q}\boldsymbol{\mu}$, and we know that $\underline{Q}\boldsymbol{\mu} \in \arg\min_{\boldsymbol{w}:h_{\boldsymbol{w}}\in\mathcal{H}} \widetilde{\mathcal{R}}_{02}(\boldsymbol{w})$ (by Lemma A.2), so we know that $\widetilde{\mathcal{R}}_1(\boldsymbol{w}_{(02)}) \leq \widetilde{\mathcal{R}}_2(\boldsymbol{w}_{(01)}) = \widetilde{\mathcal{R}}_0(\boldsymbol{w}_{(12)})$ since $\inf_{\boldsymbol{w}:h_{\boldsymbol{w}}\in\mathcal{H}} \widetilde{\mathcal{R}}_0(\boldsymbol{w}) = \inf_{\boldsymbol{w}:h_{\boldsymbol{w}}\in\mathcal{H}} \widetilde{\mathcal{R}}_1(\boldsymbol{w}) = \inf_{\boldsymbol{w}:h_{\boldsymbol{w}}\in\mathcal{H}} \widetilde{\mathcal{R}}_2(\boldsymbol{w})$ (which is obvious according to Lemma A.2). And following the same direction of the proof of Theorem A.4, it's easy to see that $\boldsymbol{w}_{(01)} = \boldsymbol{\mu} + \underline{Q}\boldsymbol{\mu}$ is not in the same direction of $\underline{Q}^2\boldsymbol{\mu}$ (the weight attains smallest robust error on $\mathcal{D}^{(2)}$), so $\widetilde{\mathcal{R}}_2(\boldsymbol{w}_{(01)}) > \inf_{\boldsymbol{w}:h_{\boldsymbol{w}}\in\mathcal{H}} \widetilde{\mathcal{R}}_2(\boldsymbol{w})$, so we have $\widetilde{\mathcal{R}}_1(\boldsymbol{w}_{(02)}) < \widetilde{\mathcal{R}}_2(\boldsymbol{w}_{(01)}) = \widetilde{\mathcal{R}}_0(\boldsymbol{w}_{(12)})$. $\qquad\square$

**Theorem A.4.** *Let $\underline{Q}$ be a rotation transformation on $\mathbb{R}^d$ and $\underline{Q} \neq \underline{I}$, then there exists at least one 2-dimensional subspace $\mathfrak{R} \in \mathbb{R}^d$ such that there exists a scalar $k$ s.t. $\boldsymbol{r} + \underline{Q}^2\boldsymbol{r} = k\boldsymbol{r}$ for any $\boldsymbol{r} \in \mathfrak{R}$.*

*Proof of Theorem A.4.* By the results of matrix theory, we know that since $\underline{Q}$ is an orthonormal transformation, then there exists an orthonormal basis $\boldsymbol{\eta}_1, \ldots, \boldsymbol{\eta}_d$ under which the corresponding matrix of $\underline{Q}$, $Q$, has the form:

$$Q = \text{diag}\left\{R_1, \ldots, R_m, \lambda_1, \ldots, \lambda_r\right\},$$

where $\lambda_i = 1$ or $-1$, $i = 1, 2, \cdots, r, 0 \leq r \leq d$; $R_j = \begin{bmatrix} \cos\theta_j & -\sin\theta_j \\ \sin\theta_j & \cos\theta_j \end{bmatrix}$, $0 < \theta_j < \pi, j = 1, 2, \cdots, m, 0 \leq m \leq \frac{d}{2}$.

Since $\underline{Q}$ is a rotation transformation, $Q$ is a rotation matrix, so $\lambda_1 = \cdots = \lambda_r = 1$. Once given the basis, linear transformations on $\mathbb{R}^d$ and matrices in $\mathbb{R}^{d \times d}$ has a one-to-one correspondence, so $Q \neq \underline{I}$ tells us that $Q \neq I$, where $I$ is the matrix of $\underline{I}$ under the orthonormal basis $\boldsymbol{\eta}_1, \ldots \boldsymbol{\eta}_d$. So we have $m \geq 1$ in our case, without loss of generality we consider the subspace $\mathfrak{R} = \mathrm{span} \langle \boldsymbol{\eta}_1, \boldsymbol{\eta}_2 \rangle$ be the subspace spanned by the vectors $\{\boldsymbol{\eta}_1, \boldsymbol{\eta}_2\}$.

For any $\boldsymbol{\alpha} \in \mathfrak{R}$, the coordinates of $\boldsymbol{\alpha}$ under the basis $\boldsymbol{\eta}_1, \ldots \boldsymbol{\eta}_d$ is then $[\alpha_1, \alpha_2, 0, \cdots, 0]^T$ and the coordinates of $\underline{Q}\boldsymbol{\alpha}, \underline{Q}^2\boldsymbol{\alpha}$ is $Q[\alpha_1, \alpha_2, 0, \cdots, 0]^T, Q^2[\alpha_1, \alpha_2, 0, \cdots, 0]^T$ respectively. We now prove that there exists $k = 2\cos\theta_1$ such that $[\alpha_1, \alpha_2, 0, \cdots, 0]^T + Q^2[\alpha_1, \alpha_2, 0, \cdots, 0]^T = kQ[\alpha_1, \alpha_2, 0, \cdots, 0]^T$. We have:

$$Q \begin{bmatrix} \alpha_1 \\ \alpha_2 \\ 0 \\ \vdots \\ 0 \end{bmatrix} = \begin{bmatrix} R_1 \begin{pmatrix} \alpha_1 \\ \alpha_2 \end{pmatrix} \\ 0 \\ \vdots \\ 0 \end{bmatrix} = \begin{bmatrix} \alpha_1 \cos\theta_1 - \alpha_2 \sin\theta_1 \\ \alpha_1 \sin\theta_1 + \alpha_2 \cos\theta_1 \\ 0 \\ \vdots \\ 0 \end{bmatrix}, \text{ and}$$

$$Q^2 \begin{bmatrix} \alpha_1 \\ \alpha_2 \\ 0 \\ \vdots \\ 0 \end{bmatrix} = \begin{bmatrix} R_1^2 \begin{pmatrix} \alpha_1 \\ \alpha_2 \end{pmatrix} \\ 0 \\ \vdots \\ 0 \end{bmatrix} = \begin{bmatrix} \alpha_1 \cos 2\theta_1 - \alpha_2 \sin 2\theta_1 \\ \alpha_1 \sin 2\theta_1 + \alpha_2 \cos 2\theta_1 \\ 0 \\ \vdots \\ 0 \end{bmatrix}.$$

So the coordinates of $\boldsymbol{\alpha} + \underline{Q}^2\boldsymbol{\alpha}$ is:

$$\begin{bmatrix} \alpha_1 \\ \alpha_2 \\ 0 \\ \vdots \\ 0 \end{bmatrix} + Q^2 \begin{bmatrix} \alpha_1 \\ \alpha_2 \\ 0 \\ \vdots \\ 0 \end{bmatrix} = \begin{bmatrix} \alpha_1 (\cos 2\theta_1 + 1) - \alpha_2 \sin 2\theta_1 \\ \alpha_1 \sin 2\theta_1 + \alpha_2 (\cos 2\theta_1 + 1) \\ 0 \\ \vdots \\ 0 \end{bmatrix} = \begin{bmatrix} 2\alpha_1 \cos^2\theta_1 - 2\alpha_2 \sin\theta_1 \cos\theta_1 \\ 2\alpha_2 \sin\theta_1 \cos\theta_1 + 2\alpha_1 \cos^2\theta_1 \\ 0 \\ \vdots \\ 0 \end{bmatrix}$$

$$= 2\cos\theta_1 \begin{bmatrix} \alpha_1 \cos\theta_1 - \alpha_2 \sin\theta_1 \\ \alpha_1 \sin\theta_1 + \alpha_2 \cos\theta_1 \\ 0 \\ \vdots \\ 0 \end{bmatrix} = k \begin{bmatrix} \alpha_1 \cos\theta_1 - \alpha_2 \sin\theta_1 \\ \alpha_1 \sin\theta_1 + \alpha_2 \cos\theta_1 \\ 0 \\ \vdots \\ 0 \end{bmatrix},$$

which is just $k$ times of the coordinates of $\underline{Q}\boldsymbol{\alpha}$. So we get $\boldsymbol{\alpha} + \underline{Q}^2\boldsymbol{\alpha} = k\underline{Q}\boldsymbol{\alpha}$ for any $\boldsymbol{\alpha} \in \mathfrak{R}$. $\square$

### A.2 Proofs of the Average Case

**Theorem 3.1.** *Suppose the loss function is bounded, i.e., $\ell \in [0, U]$, then with probability at least $1 - \delta$ over the sampling of $\widehat{\mathcal{D}}_1, \cdots, \widehat{\mathcal{D}}_t$, for all $h \in \mathcal{H}$, we have:*

$$\mathcal{L}_p^{\mathcal{B}}(\ell, h) \leq \mathcal{L}_{\widehat{\mathcal{D}}}^{\mathcal{B}}(\ell, h) + 2\mathfrak{R}_t(\widetilde{\mathcal{G}}) + 2\mathfrak{R}_{tn}(\widetilde{\mathcal{G}}) + 3U\sqrt{\frac{\ln 4/\delta}{2t}} + 3U\sqrt{\frac{\ln 4/\delta}{2tn}},$$

*where*

$$\widetilde{\mathcal{G}} = \left\{ g_h : \mathcal{X} \times \mathcal{Y} \to \mathbb{R}_+ \middle| g_h(\boldsymbol{x}, y) = \sup_{\boldsymbol{x}' \in \mathcal{B}(\boldsymbol{x})} \ell(h(\boldsymbol{x}', y)), h \in \mathcal{H} \right\}.$$

*Proof of Theorem 3.1.* Our proof consists of 2 steps, our 2-step uniform convergence analysis is different from (but based on) the standard Rademacher complexity generalization error bounds.

**Step 1, upper bound the difference between $\mathcal{L}_p^{\mathcal{B}}(\ell, h)$ and $\mathcal{L}_{\widehat{p}}^{\mathcal{B}}(\ell, h)$ for any $h \in \mathcal{H}$.**

Let $\mathfrak{D} = \{\mathcal{D}_1, \cdots, \mathcal{D}_t\}$, we define:

$$\Phi^{\mathcal{B}}(\mathfrak{D}) = \sup_{h \in \mathcal{H}} \left[ \mathcal{L}_p^{\mathcal{B}}(\ell, h) - \mathcal{L}_{\hat{p}}^{\mathcal{B}}(\ell, h) \right].$$

Since $\ell \in [0, U]$, changing one of the $\mathcal{D}_i$ in $\mathfrak{D}$ will lead to at most $\frac{U}{t}$ change in $\Phi^{\mathcal{B}}(\mathfrak{D})$. So by McDiarmid's inequality we have:

$$\mathbb{P}\left[ \Phi^{\mathcal{B}}(\mathfrak{D}) - \mathbb{E}(\Phi^{\mathcal{B}}(\mathfrak{D})) \right] \le \exp\left\{ -\frac{2t\epsilon^2}{U^2} \right\},$$

so with probability at least $1 - \frac{\delta}{4}$ over the choice of $\mathfrak{D}$, we have:

$$\Phi^{\mathcal{B}}(\mathfrak{D}) \le \mathbb{E}(\Phi^{\mathcal{B}}(\mathfrak{D})) + U\sqrt{\frac{\ln 4/\delta}{2t}}. \tag{A.1}$$

Then we give an upper bound for $\mathbb{E}(\Phi^{\mathcal{B}}(\mathfrak{D}))$, by the **symmetric technique**, let $\mathfrak{S} = \{\mathcal{S}_1, \cdots, \mathcal{S}_t\}$ be a set of independent copy of $\mathfrak{D}$, we have:

$$
\begin{aligned}
\mathbb{E}(\Phi^{\mathcal{B}}(\mathfrak{D})) &= \mathbb{E}_{\mathfrak{D} \sim p^t} \left[ \sup_{h \in \mathcal{H}} \left( \mathcal{L}_p^{\mathcal{B}}(\ell, h) - \mathcal{L}_{\hat{p}}^{\mathcal{B}}(\ell, h) \right) \right] \\
&= \mathbb{E}_{\mathfrak{D} \sim p^t} \left[ \sup_{h \in \mathcal{H}} \left( \mathbb{E}_{\mathcal{S} \sim p} \mathcal{R}_{\mathcal{S}}^{\mathcal{B}}(\ell, h) - \frac{1}{t} \sum_{i=1}^t \mathcal{R}_{\mathcal{D}_i}^{\mathcal{B}}(\ell, h) \right) \right] \\
&= \mathbb{E}_{\mathfrak{D} \sim p^t} \left[ \sup_{h \in \mathcal{H}} \frac{1}{t} \sum_{i=1}^t \mathbb{E}_{\mathcal{S}_i \sim p} \left( \mathcal{R}_{\mathcal{S}_i}^{\mathcal{B}}(\ell, h) - \mathcal{R}_{\mathcal{D}_i}^{\mathcal{B}}(\ell, h) \right) \right] \\
&\overset{a}{\le} \mathbb{E}_{\mathfrak{D} \sim p^t} \left[ \mathbb{E}_{\mathfrak{S} \sim p^t} \sup_{h \in \mathcal{H}} \frac{1}{t} \sum_{i=1}^t \left( \mathcal{R}_{\mathcal{S}_i}^{\mathcal{B}}(\ell, h) - \mathcal{R}_{\mathcal{D}_i}^{\mathcal{B}}(\ell, h) \right) \right] \\
&\overset{b}{=} \mathbb{E}_{\mathfrak{D} \sim p^t, \mathfrak{S} \sim p^t} \mathbb{E}_{\boldsymbol{\sigma}} \left[ \sup_{h \in \mathcal{H}} \frac{1}{t} \sum_{i=1}^t \sigma_i \left( \mathcal{R}_{\mathcal{S}_i}^{\mathcal{B}}(\ell, h) - \mathcal{R}_{\mathcal{D}_i}^{\mathcal{B}}(\ell, h) \right) \right] \\
&\overset{c}{\le} 2 \mathbb{E}_{\mathfrak{D} \sim p^t} \mathbb{E}_{\boldsymbol{\sigma}} \left[ \sup_{h \in \mathcal{H}} \frac{1}{t} \sum_{i=1}^t \sigma_i \mathcal{R}_{\mathcal{D}_i}^{\mathcal{B}}(\ell, h) \right] \\
&= 2 \mathbb{E}_{\mathfrak{D} \sim p^t} \mathbb{E}_{\boldsymbol{\sigma}} \left[ \sup_{h \in \mathcal{H}} \frac{1}{t} \sum_{i=1}^t \sigma_i \mathbb{E}_{(\boldsymbol{x}_i, y_i) \sim \mathcal{D}_i} \left[ \sup_{\boldsymbol{x}_i' \in \mathcal{B}(\boldsymbol{x}_i)} \ell(h(\boldsymbol{x}_i'), y_i) \right] \right] \\
&\overset{d}{\le} 2 \mathbb{E}_{\mathfrak{D} \sim p^t} \mathbb{E}_{(\boldsymbol{x}_i, y_i) \sim \mathcal{D}_i} \mathbb{E}_{\boldsymbol{\sigma}} \left[ \sup_{h \in \mathcal{H}} \frac{1}{t} \sum_{i=1}^t \sigma_i \left[ \sup_{\boldsymbol{x}_i' \in \mathcal{B}(\boldsymbol{x}_i)} \ell(h(\boldsymbol{x}_i'), y_i) \right] \right] \\
&= 2 \mathbb{E}_{\mathfrak{D} \sim p^t} \mathbb{E}_{(\boldsymbol{x}_i, y_i) \sim \mathcal{D}_i} \left[ \mathfrak{R}_t(\widetilde{\mathcal{G}}) \right],
\end{aligned}
$$

where: $a$ uses Jensen's inequality; $b$ comes from the fact that the random variables $\sigma_i \left( \mathcal{R}_{\mathcal{S}_i}^{\mathcal{B}}(\ell, h) - \mathcal{R}_{\mathcal{D}_i}^{\mathcal{B}}(\ell, h) \right)$ and $\mathcal{R}_{\mathcal{S}_i}^{\mathcal{B}}(\ell, h) - \mathcal{R}_{\mathcal{D}_i}^{\mathcal{B}}(\ell, h)$ are identically distributed, where $\sigma_i \sim$ Uniform($\{\pm 1\}$) and $\boldsymbol{\sigma} \in \{\pm 1\}^t$ has independent elements $\sigma_1, \cdots, \sigma_t$; $c$ is from the property of sup that $\sup(a + b) \le \sup(a) + \sup(b)$ and the fact that $\boldsymbol{\sigma}$ has the same distribution as $-\boldsymbol{\sigma}$; $d$ uses Jensen's inequality.

Since $\ell \in [0, U]$, change in one data point $(\boldsymbol{x}_i, y_i)$ will cause at most $\frac{U}{t}$ difference in $\mathfrak{R}_t(\widetilde{\mathcal{G}})$, we then use McDiarmid's inequality to get: with probability at least $1 - \frac{\delta}{4}$,

$$\mathbb{E}_{\mathfrak{D} \sim p^t} \mathbb{E}_{(\boldsymbol{x}_i, y_i) \sim \mathcal{D}_i} \left[ \mathfrak{R}_t(\widetilde{\mathcal{G}}) \right] \le \mathfrak{R}_t(\widetilde{\mathcal{G}}) + U\sqrt{\frac{\ln 4/\delta}{2t}}. \tag{A.2}$$

Combine (A.1) with (A.2) we get: with probability at least $1 - \frac{\delta}{2}$,

$$\Phi^{\mathcal{B}}(\mathfrak{D}) \le 2\mathfrak{R}_t(\widetilde{\mathcal{G}}) + 3U\sqrt{\frac{\ln 4/\delta}{2t}}. \tag{A.3}$$

**Step 2, upper bound the difference between $\mathcal{L}_{\hat{p}}^{\mathcal{B}}(\ell, h)$ and $\mathcal{L}_{\widehat{\mathcal{D}}}^{\mathcal{B}}(\ell, h)$ for any $h \in \mathcal{H}$.**

$$\mathcal{L}_{\hat{p}}^{\mathcal{B}}(\ell, h) - \mathcal{L}_{\widehat{\mathcal{D}}}^{\mathcal{B}}(\ell, h) = \frac{1}{t} \sum_{i=1}^{t} \left( \mathcal{R}_{\mathcal{D}_i}^{\mathcal{B}}(\ell, h) - \mathcal{R}_{\widehat{\mathcal{D}}_i}^{\mathcal{B}}(\ell, h) \right).$$

Let $\widehat{\mathfrak{D}} = \widehat{\mathcal{D}}_i \cup \cdots \cup \widehat{\mathcal{D}}_t$ and then define:

$$\Psi^{\mathcal{B}}(\widehat{\mathfrak{D}}) = \sup_{h \in \mathcal{H}} \left( \mathcal{L}_{\hat{p}}^{\mathcal{B}}(\ell, h) - \mathcal{L}_{\widehat{\mathcal{D}}}^{\mathcal{B}}(\ell, h) \right) = \sup_{h \in \mathcal{H}} \frac{1}{t} \sum_{i=1}^{t} \left( \mathcal{R}_{\mathcal{D}_i}^{\mathcal{B}}(\ell, h) - \mathcal{R}_{\widehat{\mathcal{D}}_i}^{\mathcal{B}}(\ell, h) \right).$$

Since $\ell \in [0, U]$ and there are $tn$ examples in $\widehat{\mathfrak{D}}$, changing the $j$-th example of $\widehat{\mathcal{D}}_i$, $(\boldsymbol{x}_{ij}, y_{ij})$, will cause at most $\frac{U}{tn}$ change $\Psi^{\mathcal{B}}(\widehat{\mathfrak{D}})$. So by McDiarmid's inequality we know: with probability at least $1 - \frac{\delta}{4}$ over the choice of $\widehat{\mathfrak{D}}$,

$$\Psi^{\mathcal{B}}(\widehat{\mathfrak{D}}) \leq \mathbb{E}\left[\Psi^{\mathcal{B}}(\widehat{\mathfrak{D}})\right] + U\sqrt{\frac{\ln 4/\delta}{2tn}}. \tag{A.4}$$

Next we upper bound $\mathbb{E}\left[\Psi^{\mathcal{B}}(\widehat{\mathfrak{D}})\right]$, by the symmetric technique, let $\widehat{\mathfrak{S}} = \{\widehat{\mathcal{S}}_1, \cdots, \widehat{\mathcal{S}}_t\}$ be a set of independent copies of $\widehat{\mathfrak{D}}$ and let $(\tilde{\boldsymbol{x}}_{ij}, \tilde{y}_{ij})$ be the $j$-th example in $\widehat{\mathcal{S}}_i$, we have:

$$\mathbb{E}\left[\Psi^{\mathcal{B}}(\widehat{\mathfrak{D}})\right] = \mathbb{E}_{\widehat{\mathfrak{D}}}\left[ \sup_{h \in \mathcal{H}} \frac{1}{t} \sum_{i=1}^{t} \left( \mathcal{R}_{\mathcal{D}_i}^{\mathcal{B}}(\ell, h) - \mathcal{R}_{\widehat{\mathcal{D}}_i}^{\mathcal{B}}(\ell, h) \right) \right]$$

$$= \mathbb{E}_{\widehat{\mathfrak{D}}}\left\{ \sup_{h \in \mathcal{H}} \frac{1}{t} \sum_{i=1}^{t} \left[ \mathbb{E}_{(\tilde{\boldsymbol{x}}_i, \tilde{y}_i) \sim \mathcal{D}_i} \left( \sup_{\tilde{\boldsymbol{x}}_i' \in \mathcal{B}(\tilde{\boldsymbol{x}}_i)} \ell(h(\tilde{\boldsymbol{x}}_i'), \tilde{y}_i) \right) - \frac{1}{n} \sum_{j=1}^{n} \sup_{\boldsymbol{x}_{ij}' \in \mathcal{B}(\boldsymbol{x}_{ij})} \ell(h(\boldsymbol{x}_{ij}'), y_{ij}) \right] \right\}$$

$$= \mathbb{E}_{\widehat{\mathfrak{D}}}\left\{ \sup_{h \in \mathcal{H}} \frac{1}{t} \sum_{i=1}^{t} \frac{1}{n} \sum_{j=1}^{n} \left[ \mathbb{E}_{(\tilde{\boldsymbol{x}}_{ij}, \tilde{y}_{ij}) \sim \widehat{\mathcal{S}}_i} \left( \sup_{\tilde{\boldsymbol{x}}_{ij}' \in \mathcal{B}(\tilde{\boldsymbol{x}}_{ij})} \ell(h(\tilde{\boldsymbol{x}}_{ij}'), \tilde{y}_{ij}) - \sup_{\boldsymbol{x}_{ij}' \in \mathcal{B}(\boldsymbol{x}_{ij})} \ell(h(\boldsymbol{x}_{ij}'), y_{ij}) \right) \right] \right\}$$

$$\overset{a}{\leq} \mathbb{E}_{\widehat{\mathfrak{D}}, \widehat{\mathfrak{S}}}\left\{ \sup_{h \in \mathcal{H}} \frac{1}{t} \sum_{i=1}^{t} \frac{1}{n} \sum_{j=1}^{n} \left[ \sup_{\tilde{\boldsymbol{x}}_{ij}' \in \mathcal{B}(\tilde{\boldsymbol{x}}_{ij})} \ell(h(\tilde{\boldsymbol{x}}_{ij}'), \tilde{y}_{ij}) - \sup_{\boldsymbol{x}_{ij}' \in \mathcal{B}(\boldsymbol{x}_{ij})} \ell(h(\boldsymbol{x}_{ij}'), y_{ij}) \right] \right\}$$

$$\overset{b}{\leq} \mathbb{E}_{\widehat{\mathfrak{D}}, \widehat{\mathfrak{S}}} \mathbb{E}_{\boldsymbol{\Sigma}}\left\{ \sup_{h \in \mathcal{H}} \frac{1}{t} \sum_{i=1}^{t} \frac{1}{n} \sum_{j=1}^{n} \left[ \Sigma_{ij} \left( \sup_{\tilde{\boldsymbol{x}}_{ij}' \in \mathcal{B}(\tilde{\boldsymbol{x}}_{ij})} \ell(h(\tilde{\boldsymbol{x}}_{ij}'), \tilde{y}_{ij}) - \sup_{\boldsymbol{x}_{ij}' \in \mathcal{B}(\boldsymbol{x}_{ij})} \ell(h(\boldsymbol{x}_{ij}'), y_{ij}) \right) \right] \right\}$$

$$\overset{c}{\leq} 2 \mathbb{E}_{\widehat{\mathfrak{D}}, \boldsymbol{\Sigma}}\left\{ \sup_{h \in \mathcal{H}} \frac{1}{t} \sum_{i=1}^{t} \frac{1}{n} \sum_{j=1}^{n} \left[ \Sigma_{ij} \sup_{\boldsymbol{x}_{ij}' \in \mathcal{B}(\boldsymbol{x}_{ij})} \ell(h(\boldsymbol{x}_{ij}'), y_{ij}) \right] \right\}$$

$$\overset{d}{=} 2 \mathbb{E}_{\widehat{\mathfrak{D}}}\left[ \mathfrak{R}_{tn}(\widetilde{\mathcal{G}}) \right],$$

where: $a$ is from Jensen's inequality; $b$ is from the fact that $\Sigma_{ij} \left( \sup_{\tilde{\boldsymbol{x}}_{ij}' \in \mathcal{B}(\tilde{\boldsymbol{x}}_{ij})} \ell(h(\tilde{\boldsymbol{x}}_{ij}'), \tilde{y}_{ij}) - \sup_{\boldsymbol{x}_{ij}' \in \mathcal{B}(\boldsymbol{x}_{ij})} \ell(h(\boldsymbol{x}_{ij}'), y_{ij}) \right)$ has the same distribution as $\sup_{\tilde{\boldsymbol{x}}_{ij}' \in \mathcal{B}(\tilde{\boldsymbol{x}}_{ij})} \ell(h(\tilde{\boldsymbol{x}}_{ij}'), \tilde{y}_{ij}) - \sup_{\boldsymbol{x}_{ij}' \in \mathcal{B}(\boldsymbol{x}_{ij})} \ell(h(\boldsymbol{x}_{ij}'), y_{ij})$, where $\Sigma_{ij} \sim \text{Uniform}(\{\pm 1\})$ and $\boldsymbol{\Sigma}$ has independent entries; $c$ is from the property of sup that $\sup(a + b) \leq \sup(a) + \sup(b)$ and the fact that $\boldsymbol{\Sigma}$ has the same distribution as $-\boldsymbol{\Sigma}$.

Similar as before, if we change one of the $\{(\boldsymbol{x}_{ij}, y_{ij})\}_{i,j=1}^{n}$, the change of $\mathfrak{R}_{tn}(\widetilde{\mathcal{G}})$ is at most $\frac{U}{tn}$, so by McDiarmid's inequality, with probability at least $1 - \frac{\delta}{4}$, we have:

$$\mathbb{E}_{\widehat{\mathfrak{D}}}\left[ \mathfrak{R}_{tn}(\widetilde{\mathcal{G}}) \right] \leq \mathfrak{R}_{tn}(\widetilde{\mathcal{G}}) + U\sqrt{\frac{\ln 4/\delta}{2tn}}. \tag{A.5}$$

Combine (A.4) and (A.5) we have: with probability at least $1 - \frac{\delta}{2}$,

$$\Psi^{\mathcal{B}}(\widehat{\mathfrak{D}}) \leq 2\mathfrak{R}_{tn}(\widetilde{\mathcal{G}}) + 3U\sqrt{\frac{\ln 4/\delta}{2tn}}. \tag{A.6}$$

Combine (A.3) and (A.6) we have: with probability at least $1 - \delta$:

$$\Phi^{\mathcal{B}}(\mathfrak{D}) + \Psi^{\mathcal{B}}(\widehat{\mathfrak{D}}) \leq 2\mathfrak{R}_t(\widetilde{\mathcal{G}}) + 2\mathfrak{R}_{tn}(\widetilde{\mathcal{G}}) + 3U\sqrt{\frac{\ln 4/\delta}{2t}} + 3U\sqrt{\frac{\ln 4/\delta}{2tn}}.$$

By the property of supremum, we have: with probability at least $1 - \delta$,

$$\begin{aligned}
\sup_{h \in \mathcal{H}} \left( \mathcal{L}_p^{\mathcal{B}}(\ell, h) - \mathcal{L}_{\widehat{\mathcal{D}}}^{\mathcal{B}}(\ell, h) \right) &= \sup_{h \in \mathcal{H}} \left( \mathcal{L}_p^{\mathcal{B}}(\ell, h) - \mathcal{L}_{\hat{p}}^{\mathcal{B}}(\ell, h) + \mathcal{L}_{\hat{p}}^{\mathcal{B}}(\ell, h) - \mathcal{L}_{\widehat{\mathcal{D}}}^{\mathcal{B}}(\ell, h) \right) \\
&\leq \sup_{h \in \mathcal{H}} \left( \mathcal{L}_p^{\mathcal{B}}(\ell, h) - \mathcal{L}_{\hat{p}}^{\mathcal{B}}(\ell, h) \right) + \sup_{h \in \mathcal{H}} \left( \mathcal{L}_{\hat{p}}^{\mathcal{B}}(\ell, h) - \mathcal{L}_{\widehat{\mathcal{D}}}^{\mathcal{B}}(\ell, h) \right) \\
&= \Phi^{\mathcal{B}}(\mathfrak{D}) + \Psi^{\mathcal{B}}(\widehat{\mathfrak{D}}) \\
&\leq 2\mathfrak{R}_t(\widetilde{\mathcal{G}}) + 2\mathfrak{R}_{tn}(\widetilde{\mathcal{G}}) + 3U\sqrt{\frac{\ln 4/\delta}{2t}} + 3U\sqrt{\frac{\ln 4/\delta}{2tn}}.
\end{aligned}$$

$\square$

*Proof of Corollary 3.2.* The main idea of the proof is to use risk decomposition to reduce $\mathcal{L}_p^{\mathcal{B}}(\ell, \hat{h}) - \mathcal{L}_p^{\mathcal{B}}(\ell, h^\star)$ to $\mathcal{L}_p^{\mathcal{B}}(\ell, h) - \mathcal{L}_{\widehat{\mathcal{D}}}^{\mathcal{B}}(\ell, h)$ for some $h$. Then we have: with probability at least $1 - \delta$,

$$\begin{aligned}
\mathcal{L}_p^{\mathcal{B}}(\ell, \hat{h}) - \mathcal{L}_p^{\mathcal{B}}(\ell, h^\star) &= \mathcal{L}_p^{\mathcal{B}}(\ell, \hat{h}) - \mathcal{L}_{\widehat{\mathcal{D}}}^{\mathcal{B}}(\ell, \hat{h}) + \mathcal{L}_{\widehat{\mathcal{D}}}^{\mathcal{B}}(\ell, \hat{h}) - \mathcal{L}_{\widehat{\mathcal{D}}}^{\mathcal{B}}(\ell, h^\star) + \mathcal{L}_{\widehat{\mathcal{D}}}^{\mathcal{B}}(\ell, h^\star) - \mathcal{L}_p^{\mathcal{B}}(\ell, h^\star) \\
&\overset{a}{\leq} \mathcal{L}_p^{\mathcal{B}}(\ell, \hat{h}) - \mathcal{L}_{\widehat{\mathcal{D}}}^{\mathcal{B}}(\ell, \hat{h}) + \mathcal{L}_{\widehat{\mathcal{D}}}^{\mathcal{B}}(\ell, h^\star) - \mathcal{L}_p^{\mathcal{B}}(\ell, h^\star) \\
&\leq 2\sup_{h \in \mathcal{H}} \left| \mathcal{L}_p^{\mathcal{B}}(\ell, h) - \mathcal{L}_{\widehat{\mathcal{D}}}^{\mathcal{B}}(\ell, h) \right| \\
&\overset{b}{\leq} 4\mathfrak{R}_t(\widetilde{\mathcal{G}}) + 4\mathfrak{R}_{tn}(\widetilde{\mathcal{G}}) + 3U\sqrt{\frac{\ln 8/\delta}{2t}} + 3U\sqrt{\frac{\ln 8/\delta}{2tn}},
\end{aligned}$$

where: $a$ is from the fact that $\mathcal{L}_{\widehat{\mathcal{D}}}^{\mathcal{B}}(\ell, \hat{h}) \leq \mathcal{L}_{\widehat{\mathcal{D}}}^{\mathcal{B}}(\ell, h^\star)$ by the definition of $\hat{h}$; $b$ is from Theorem 3.1, although in Theorem 3.1, we only have bounds for $\sup_{h \in \mathcal{H}} \left( \mathcal{L}_p^{\mathcal{B}}(\ell, h) - \mathcal{L}_{\widehat{\mathcal{D}}}^{\mathcal{B}}(\ell, h) \right)$, we can use a similar argumentation as in Theorem 3.1 to get the same bound for $\sup_{h \in \mathcal{H}} \left( \mathcal{L}_{\widehat{\mathcal{D}}}^{\mathcal{B}}(\ell, h) - \mathcal{L}_p^{\mathcal{B}}(\ell, h) \right)$, both with probability at least $1 - \delta'$, using union bound for these two cases we get a bound for $\sup_{h \in \mathcal{H}} \left| \mathcal{L}_p^{\mathcal{B}}(\ell, h) - \mathcal{L}_{\widehat{\mathcal{D}}}^{\mathcal{B}}(\ell, h) \right|$ with probability at least $1 - 2\delta'$, take $\delta = \frac{\delta'}{2}$, we get the bound for $\sup_{h \in \mathcal{H}} \left| \mathcal{L}_p^{\mathcal{B}}(\ell, h) - \mathcal{L}_{\widehat{\mathcal{D}}}^{\mathcal{B}}(\ell, h) \right|$ with probability at least $1 - \delta$, this is why we get $\ln 4/\delta$ in Theorem 3.1 but $\ln 8/\delta$ here. $\square$

## A.3 Proof of the Limited Environment Case

*Proof $d_{\ell(\mathcal{H})}^{\mathcal{B}}(\cdot, \cdot)$ is a pseudometric.* It is obvious that $d_{\ell(\mathcal{H})}^{\mathcal{B}}(\cdot, \cdot)$ is symmetric, now we proof that $d_{\ell(\mathcal{H})}^{\mathcal{B}}(\cdot, \cdot)$ satisfies the triangle inequality. By definition, for distribution $P, Q, R$:

$$
\begin{aligned}
d_{\ell(\mathcal{H})}^{\mathcal{B}}(P, Q) &= \sup_{h \in \mathcal{H}} \left| \mathbb{E}_{(\boldsymbol{x}, y) \sim P} \left[ \sup_{\boldsymbol{x}' \in \mathcal{B}(\boldsymbol{x})} \ell(h(\boldsymbol{x}'), y) \right] - \mathbb{E}_{(\boldsymbol{x}, y) \sim Q} \left[ \sup_{\boldsymbol{x}' \in \mathcal{B}(\boldsymbol{x})} \ell(h(\boldsymbol{x}'), y) \right] \right| \\
&= \sup_{h \in \mathcal{H}} \left| \mathbb{E}_{(\boldsymbol{x}, y) \sim P} \left[ \sup_{\boldsymbol{x}' \in \mathcal{B}(\boldsymbol{x})} \ell(h(\boldsymbol{x}'), y) \right] - \mathbb{E}_{(\boldsymbol{x}, y) \sim R} \left[ \sup_{\boldsymbol{x}' \in \mathcal{B}(\boldsymbol{x})} \ell(h(\boldsymbol{x}'), y) \right] \right. \\
&\quad + \left. \mathbb{E}_{(\boldsymbol{x}, y) \sim R} \left[ \sup_{\boldsymbol{x}' \in \mathcal{B}(\boldsymbol{x})} \ell(h(\boldsymbol{x}'), y) \right] - \mathbb{E}_{(\boldsymbol{x}, y) \sim Q} \left[ \sup_{\boldsymbol{x}' \in \mathcal{B}(\boldsymbol{x})} \ell(h(\boldsymbol{x}'), y) \right] \right| \\
&\overset{a}{\leq} \sup_{h \in \mathcal{H}} \left| \mathbb{E}_{(\boldsymbol{x}, y) \sim P} \left[ \sup_{\boldsymbol{x}' \in \mathcal{B}(\boldsymbol{x})} \ell(h(\boldsymbol{x}'), y) \right] - \mathbb{E}_{(\boldsymbol{x}, y) \sim R} \left[ \sup_{\boldsymbol{x}' \in \mathcal{B}(\boldsymbol{x})} \ell(h(\boldsymbol{x}'), y) \right] \right| \\
&\quad + \sup_{h \in \mathcal{H}} \left| \mathbb{E}_{(\boldsymbol{x}, y) \sim R} \left[ \sup_{\boldsymbol{x}' \in \mathcal{B}(\boldsymbol{x})} \ell(h(\boldsymbol{x}'), y) \right] - \mathbb{E}_{(\boldsymbol{x}, y) \sim Q} \left[ \sup_{\boldsymbol{x}' \in \mathcal{B}(\boldsymbol{x})} \ell(h(\boldsymbol{x}'), y) \right] \right| \\
&= d_{\ell(\mathcal{H})}^{\mathcal{B}}(P, R) + d_{\ell(\mathcal{H})}^{\mathcal{B}}(R, Q),
\end{aligned}
$$

where: $a$ is from the subadditivity of the supremum operator. So $d_{\ell(\mathcal{H})}^{\mathcal{B}}(\cdot, \cdot)$ is a pseudometric. $\square$

Before prove Thorem 3.3, we first give a useful lemma.

**Lemma A.5.** *For any $h \in \mathcal{H}$, any distribution $P, Q$ on , we have:*

$$
\left| \mathcal{R}_P^{\mathcal{B}}(\ell, h) - \mathcal{R}_Q^{\mathcal{B}}(\ell, h) \right| \leq d_{\ell(\mathcal{H})}^{\mathcal{B}}(P, Q)
$$

*Proof of Lemma A.5.* From the definition of $d_{\ell(\mathcal{H})}^{\mathcal{B}}(P, Q)$, we can know that:

$$
d_{\ell(\mathcal{H})}^{\mathcal{B}}(P, Q) = \sup_{h \in \mathcal{H}} \left| \mathcal{R}_P^{\mathcal{B}}(\ell, h) - \mathcal{R}_Q^{\mathcal{B}}(\ell, h) \right| \geq \left| \mathcal{R}_P^{\mathcal{B}}(\ell, h) - \mathcal{R}_Q^{\mathcal{B}}(\ell, h) \right|, \ \forall h \in \mathcal{H},
$$

which is just what we want. $\square$

**Theorem 3.3.** *For a given but unknown target distribution $\mathcal{T}$, let $\Delta^{t-1} := \{(\lambda_1, \cdots, \lambda_t) | \lambda_i \geq 0, \sum_{i=1}^t \lambda_i = 1\}$ be the $t$-dimensional simplex and $Conv(\mathfrak{D}) := \left\{ \sum_{i=1}^t \lambda_i \mathcal{D}_i \big| \boldsymbol{\lambda} \in \Delta^{t-1} \right\}$ be the convex hull of $\mathfrak{D} = \{\mathcal{D}_1, \ldots, \mathcal{D}_t\}$, define $\mathcal{T}_P \in \underset{\mathcal{D} \in Conv(\mathfrak{D})}{\arg\inf} d_{\ell(\mathcal{H})}^{\mathcal{B}}(\mathcal{D}, \mathcal{T})$ be the "projection" of $\mathcal{T}$ onto $Conv(\mathfrak{D})$ and $\boldsymbol{\lambda}^\star$ be the weight vector where $\mathcal{T}_P = \sum_{i=1}^t \lambda_i^\star \mathcal{D}_i$, $\ell \in [0, U]$, then we have: with probability at least $1 - \delta$, for all $h \in \mathcal{H}$,*

$$
\mathcal{R}_{\mathcal{T}}^{\mathcal{B}}(\ell, h) \leq \frac{1}{t} \sum_{i=1}^t \mathcal{R}_{\widehat{\mathcal{D}}_i}^{\mathcal{B}}(\ell, h) + \frac{1}{t} \sum_i \sum_j \lambda_j^\star d_{\ell(\mathcal{H})}^{\mathcal{B}}(\widehat{\mathcal{D}}_i, \widehat{\mathcal{D}}_j) + d_{\ell(\mathcal{H})}^{\mathcal{B}}(\mathcal{T}, \mathcal{T}_P)
$$

$$
+ 4 \mathfrak{R}_{tn}(\widetilde{\mathcal{G}}) + 2 \mathfrak{R}_n(\widetilde{\mathcal{G}}) + 6U \sqrt{\frac{ln 8/\delta}{2tn}} + 3U \sqrt{\frac{ln(16t/\delta)}{2n}}
$$

*Proof of Theorem 3.3.* Our proof is divided into 2 steps, the first step gets results for the population distributions and the second step uses finite sample approximation to get results for empirical distributions.

**Step 1. Get the relationship between the population distributions**.

Lemma A.5 tells us that: for all $h \in \mathcal{H}$,

$$
\mathcal{R}_{\mathcal{T}}^{\mathcal{B}}(\ell, h) \leq \mathcal{R}_{\bar{\mathcal{D}}}^{\mathcal{B}}(\ell, h) + d_{\ell(\mathcal{H})}^{\mathcal{B}}(\mathcal{T}, \bar{\mathcal{D}}) \leq \mathcal{R}_{\bar{\mathcal{D}}}^{\mathcal{B}}(\ell, h) + d_{\ell(\mathcal{H})}^{\mathcal{B}}(\mathcal{T}, \mathcal{T}_P) + d_{\ell(\mathcal{H})}^{\mathcal{B}}(\mathcal{T}_P, \bar{\mathcal{D}}),
$$

where the last inequality is from the triangle inequality of $d_{\ell(\mathcal{H})}^{\mathcal{B}}(\cdot, \cdot)$ and $\bar{\mathcal{D}} = \frac{1}{t} \sum_{i=1}^{t} \mathcal{D}_i$. Recall that $\widehat{\mathcal{D}} = \frac{1}{t} \sum_{i=1}^{t} \widehat{\mathcal{D}}_i$, then we have:

$$
d_{\ell(\mathcal{H})}^{\mathcal{B}}(\mathcal{T}_P, \bar{\mathcal{D}}) \overset{a}{\leq} d_{\ell(\mathcal{H})}^{\mathcal{B}}(\mathcal{T}_P, \widehat{\mathcal{T}}_P) + d_{\ell(\mathcal{H})}^{\mathcal{B}}(\widehat{\mathcal{T}}_P, \widehat{\mathcal{D}}) + d_{\ell(\mathcal{H})}^{\mathcal{B}}(\widehat{\mathcal{D}}, \bar{\mathcal{D}})
$$

$$
= d_{\ell(\mathcal{H})}^{\mathcal{B}}(\mathcal{T}_P, \widehat{\mathcal{T}}_P) + d_{\ell(\mathcal{H})}^{\mathcal{B}}(\widehat{\mathcal{D}}, \bar{\mathcal{D}}) + d_{\ell(\mathcal{H})}^{\mathcal{B}}\left( \sum_{i=1}^{t} \lambda_i^\star \widehat{\mathcal{D}}_i, \frac{1}{t} \sum_{i=1}^{t} \widehat{\mathcal{D}}_i \right)
$$

$$
= d_{\ell(\mathcal{H})}^{\mathcal{B}}(\mathcal{T}_P, \widehat{\mathcal{T}}_P) + d_{\ell(\mathcal{H})}^{\mathcal{B}}(\widehat{\mathcal{D}}, \bar{\mathcal{D}}) + \sup_{h \in \mathcal{H}} \left| \mathcal{R}_{\sum_i \lambda_i^\star \widehat{\mathcal{D}}_i}^{\mathcal{B}}(\ell, h) - \mathcal{R}_{\sum_i \frac{1}{t} \widehat{\mathcal{D}}_i}^{\mathcal{B}}(\ell, h) \right|
$$

$$
= d_{\ell(\mathcal{H})}^{\mathcal{B}}(\mathcal{T}_P, \widehat{\mathcal{T}}_P) + d_{\ell(\mathcal{H})}^{\mathcal{B}}(\widehat{\mathcal{D}}, \bar{\mathcal{D}}) + \sup_{h \in \mathcal{H}} \left| \mathcal{R}_{\frac{1}{t} \sum_i \sum_j \lambda_j^\star \widehat{\mathcal{D}}_j}^{\mathcal{B}}(\ell, h) - \mathcal{R}_{\sum_i \frac{1}{t} \sum_j \lambda_j^\star \widehat{\mathcal{D}}_i}^{\mathcal{B}}(\ell, h) \right|
$$

$$
\overset{b}{\leq} d_{\ell(\mathcal{H})}^{\mathcal{B}}(\mathcal{T}_P, \widehat{\mathcal{T}}_P) + d_{\ell(\mathcal{H})}^{\mathcal{B}}(\widehat{\mathcal{D}}, \bar{\mathcal{D}}) + \frac{1}{t} \sum_i \sum_j \lambda_j^\star \sup_{h \in \mathcal{H}} \left| \mathcal{R}_{\widehat{\mathcal{D}}_i}^{\mathcal{B}}(\ell, h) - \mathcal{R}_{\widehat{\mathcal{D}}_j}^{\mathcal{B}}(\ell, h) \right|
$$

$$
= d_{\ell(\mathcal{H})}^{\mathcal{B}}(\mathcal{T}_P, \widehat{\mathcal{T}}_P) + d_{\ell(\mathcal{H})}^{\mathcal{B}}(\widehat{\mathcal{D}}, \bar{\mathcal{D}}) + \frac{1}{t} \sum_i \sum_j \lambda_j^\star d_{\ell(\mathcal{H})}^{\mathcal{B}}(\widehat{\mathcal{D}}_i, \widehat{\mathcal{D}}_j),
$$

where: $a$ is from the triangle inequality of $d_{\ell(\mathcal{H})}^{\mathcal{B}}(\cdot, \cdot)$; $b$ follows from the linearity of $\mathcal{R}_{\mathcal{D}}^{\mathcal{B}}(\ell, h)$ and the subadditivity of the supremum operator. So we have, for any $h \in \mathcal{H}$:

$$
\mathcal{R}_{\mathcal{T}}^{\mathcal{B}}(\ell, h) \leq \mathcal{R}_{\mathcal{D}}^{\mathcal{B}}(\ell, h) + d_{\ell(\mathcal{H})}^{\mathcal{B}}(\mathcal{T}, \mathcal{T}_P) + d_{\ell(\mathcal{H})}^{\mathcal{B}}(\mathcal{T}_P, \widehat{\mathcal{T}}_P) + d_{\ell(\mathcal{H})}^{\mathcal{B}}(\widehat{\mathcal{D}}, \bar{\mathcal{D}}) + \frac{1}{t} \sum_i \sum_j \lambda_j^\star d_{\ell(\mathcal{H})}^{\mathcal{B}}(\widehat{\mathcal{D}}_i, \widehat{\mathcal{D}}_j).
$$

$$(A.7)$$

**Step 2. We now show the bound of the finite sample approximation error $d_{\ell(\mathcal{H})}^{\mathcal{B}}(\mathcal{T}_P, \widehat{\mathcal{T}}_P)$ and $d_{\ell(\mathcal{H})}^{\mathcal{B}}(\widehat{\mathcal{D}}, \bar{\mathcal{D}})$. Since we have no access to the population distribution $\bar{\mathcal{D}}$, we also give a finite sample approximation of $\mathcal{R}_{\bar{\mathcal{D}}}^{\mathcal{B}}(\ell, h)$ and bound the corresponding approximation error.**

The empirical distribution of $\bar{\mathcal{D}}$ is $\widehat{\mathcal{D}} = \frac{1}{t} \sum_{i=1}^{t} \widehat{\mathcal{D}}_i$, then we have:

$$
\sup_{h \in \mathcal{H}} \left( \mathcal{R}_{\bar{\mathcal{D}}}^{\mathcal{B}}(\ell, h) - \mathcal{R}_{\widehat{\mathcal{D}}}^{\mathcal{B}}(\ell, h) \right) = \sup_{h \in \mathcal{H}} \frac{1}{t} \sum_{i=1}^{t} \left( \mathcal{R}_{\mathcal{D}_i}^{\mathcal{B}}(\ell, h) - \mathcal{R}_{\widehat{\mathcal{D}}_i}^{\mathcal{B}}(\ell, h) \right) = \Psi^{\mathcal{B}}(\widehat{\mathfrak{D}}),
$$

where $\Psi^{\mathcal{B}}(\widehat{\mathfrak{D}})$ is defined in the proof of Theorem 3.1. Then we have, with probability at least $1 - \frac{\delta}{4}$, for all $h \in \mathcal{H}$,

$$
\mathcal{R}_{\bar{\mathcal{D}}}^{\mathcal{B}}(\ell, h) = \frac{1}{t} \sum_{i=1}^{t} \mathcal{R}_{\mathcal{D}_i}^{\mathcal{B}}(\ell, h) \leq \frac{1}{t} \sum_{i=1}^{t} \mathcal{R}_{\widehat{\mathcal{D}}_i}^{\mathcal{B}}(\ell, h) + 2\mathfrak{R}_{tn}(\widetilde{\mathcal{G}}) + 3U \sqrt{\frac{\ln 8/\delta}{2tn}}. \qquad (A.8)
$$

Note that $d_{\ell(\mathcal{H})}^{\mathcal{B}}(\widehat{\mathcal{D}}, \bar{\mathcal{D}}) = \sup_{h \in \mathcal{H}} \left| \mathcal{R}_{\bar{\mathcal{D}}}^{\mathcal{B}}(\ell, h) - \mathcal{R}_{\widehat{\mathcal{D}}}^{\mathcal{B}}(\ell, h) \right|$, similar as the bound of $\Psi^{\mathcal{B}}(\widehat{\mathfrak{D}}) = \sup_{h \in \mathcal{H}} \left( \mathcal{R}_{\bar{\mathcal{D}}}^{\mathcal{B}}(\ell, h) - \mathcal{R}_{\widehat{\mathcal{D}}}^{\mathcal{B}}(\ell, h) \right)$, we have that with probability at least $1 - \frac{\delta}{4}$:

$$
\sup_{h \in \mathcal{H}} \left( \mathcal{R}_{\widehat{\mathcal{D}}}^{\mathcal{B}}(\ell, h) - \mathcal{R}_{\bar{\mathcal{D}}}^{\mathcal{B}}(\ell, h) \right) \leq 2\mathfrak{R}_{tn}(\widetilde{\mathcal{G}}) + 3U \sqrt{\frac{\ln 8/\delta}{2tn}},
$$

so with probability at least $1 - \frac{\delta}{2}$:

$$
d_{\ell(\mathcal{H})}^{\mathcal{B}}(\widehat{\mathcal{D}}, \bar{\mathcal{D}}) = \sup_{h \in \mathcal{H}} \left| \mathcal{R}_{\bar{\mathcal{D}}}^{\mathcal{B}}(\ell, h) - \mathcal{R}_{\widehat{\mathcal{D}}}^{\mathcal{B}}(\ell, h) \right| \leq 2\mathfrak{R}_{tn}(\widetilde{\mathcal{G}}) + 3U \sqrt{\frac{\ln 8/\delta}{2tn}}. \qquad (A.9)
$$

Now we bound $d_{\ell(\mathcal{H})}^{\mathcal{B}}(\mathcal{T}_P, \widehat{\mathcal{T}}_P)$, by the definition of $\mathcal{T}_P$, we have:

$$
d_{\ell(\mathcal{H})}^{\mathcal{B}}(\mathcal{T}_P, \widehat{\mathcal{T}}_P) = d_{\ell(\mathcal{H})}^{\mathcal{B}}\left( \sum_{i=1}^{t} \lambda_i^\star \mathcal{D}_i, \sum_{i=1}^{t} \lambda_i^\star \widehat{\mathcal{D}}_i \right) \leq \sum_{i=1}^{t} \lambda_i^\star d_{\ell(\mathcal{H})}^{\mathcal{B}}(\mathcal{D}_i, \widehat{\mathcal{D}}_i)
$$

For each $i$, we define:

$$\Gamma^{\mathcal{B}}(\widehat{\mathcal{D}}_i) = \sup_{h \in \mathcal{H}} \left( \mathcal{R}^{\mathcal{B}}_{\mathcal{D}_i}(\ell, h) - \mathcal{R}^{\mathcal{B}}_{\widehat{\mathcal{D}}_i}(\ell, h) \right).$$

Since $\ell \in [0, U]$, changing one example in $\widehat{\mathcal{D}}_i$ will lead to at most $\frac{1}{n}$ chang in $\Gamma^{\mathcal{B}}(\widehat{\mathcal{D}}_i)$, so by McDiarmid's inequality we know: with probability at least $1 - \frac{\delta}{16t}$,

$$\Gamma^{\mathcal{B}}(\widehat{\mathcal{D}}_i) \leq \mathbb{E}\left[ \Gamma^{\mathcal{B}}(\widehat{\mathcal{D}}_i) \right] + U \sqrt{\frac{\ln(16t/\delta)}{2n}}.$$

Next we upper bound $\mathbb{E}\left[ \Gamma^{\mathcal{B}}(\widehat{\mathcal{D}}_i) \right]$, by similar analysis as in the proof of Theorem 3.1, we have: with probability at least $1 - \frac{\delta}{16t}$,

$$\mathbb{E}_{\widehat{\mathcal{D}}_i}\left[ \Gamma^{\mathcal{B}}(\widehat{\mathcal{D}}_i) \right] \leq \mathfrak{R}_n(\widetilde{\mathcal{G}}) + U \sqrt{\frac{\ln(16t/\delta)}{2n}}.$$

Then with probability at least $1 - \frac{\delta}{8t}$,

$$\sup_{h \in \mathcal{H}} \left( \mathcal{R}^{\mathcal{B}}_{\mathcal{D}_i}(\ell, h) - \mathcal{R}^{\mathcal{B}}_{\widehat{\mathcal{D}}_i}(\ell, h) \right) = \Gamma^{\mathcal{B}}(\widehat{\mathcal{D}}_i) \leq 2\mathfrak{R}_n(\widetilde{\mathcal{G}}) + 3U \sqrt{\frac{\ln(16t/\delta)}{2n}}.$$

With similar argument, we know that with probability at least $1 - \frac{\delta}{8t}$,

$$\sup_{h \in \mathcal{H}} \left( \mathcal{R}^{\mathcal{B}}_{\widehat{\mathcal{D}}_i}(\ell, h) - \mathcal{R}^{\mathcal{B}}_{\mathcal{D}_i}(\ell, h) \right) \leq 2\mathfrak{R}_n(\widetilde{\mathcal{G}}) + 3U \sqrt{\frac{\ln(16t/\delta)}{2n}}.$$

So with probability at least $1 - \frac{\delta}{4t}$,

$$d^{\mathcal{B}}_{\ell(\mathcal{H})}(\widehat{\mathcal{D}}_j, \mathcal{D}_j) = \sup_{h \in \mathcal{H}} \left| \mathcal{R}^{\mathcal{B}}_{\mathcal{D}_i}(\ell, h) - \mathcal{R}^{\mathcal{B}}_{\widehat{\mathcal{D}}_i}(\ell, h) \right| \leq 2\mathfrak{R}_n(\widetilde{\mathcal{G}}) + 3U \sqrt{\frac{\ln(16t/\delta)}{2n}}.$$

So with probability at least $1 - \frac{\delta}{4}$,

$$d^{\mathcal{B}}_{\ell(\mathcal{H})}(\mathcal{T}_P, \widehat{\mathcal{T}}_P) \leq \sum_{j=1}^{t} \lambda^\star_j d^{\mathcal{B}}_{\ell(\mathcal{H})}(\widehat{\mathcal{D}}_j, \mathcal{D}_j) \leq 2\mathfrak{R}_n(\widetilde{\mathcal{G}}) + 3U \sqrt{\frac{\ln(16t/\delta)}{2n}}. \tag{A.10}$$

Now combine (A.7), (A.8), (A.9) and (A.10), we have: with probability at least $1 - \delta$, for any $h \in \mathcal{H}$,

$$\mathcal{R}^{\mathcal{B}}_{\mathcal{T}}(\ell, h) \leq \mathcal{R}^{\mathcal{B}}_{\mathcal{D}}(\ell, h) + d^{\mathcal{B}}_{\ell(\mathcal{H})}(\mathcal{T}, \mathcal{T}_P) + d^{\mathcal{B}}_{\ell(\mathcal{H})}(\mathcal{T}_P, \widehat{\mathcal{T}}_P) + d^{\mathcal{B}}_{\ell(\mathcal{H})}(\widehat{\mathcal{D}}, \bar{\mathcal{D}}) + \frac{1}{t} \sum_i \sum_j \lambda^\star_j d^{\mathcal{B}}_{\ell(\mathcal{H})}(\widehat{\mathcal{D}}_i, \widehat{\mathcal{D}}_j)$$

$$\leq \frac{1}{t} \sum_{i=1}^{t} \mathcal{R}^{\mathcal{B}}_{\widehat{\mathcal{D}}_i}(\ell, h) + 2\mathfrak{R}_{tn}(\widetilde{\mathcal{G}}) + 3U \sqrt{\frac{\ln 8/\delta}{2tn}} + d^{\mathcal{B}}_{\ell(\mathcal{H})}(\mathcal{T}, \mathcal{T}_P) + 2\mathfrak{R}_n(\widetilde{\mathcal{G}})$$

$$+ 3U \sqrt{\frac{\ln(16t/\delta)}{2n}} + 2\mathfrak{R}_{tn}(\widetilde{\mathcal{G}}) + 3U \sqrt{\frac{\ln 8/\delta}{2tn}} + \frac{1}{t} \sum_i \sum_j \lambda^\star_j d^{\mathcal{B}}_{\ell(\mathcal{H})}(\widehat{\mathcal{D}}_i, \widehat{\mathcal{D}}_j)$$

$$= \frac{1}{t} \sum_{i=1}^{t} \mathcal{R}^{\mathcal{B}}_{\widehat{\mathcal{D}}_i}(\ell, h) + \frac{1}{t} \sum_i \sum_j \lambda^\star_j d^{\mathcal{B}}_{\ell(\mathcal{H})}(\widehat{\mathcal{D}}_i, \widehat{\mathcal{D}}_j) + d^{\mathcal{B}}_{\ell(\mathcal{H})}(\mathcal{T}, \mathcal{T}_P) + 4\mathfrak{R}_{tn}(\widetilde{\mathcal{G}})$$

$$+ 2\mathfrak{R}_n(\widetilde{\mathcal{G}}) + 6U \sqrt{\frac{\ln 8/\delta}{2tn}} + 3U \sqrt{\frac{\ln(16t/\delta)}{2n}}.$$

$\square$

# B   Toy Example to Model the ColoredMNIST

**Example 2.** Consider data point $(\boldsymbol{x}, y)$, where $\boldsymbol{x} \in \mathcal{X} \subseteq \mathbb{R}^{d+1}$ consists of invariant feature $\boldsymbol{x}_{\text{inv}} \in \mathcal{X}_{\text{inv}} \subseteq \mathbb{R}^d$ and spuriously correlated feature $x_{\text{sp}} \in \{\pm s\} \subsetneq \mathbb{R}$, to simplify the example, let $\boldsymbol{x} =$

$(\boldsymbol{x}_{\text{inv}}, x_{\text{sp}})$ be the feature observed. Consider the case there is a linear classifier $h(\boldsymbol{x}_{\text{inv}}) = \boldsymbol{w}_{\text{inv}} \cdot \boldsymbol{x}_{\text{inv}}$ such that $y \cdot h(\boldsymbol{x}_{\text{inv}}) > 0$, i.e., $\boldsymbol{x}_{\text{inv}}$ is invariant for all distributions on $\mathcal{X}_{\text{inv}} \times \{\pm 1\}$. Suppose the dataset $S = \{(\boldsymbol{x}_i, y_i)\}_{i=1}^{n}$ is linearly separable, and the dataset $S$ is induced into two disjoint groups: a **majority group** $S_{\text{maj}}$ where $x_{\text{sp}} \cdot y > 0$ and a **minority group** $S_{\text{min}}$ where $x_{\text{sp}} \cdot y < 0$, let $p = \mathbb{P}[x_{\text{sp}} \cdot y > 0] > 0.5$, which means that the spurious feature is positively related to the label $y$.

The next theorem considers the degree of dependence on the spurious feature of the trained linear classifier.

**Theorem B.1** (Theorem 4 of [48]). *Let $\mathbb{H}$ be the set fo linear classifiers, $h(\boldsymbol{x}) = \boldsymbol{w}_{inv}\boldsymbol{x}_{inv} + w_{sp}x_{sp}$. Consider any task that satisfies all the constraints in Section 3.1 of [48]. Consider a dataset $S$ drawn from $\mathcal{D}$ such that the empirical distribution of $\boldsymbol{x}_{inv}$ given $x_{sp} \cdot y > 0$ is identical to the empirical distribution of $\boldsymbol{x}_{inv}$ given $x_{sp} \cdot y < 0$. Let $\boldsymbol{w}_{inv}(t)\boldsymbol{x}_{inv} + w_{sp}x_{sp}$ be initialized to the origin, and trained with an infinitesimal learning rate gradient descent to minimize the exponential loss on a dataset $S$. Then, for any $(\boldsymbol{x}, y) \in S$, we have:*

$$\Omega \left( \frac{ln\frac{c+p}{c+\sqrt{p(1-p)}}}{\mathcal{M}ln(t+1)} \right) \leq \frac{w_{sp}(t)s}{|\boldsymbol{w}_{inv}(t) \cdot \boldsymbol{x}_{inv}|} \leq \mathcal{O} \left( \frac{ln\frac{p}{1-p}}{ln(t+1)} \right),$$

*where:*

1. *$p$ denotes the empirical level of spurious correlation, $p = \frac{1}{|S|} \sum_{(\boldsymbol{x},y) \in S} \mathbf{1}[x_{sp} \cdot y > 0]$ which without generality is assumed to satisfy $p \in [0.5, 1)$.*

2. *$\mathcal{M}$ denotes the maximum value of the margin of the max-margin classifier on $S$, i.e., $\mathcal{M} = max_{\boldsymbol{x} \in S}\hat{\boldsymbol{w}} \cdot \boldsymbol{x}$ where $\hat{\boldsymbol{w}}$ is the max-margin classifier on $S$.*

3. *$c := \frac{2(2\mathcal{M}-1)}{s^2}$.*

**Remark 4.** *In the proof of Theorem B.1 (please refer to [48]), the loss function is:*

$$L(\boldsymbol{w}_{inv}, w_{sp}) = p\mathbb{E}_{\boldsymbol{x}_{inv} \sim \hat{D}_{inv}} \left[ e^{-(\boldsymbol{w}_{inv}\boldsymbol{x}_{inv} + w_{sp}s)} \right] + (1-p)\mathbb{E}_{\boldsymbol{x}_{inv} \sim \hat{D}_{inv}} \left[ e^{-(\boldsymbol{w}_{inv}\boldsymbol{x}_{inv} - w_{sp}s)} \right]. \tag{B.11}$$

*We denote $L_1 = \mathbb{E}_{\boldsymbol{x}_{inv} \sim \hat{D}_{inv}} \left[ e^{-(\boldsymbol{w}_{inv}\boldsymbol{x}_{inv} + w_{sp}s)} \right]$ and $L_2 = \mathbb{E}_{\boldsymbol{x}_{inv} \sim \hat{D}_{inv}} \left[ e^{-(\boldsymbol{w}_{inv}\boldsymbol{x}_{inv} - w_{sp}s)} \right]$, so $L = pL_1 + (1-p)L_2$. To model the case in the ColoredMNIST dataset, we let $S_1$ denote the dataset with corruption rate 0.1, $S_2$ the dataset with corruption rate 0.2, $S_3$ the dataset with corruption rate 0.9. For training environment with corruption rate $p_{train}$, we have $L_{train} = p_{train}L_1 + (1 - p_{train})L_2$ and $L_{test} = p_{test}L_1 + (1 - p_{test})L_2$ if $p_{train} > 0.5$. otherwise we have $L_{train} = p_{train}L_2 + (1 - p_{train})L_1$ and $L_{test} = p_{test}L_2 + (1 - p_{test})L_1$. What's more, since the lower bound $> 0$ for $p > \frac{1}{2}$, we have $L_1 < L_2$. So in the standard training case:*

1. *When we train with $S_1, S_3$, then the overall $p_{train} = \frac{0.1+0.9}{2} = \frac{1}{2}$. So according to Theorem B.1 we know that $w_{sp}(t) = 0$, which means that the classifier just depends on the invariant feature, so the performance on $S_2$ is good. Since $w_{sp}(t) = 0$, $L_1 = L_2$, so we have $L_{train} = p_{train}L_1 + (1 - p_{train})L_2 = \mathbb{E}_{\boldsymbol{x}_{inv} \sim \hat{D}_{inv}} \left[ e^{-(\boldsymbol{w}_{inv}\boldsymbol{x}_{inv})} \right] = L_{test}$. So the mixed training distribution here is the same as the test distribution, which means that we can generalize well in this case, which is consistent with our experimental results.*

2. *When training with $S_2, S_3$, then the overall $p_{train} = \frac{0.2+0.9}{2} = 0.55$, which is quite close to 0.5, which means that the correlation between $x_{sp}$ and $y$ is not very strong. It means that the upper bound and the lower bound in Theorem B.1 is small, so $w_{sp}$ is small, the classifier mainly depends on the invariant feature and slightly depends on the spurious feature, which is just a little negatively related to $y$ when $p_{test} = 0.1$ in the test environment. In this case, $L_{train} = p_{train}L_1 + (1 - p_{train})L_2 = 0.55L_1 + 0.45L_2$ and $L_{test} = p_{test}L_1 + (1 - p_{test})L_2 = 0.1L_1 + 0.9L_2$, and $L_{test} - L_{train} = -0.45L_1 + 0.45L_2 = 0.45(L_2 - L_1) > 0$, but in this case $w_{sp}$ is small, so we have $L_2 - L_1$ is small according to (B.11), so the difference between training error and test error is not too large, so the generalization performance for this case is not bad, which is consistent with our experimental results.*

3. *Finally, it comes to the failure case in our experimental results, i.e., using $S_1, S_2$ as the training environments and $S_3$ as the test environment. In this case, the overall $p_{train} =*

$\frac{0.1+0.2}{2} = 0.15$, *so the classifier depends more on the spurious features compared with the above two cases, what's more, since $p_{test} = 0.9$, the relationship between $x_{sp}$ and $y$ is almost reversed when we switch to the test stage, so we have $L_{train} = p_{train}L_2 + (1 - p_{train})L_1 = 0.85L_1 + 0.15L_2$ and $L_{test} = p_{test}L_2 + (1 - p_{test})L_1 = 0.1L_1 + 0.9L_2$ and $L_{test} - L_{train} = -0.75L_1 + 0.75L_2 = 0.75(L_2 - L_1) > 0$, and the $w_{sp}$ here is much larger than that in case 2, which means that $L_2 - L_1$ here is much larger than that in case 2, so the gap between training error and test error here map be much larger than that in case two, leading to the poor generalization performance in practice.*

*We can see that the analysing results here are consistent with our experimental results for ColoredM-NIST in Appendix D.2.*

## C Experimental Settings

We run each algorithm 20 times and 1 trial (since adversarial training is time-consuming, we just run 1 trial rather than 3 trials as done in DomainBed). We use part of the training data as the validation set to select the best model of the 20 runs according to the adversarial robustness of the training environments. Following DomainBed, we use random hyper-parameters in the 20 runs.

### C.1 Attacked Algorithms

1. Empirical Risk Minimization (**ERM**, [61]) minimizes the errors across domains.

2. Meta-Learning for Domain Generalization (**MLDG**, [39]) leverages MAML [16] to meta-learn how to generalize across domains.

3. Class-conditional DANN (**C-DANN**, [42]) is a variant of DANN [17] matching the conditional distributions $\mathbb{P}[\phi(X^d)|Y^d = y]$ across domains, for all labels $y$.

4. Risk Extrapolation (**VREx**, [33]) approximates IRM with a variance penalty.

5. Representation Self-Challenging (**RSC**, [28]) learns robust neural networks by iteratively discarding (challenging) the most activated features.

6. Domain-wise Multiple-perturbation Adversarial Training (**MAT**, [64]) use an universal adversarial perturbation (UAP) with low rank along the dimension of examples (for $n$ examples, use the convex combination of $k$ ($k < n$) perturbations as the universal perturbation) to conduct universal adversarial training (UAT) to improve the OOD generalization.

7. Adversarial Training with Low-rank Decomposed perturbations(**LDAT**, [64]) shares similar idea with MAT, but it uses an UAP with low rank along the input space (for $N \times N$ images, it constrains the $N \times N$ UAP matrix has rank $l < N$) to conduct universal adversarial training (UAT) to improve the OOD generalization.

### C.2 Datasets

We use the following datasets provided by the DomainBed [26]:

1. **ColoredMNIST** [4] is a variant of the MNIST handwritten digit classification dataset [35]. Domain $d \in \{0.1, 0.3, 0.9\}$ contains a disjoint set of digits colored either red or blue. The label is a noisy function of the digit and color, such that color bears correlation $d$ with the label and the digit bears correlation $0.75$ with the label. This dataset contains 70000 examples of dimension $(2, 28, 28)$ and 2 classes.

2. **RotatedMNIST** [21] is a variant of MNIST where domain $d \in \{0, 15, 30, 45, 60, 75\}$ contains digits rotated by $d$ degrees. Our dataset contains 70000 examples of dimension $(1, 28, 8)$ and 10 classes.

3. **PACS** [38] comprises four domains $d \in \{$ art, cartoons, photos, sketches $\}$. This dataset contains 9991 examples of dimension $(3, 224, 224)$ and 7 classes.

4. **VLCS** [15] comprises four photographic domains $d \in \{$ Caltech101, LabelMe, SUN09, VOC2007 $\}$. This dataset contains 10729 examples of dimension $(3, 224, 224)$ and 5 classes.

5. **OfficeHome** [62] includes domains four $d \in \{$ art, clipart, product, real $\}$. This dataset contains 15588 examples of dimension $(3, 224, 224)$ and 65 classes.

## C.3 Settings for attacking standard OOD algorithms

We evaluate the adversarial robustness of some of the algorithms in Table 1. For adversarial attacks, we use $\ell_\infty$ adversarial perturbation upper bound $\epsilon = 0.1$ for RotatedMNIST and ColoredMNIST, $\epsilon = 4/255$ for others (it is because $\epsilon = 4/255$ is use usually used in $224 \times 224$ colored images); we use the classical FGSM and PGD-20 as the attack methods, for PGD-20, the step size $\alpha = \epsilon/4$, we realize the attacks according to the package 'torchattacks' [32].

We train the models in the same hyper-parameter random search setting (except that we just train the model for one trial) in DomainBed, and the we attack the models trained with the best hyper-parameters choosed by the **training-domain validation set** model selection method in DomainBed with the same seed as used during training. For MAT and LDAT, we use the same hyper-parameter random search setting in [64] except that we train with 20 random hyper-parameter for 1 trial and they train with 8 random hyper-parameter for 3 trials.

## C.4 Settings for OOD adversarial algorithms

In this subsection, we introduce the settings in our experiments in Section 5. We use the **training-domain validation set** model selection method in the experiment and train the model with 20 random hyper-parameter for 1 trial.

**Basic setting for training hyper-parameter random search.**

1. For AT, we use the same basic random search setting as ERM.

2. For RDANN, we use the same basic random search setting as DANN.

**Setting for adversarial training.** We use PGD-10 as the adversarial attack to generate adversarial examples for adversarial training, with $\epsilon = 0.1$ for RotatedMNIST and ColoredMNIST and $\epsilon = 4/255$ for the other $224 \times 224$ datasets, and we use the attack step size $\alpha = \epsilon/4$ as usually done in the adversarial training community.

**Setting for evaluating adversarial robustness.** We consider two attack methods, FGSM and PGD-20, to evaluate the OOD adversarial robustness of the trained models. We use the same $\epsilon$ as that used for training but we do 20 iterations for PGD in the evaluation stage rather than just 10 iterations in the training stage to avoid overfitting, which is also a strategy used by the adversarial robustness community.

# D Additional Experimental Results

In this section, we show the detailed adversarial robustness of the current OOD generalization algorithms and our proposed algorithms, here we show the adversarial robustness in each test environment, for the ColoredMNIST dataset and RotatedMNIST dataset, we set $\epsilon = 0.1$, and for other $224 \times 224$ datasets, we set $\epsilon = \frac{4}{255}$.

In Appendices D.1 to D.3, we use FGSM and PGD-20 as the attacks to evaluate the adversarial robustness and the results are shown in the form of triple tuple **(clean accuracy / accuracy under FGSM attak / accuracy under PGD-20 attack)**. Each column represents the results for a test environment. The tables show that compared with existing methods, our methods (AT and RDANN) significantly improve the adversarial robustness of the model on the target domains.

In Appendix D.6, we use AutoAttack as the attacks to evaluate the adversarial robustness. Each column represents the results for a test environment. The results show that under AutoAttack, all the existing methods fails (**no more than** $1\%$ adversarial accuracy), and our methods significantly improves the adversarial robustness.

## D.1 RotatedMNIST

| Algorithm | 0 | 15 | 30 | 45 | 60 | 75 | Avg |
|---|---|---|---|---|---|---|---|
| ERM | 93.8 / 11.9 / 1.3 | 98.7 / 10.0 / 1.1 | 99.2 / 13.5 / 0.5 | 99.2 / 19.2 / 0.2 | 99.0 / 19.1 / 0.3 | 95.8 / 11.8 / 0.2 | 97.6 / 14.2 / 0.6 |
| MLDG | 94.8 / 14.1 / 0.4 | 98.7 / 15.0 / 0.2 | 99.1 / 11.8 / 0.0 | 99.1 / 17.2 / 0.0 | 98.8 / 15.3 / 0.8 | 96.7 / 16.9 / 0.0 | 97.9 / 15.1 / 0.2 |
| CDANN | 95.7 / 11.4 / 0.0 | 98.7 / 12.2 / 2.7 | 98.8 / 16.1 / 0.5 | 98.9 / 12.3 / 2.1 | 98.9 / 16.5 / 0.0 | 95.9 / 22.9 / 0.3 | 97.8 / 15.2 / 0.9 |
| VREx | 95.1 / 8.3 / 0.3 | 98.6 / 9.3 / 0.1 | 98.6 / 11.9 / 0.1 | 99.0 / 13.4 / 0.0 | 98.8 / 14.4 / 0.4 | 96.3 / 9.2 / 0.0 | 97.7 / 11.1 / 0.2 |
| RSC | 95.5 / 11.4 / 1.5 | 98.4 / 11.3 / 0.0 | 99.3 / 11.6 / 2.3 | 99.1 / 12.0 / 0.7 | 99.0 / 21.6 / 1.0 | 96.0 / 9.9 / 0.3 | 97.9 / 13.0 / 1.0 |
| AT | 97.0 / 91.8 / 90.8 | 99.2 / 96.1 / 94.8 | 99.2 / 96.6 / 95.6 | 99.0 / 96.3 / 95.4 | 99.0 / 96.2 / 95.1 | 97.2 / 90.9 / 88.9 | 98.4 / 94.7 / 93.4 |
| RDANN | 96.1 / 91.5 / 90.8 | 99.2 / 96.0 / 94.7 | 99.2 / 96.4 / 95.3 | 99.2 / 96.7 / 95.6 | 99.1 / 96.4 / 95.5 | 97.3 / 91.1 / 89.0 | 98.4 / 94.7 / 93.5 |

## D.2 ColoredMNIST

| Algorithm | +90% | +80% | -90% | Avg |
|---|---|---|---|---|
| ERM | 71.7 / 20.0 / 2.2 | 72.8 / 29.8 / 5.5 | 9.8 / 9.8 / 9.8 | 51.5 / 19.9 / 5.8 |
| MLDG | 72.3 / 16.6 / 0.1 | 73.4 / 29.5 / 5.3 | 9.7 / 9.5 / 9.0 | 51.8 / 18.5 / 4.8 |
| CDANN | 71.4 / 31.3 / 11.7 | 72.5 / 29.5 / 4.6 | 10.0 / 8.9 / 8.1 | 51.3 / 23.2 / 8.2 |
| VREx | 72.9 / 24.3 / 3.7 | 73.9 / 29.8 / 5.9 | 10.3 / 10.1 / 9.8 | 52.3 / 21.4 / 6.4 |
| RSC | 72.7 / 36.7 / 1.2 | 72.9 / 36.3 / 3.7 | 10.2 / 9.1 / 6.7 | 51.9 / 27.3 / 3.9 |
| MAT | 72.7 / 40.6 / 21.2 | 73.9 / 27.9 / 1.6 | 9.9 / 9.6 / 9.3 | 52.2 / 26.0 / 10.7 |
| LDAT | 72.1 / 28.3 / 7.0 | 72.8 / 36.4 / 7.4 | 9.7 / 9.5 / 9.3 | 51.5 / 24.7 / 7.9 |
| AT | 73.7 / 72.4 / 72.3 | 73.7 / 72.6 / 72.5 | 10.2 / 10.2 / 10.2 | 52.5 / 51.7 / 51.6 |
| RDANN | 73.3 / 72.1 / 72.0 | 73.2 / 71.8 / 71.7 | 9.7 / 9.7 / 9.7 | 52.1 / 51.2 / 51.1 |

## D.3 VLCS

| Algorithm | C | L | S | V | Avg |
|---|---|---|---|---|---|
| ERM | 98.2 / 47.4 / 0.0 | 62.7 / 21.7 / 0.0 | 72.1 / 4.4 / 0.0 | 71.9 / 6.0 / 0.0 | 76.2 / 19.9 / 0.0 |
| MLDG | 98.5 / 50.0 / 0.0 | 63.5 / 24.3 / 0.0 | 73.1 / 4.9 / 0.0 | 79.7 / 10.0 / 0.0 | 78.7 / 22.3 / 0.0 |
| CDANN | 98.1 / 58.4 / 0.0 | 64.8 / 11.9 / 0.0 | 74.6 / 10.4 / 0.0 | 77.1 / 25.8 / 12.1 | 78.7 / 26.6 / 3.0 |
| VREx | 98.4 / 56.2 / 0.0 | 64.9 / 16.2 / 0.0 | 69.7 / 2.7 / 0.0 | 78.4 / 13.6 / 0.0 | 77.9 / 22.2 / 0.0 |
| RSC | 97.9 / 35.7 / 0.0 | 64.9 / 12.5 / 0.0 | 75.5 / 3.0 / 0.0 | 77.8 / 4.3 / 0.0 | 79.0 / 13.9 / 0.0 |
| MAT | 98.9 / 52.7 / 0.0 | 64.8 / 18.3 / 0.0 | 68.7 / 2.3 / 0.0 | 80.7 / 9.5 / 0.0 | 78.3 / 20.7 / 0.0 |
| LDAT | 97.2 / 44.8 / 0.0 | 65.8 / 20.9 / 0.0 | 70.8 / 2.9 / 0.0 | 73.9 / 5.2 / 0.0 | 76.9 / 18.5 / 0.0 |
| AT | 74.2 / 64.8 / 63.5 | 53.0 / 40.8 / 39.5 | 47.2 / 36.2 / 34.9 | 49.6 / 35.1 / 32.4 | 56.0 / 44.2 / 42.6 |
| RDANN | 69.3 / 66.5 / 66.3 | 53.2 / 43.2 / 42.1 | 50.3 / 37.7 / 35.5 | 54.4 / 39.4 / 35.7 | 56.8 / 46.7 / 44.9 |

## D.4 PACS

| Algorithm | A | C | P | S | Avg |
|---|---|---|---|---|---|
| ERM | 86.8/6.7/0.0 | 76.1/29.5/0.9 | 98.1/55.5/0.0 | 67.9/23.5/0.2 | 82.2/28.8/0.3 |
| MLDG | 86.8/10.3/0.0 | 75.2/30.0/0.2 | 97.6/62.5/0.0 | 76.1/29.6/0.1 | 83.9/33.1/0.1 |
| CDANN | 91.3/18.3/1.0 | 78.4/31.4/0.3 | 96.7/53.7/4.0 | 77.1/31.0/0.6 | 85.9/33.6/1.5 |
| VREx | 86.3/8.1/0.0 | 80.5/30.0/0.2 | 98.1/51.2/0.0 | 68.7/27.0/0.0 | 83.4/29.1/0.0 |
| RSC | 86.2/6.4/0.0 | 76.4/29.4/0.4 | 97.5/59.7/0.0 | 72.2/27.0/0.1 | 83.1/30.6/0.1 |
| MAT | 88.3/11.8/0.0 | 80.5/33.5/1.1 | 97.4/57.6/0.1 | 77.4/32.2/1.5 | 85.9/33.8/0.7 |
| LDAT | 87.2/8.3/0.0 | 76.1/26.6/0.1 | 97.8/54.0/0.0 | 74.0/33.1/0.2 | 83.8/30.5/0.1 |
| AT | 59.8/30.9/24.5 | 71.4/56.7/54.9 | 75.7/60.4/57.6 | 60.1/55.8/55.5 | 66.7/50.9/48.1 |
| RDANN | 63.8/31.8/25.0 | 63.3/50.5/48.2 | 76.8/62.3/58.2 | 68.5/61.8/61.1 | 68.1/51.6/48.1 |

## D.5 OfficeHome

| Algorithm | A | C | P | R | Avg |
|---|---|---|---|---|---|
| ERM | 62.1/12.6/0.0 | 52.9/20.9/0.3 | 76.4/25.7/0.5 | 76.0/26.3/0.6 | 66.9/21.4/0.4 |
| MLDG | 61.1/13.7/0.6 | 52.9/22.1/0.7 | 76.5/25.7/0.3 | 77.4/24.0/0.9 | 67.0/21.4/0.6 |
| CDANN | 61.8/8.5/0.1 | 53.5/19.3/0.1 | 75.8/24.4/0.1 | 77.6/23.7/0.0 | 67.2/19.0/0.1 |
| VREx | 58.7/8.3/0.0 | 52.4/20.0/0.1 | 75.5/22.5/0.7 | 76.1/23.3/1.0 | 65.7/18.5/0.4 |
| RSC | 59.3/9.1/0.2 | 52.1/18.3/0.3 | 75.2/20.5/1.0 | 74.1/20.6/1.2 | 65.1/17.1/0.7 |
| MAT | 57.3/11.2/0.1 | 54.0/20.2/0.5 | 75.2/26.9/0.8 | 77.4/24.1/1.8 | 66.0/20.6/0.8 |
| LDAT | 61.2/12.9/0.0 | 52.7/20.6/0.4 | 74.5/27.9/0.6 | 78.0/29.5/0.8 | 66.6/22.7/0.4 |
| AT | 29.6/16.4/14.5 | 44.6/36.4/35.0 | 53.2/40.9/38.3 | 53.7/36.1/32.9 | 45.3/32.4/30.2 |
| RDANN | 30.0/17.1/15.3 | 41.9/33.6/32.1 | 48.8/38.2/36.3 | 47.1/33.5/30.8 | 41.9/30.6/28.6 |

## D.6 Results for AutoAttack

### D.6.1 RotatedMNIST

| Algorithm | 0 | 15 | 30 | 45 | 60 | 75 | Avg |
|---|---|---|---|---|---|---|---|
| ERM | 0.0 | 0.0 | 0.0 | 0.0 | 0.0 | 0.0 | 0.0 |
| MLDG | 0.0 | 0.0 | 0.0 | 0.0 | 0.0 | 0.0 | 0.0 |
| CDANN | 0.0 | 0.0 | 0.0 | 0.0 | 0.0 | 0.1 | 0.0 |
| VREx | 0.0 | 0.0 | 0.0 | 0.0 | 0.0 | 0.0 | 0.0 |
| RSC | 0.0 | 0.0 | 0.0 | 0.0 | 0.0 | 0.0 | 0.0 |
| AT | 90.6 | 94.7 | 95.5 | 95.3 | 95.0 | 88.6 | 93.3 |
| RDANN | 90.6 | 94.6 | 95.2 | 95.6 | 95.5 | 88.6 | 93.3 |

### D.6.2 VLCS

| Algorithm | C | L | S | V | Avg |
|---|---|---|---|---|---|
| ERM | 0.0 | 0.0 | 0.0 | 0.0 | 0.0 |
| MLDG | 0.0 | 0.0 | 0.0 | 0.0 | 0.0 |
| CDANN | 0.0 | 0.0 | 0.0 | 0.0 | 0.0 |
| VREx | 0.0 | 0.0 | 0.0 | 0.0 | 0.0 |
| RSC | 0.0 | 0.0 | 0.0 | 0.0 | 0.0 |
| MAT | 0.0 | 0.0 | 0.0 | 0.0 | 0.0 |
| LDAT | 0.0 | 0.0 | 0.0 | 0.0 | 0.0 |
| AT | 63.2 | 38.7 | 33.9 | 31.4 | 41.8 |
| RDANN | 65.6 | 41.1 | 34.7 | 34.3 | 43.9 |

### D.6.3 PACS

| Algorithm | A | C | P | S | Avg |
|---|---|---|---|---|---|
| ERM | 0.0 | 1.8 | 0.0 | 0.7 | 0.6 |
| MLDG | 0.0 | 0.5 | 0.0 | 0.7 | 0.3 |
| CDANN | 0.0 | 0.6 | 0.0 | 0.4 | 0.3 |
| VREx | 0.0 | 0.5 | 0.0 | 0.9 | 0.3 |
| RSC | 0.0 | 0.9 | 0.0 | 0.6 | 0.4 |
| MAT | 0.0 | 1.5 | 0.1 | 4.0 | 1.4 |
| LDAT | 0.0 | 0.5 | 0.0 | 0.9 | 0.3 |
| AT | 23.4 | 54.7 | 57.0 | 55.2 | 47.6 |
| ADANN | 24.5 | 48.2 | 58.0 | 61.1 | 48.0 |

## D.7   OfficeHome

| Algorithm | A | C | P | R | Avg |
|---|---|---|---|---|---|
| ERM | 0.0 | 0.0 | 0.1 | 0.1 | 0.0 |
| MLDG | 0.0 | 0.1 | 0.2 | 0.1 | 0.1 |
| CDANN | 0.0 | 0.1 | 0.0 | 0.1 | 0.0 |
| VREx | 0.0 | 0.1 | 0.2 | 0.1 | 0.1 |
| RSC | 0.0 | 0.1 | 0.1 | 0.0 | 0.0 |
| MAT | 0.0 | 0.3 | 0.2 | 0.1 | 0.1 |
| LDAT | 0.0 | 0.0 | 0.1 | 0.1 | 0.1 |
| AT | 14.2 | 34.6 | 38.1 | 32.4 | 29.8 |
| ADANN | 14.0 | 31.5 | 34.4 | 29.7 | 27.4 |

