# OpenReview forum: "On the Adversarial Robustness of Out-of-distribution Generalization Models"
_NeurIPS.cc/2023/Conference — NeurIPS 2023 poster_

### Official Review · Reviewer_sycY · 2023-07-02

**Soundness:** 3 good
**Presentation:** 3 good
**Contribution:** 3 good
**Rating:** 7
**Confidence:** 4

**Summary:**

This paper studies the adversarial robustness of models on the unseen target domain. Firstly, exhaustive experiments show that existing OOD generalization methods are vulnerable to adversarial attacks. Then, a theoretical analysis for the OOD adversarial robustness is presented. Finally, this paper proposes two methods (AERM and RDANN) based on the theoretical results.

**Strengths:**

- The topic of this paper is novel and the paper considers two perspectives of trustworthy machine learning, which is very useful in practice. I agree with the authors that the adversarial robustness of the model on the OOD domain is very important since in the real-world task, the data we face always has a different style from what we use during the training stage.
- The theoretical analysis is strong and interesting. The authors provide theoretical analyses in two different settings (the average setting and the agnostic setting where the distance between the training domains and the test domain is unknown), which cover the practical cases according to how many domains we can sample from. I think the theoretical analysis of this paper is very comprehensive.
- According to the experimental results, the proposed methods significantly improve the OOD adversarial robustness. Furthermore, the observations on RotatedMNIST and ColoredMNIST are quite interesting and are consistent with the theoretical results of the two examples, which motivates researchers to consider the hardness of the tasks in OOD generalization.
- The paper is well-written. The details of the methods are adequately explained, the proofs of the theorems are presented step-by-step, and the meaning of the theoretical results is discussed in detail.


**Weaknesses:**

- There are other datasets in the DomainBed benchmark, however, the experiments are conducted only in three datasets (RotatedMNIST, ColoredMNIST, and VLCS). Although AT is time-consuming and DomainBed involve running many groups of random parameters, I think the authors show add at least one more dataset in DomainBed.

**Questions:**

As I said in the weaknesses part, would you please provide additional experimental results for other datasets (at least one) in DomainBed?

**Limitations:**

No.

---

> ### Author Rebuttal · Authors · 2023-08-02
>
> Thank you for your meaningful suggestions.
>
> **For the weaknesses:** Thank you for your suggestion, we have extended our experimental results to the PACS dataset and OfficeHome dataset in DomainBed, please refer to **Table 1** in the reply to all reviewers part. From the table we can see that: on the datasets PACS and OfficeHome, the baselines achieve **no more than $2\%$** average adversarial accuracy under the attack PGD-20 and AutoAttack. Again, our proposed methods can significantly improve the average adversarial accuracy, which is consistent with the results of the experiments in our paper.
>
>
>
> **For the questions:** Please refer to the answer to the weaknesses.

---

> > ### Comment · Reviewer_sycY · 2023-08-15
> > **Thanks for the rebuttal**
> >
> > Thank you for your reply. Thank you for conducting experiments on additional two datasets. According to the results in Table 1, I believe that you have sufficient evidence to claim that your methods work well. I keep my score on acceptance.

---

> > > ### Author Response · Authors · 2023-08-15
> > >
> > > Thank you for your approval of our paper. We will incorporate our additional results in the revision.

---

### Official Review · Reviewer_sfFB · 2023-07-02

**Soundness:** 4 excellent
**Presentation:** 4 excellent
**Contribution:** 4 excellent
**Rating:** 7
**Confidence:** 4

**Summary:**

The paper first evaluates the adversarial robustness of current OOD generalization algorithms and shows that existing OOD generalization algorithms are vulnerable to adversarial attacks. Then, the paper presents a theoretical analysis of the adversarial robustness (of a model) on the unseen test domain. After that, the paper designs two algorithms to improve the OOD adversarial robustness inspired by the theoretical implications. Finally, extensive experiments are conducted and the improvement is significant.

**Strengths:**

(1) This paper takes the first step to study the adversarial robustness of a model in the out-of-distribution generalization setting. As the extensive experiments show, existing out-of-distribution generalization methods fail under adversarial attacks, which means that it is urgent to study this problem. So, I believe that the studied problem is meaningful for the machine learning reliability community.

(2) The writing is good and the paper is easy to follow. On the one hand, the proof steps are so detailed that it is easy to check the correctness of the theorems. On the other hand, after the theorems, the paper provides some remarks to show the implications of the theorems to help the readers understand the messages that the theorems convey.

(3) The empirical results are strong. The improvement on the out-of-distribution adversarial robustness is significant. The experiments are extensive, and the details of the experimental settings are described in detail.

(4) The proposed algorithms are inspired by the theory, which means that they are guaranteed by the theory. The theoretical analyses are technically solid. This paper not only provides theoretical results in general settings (Theorem 2.1, Corollary 2.2, and Theorem 2.3) where the data distributions and the hypothesis class are not specified but also studies an intuitive example (Example 1, Theorem 2.4 and Example 2, Theorem C.1) that presents many intuitive understandings.


**Weaknesses:**

(1) The paper [1] seems to be very relevant to this paper, but the discussions are missing.

(2) The clean out-of-distribution generalization accuracy on VLCS decreases for both AERM and RDANN, however, this does not happen for ColoredMNIST and RotatedMNIST. The authors do not provide adequate discussion on this phenomenon.

[1] Ibrahim, Adam, et al. "Towards Out-of-Distribution Adversarial Robustness." arXiv preprint arXiv:2210.03150 (2022).


**Questions:**

(1) Could you provide a discussion about the differences between this paper and [1]?

(2) Please provide a discussion about why the clean accuracy of the proposed methods decreases on VLCS but not on ColoredMNIST and RotatedMNIST.

---

> ### Author Rebuttal · Authors · 2023-08-02
>
> Thank you for your meaningful suggestions.
>
> **For the weakness (1)**: Thank you for your kind suggestion. We are sorry for not noticing such a related work. Although the title of [1] is about the OOD adversarial robustness, the topic that [1] studies is quite different from ours. [1] investigates how to improve the adversarial robustness of a model against ensemble attacks or unseen attacks. [1] regards the adversarial examples for each type of attack (such as FGSM, PGD, CW, and so on) as a domain, and utilizes the OOD generalization methods to improve the models generalization performance under different (maybe unseen) attacks. While we study the adversarial robustness of a model on the unseen target domain.
>
> **For the weakness (2)**: In the community of adversarial robustness, it is a well-known phenomenon that there is a trade-off between the clean accuracy and adversarial robustness. There are many papers trying to explain it theoretically[2] and empirically[3], so it is intuitive that there is still a trade-off between the OOD clean accuracy and OOD adversarial robustness in the OOD setting. Perhaps the more interesting phenomenon is that the OOD clean accuracies on RotatedMNIST and ColoredMNIST do not drop. The experiments in [4] show that in the supervised learning setting (no distributional shift), conducting adversarial training on the MNIST dataset does not drop the clean accuracy (Table 4 and Table 5 in [4]), which is consistent with our experiment results. For the reason that the clean OOD accuracies on MNIST datasets do not drop, we think it is that the MNIST dataset is too simple.
>
> **For the questions:** Please refer to the answer to the weaknesses.
>
>
> **References**
>
> [1] Ibrahim, Adam, et al. "Towards Out-of-Distribution Adversarial Robustness." arXiv preprint arXiv:2210.03150 (2022).
>
> [2] Javanmard, Adel, Mahdi Soltanolkotabi, and Hamed Hassani. "Precise tradeoffs in adversarial training for linear regression." Conference on Learning Theory. PMLR, 2020.
>
> [3] Madry, Aleksander, et al. "Towards deep learning models resistant to adversarial attacks." arXiv preprint arXiv:1706.06083 (2017).
>
> [4] Li, Yan, et al. "Implicit bias of gradient descent based adversarial training on separable data." International Conference on Learning Representations. 2020.

---

> > ### Comment · Reviewer_sfFB · 2023-08-15
> >
> > I would like to thank the authors for their reply.
> >
> > I think the authors have solved my questions. Please incorporate the answer to weakness 1 in the related works part. It is better to incorporate the answer to weakness 2 in the experiments part.
> >
> > I think this is a good paper and recommend acceptance.
> >
> > Best,
> > The reviewer

---

> > > ### Author Response · Authors · 2023-08-15
> > >
> > > We are glad that we have solved all your questions. Thank you for your approval of our paper.

---

### Official Review · Reviewer_rjrX · 2023-07-04

**Soundness:** 2 fair
**Presentation:** 1 poor
**Contribution:** 2 fair
**Rating:** 3
**Confidence:** 5

**Summary:**

This paper studies the adversarial robustness of OOD generalization algorithms. Authors claimed that existing OOD generalization algorithms fail in adversarial attacks (FGSM and PGD). To this end, authors first theoretically analyze the problem definition and generalization bound of OOD algorithms in adversarial settings. Then, they proposed two algorithms: adversarial ERM and robust DANN to solve the problem. Experiments on several datasets show the effectiveness of the algorithms.

**Strengths:**

1. The problem of adversarial OOD generalization is interesting and of great importance in real scenarios.
2. The theoretical parts offer valuable insights in the algorithm design.
3. The experimental results are showing the advantages of the algorithms.

**Weaknesses:**

I think this paper can be seen as a pioneering work in exploring adversarial robustness of OOD generalization algorithms, but unfortunately has many flaws:
- Lack of related work discussion. Although the discussion of adv. and OOD is rare, there are certainly some existing works on this intersection. For instance, [Yi et al'2021, ICML] theoretically studied the bounds of adversarial and OOD generalization, and they pointed that at mild conditions, adversarial robustness can guarantee OOD generalization (Thm 3.1). And [Alhamound'2022, arXiv] studies the generalization of adv. algorithms, which is of high interest to your work.
- The algorithms are intuitive and not novel. I highly appreciate the efforts in deriving the theories and then lead to the algorithm design. But these two algorithms are very simple and intuitive to design. For instance, the average ERM over several adv. distributions and average domain-adversarial training are easy to propose and can even be seen as baselines when one tries to solve the adv. OOD generalization problems. The only reason is that no one has explored this before. But to me, the two algorithms are very simple and not novel.
- The theoretical part is dense, which I give my appreciation. But this part lacks enough motivation and is hard to read. I strongly encourage authors to revise this to better align with the main topic. In addition, most of the theories are based on Ben-David et al's distribution gap theory and do not present much technical novelty.
- The experiments are lacking sufficient backbones and comparison methods:
1. Lack of various backbones in addition to ResNet50. Have you tried Vision Transformer? ViTs are known to be better cope with adv. robustness than CNNs.
2. Lack of other datasets. The largest dataset in this paper is VLCS, which is not enough for OOD generalization. You should at least utilize the DomainBed datasets.
3. Lack of adversarial comparison methods. Adversarial robustness is an active area and I'm sure most of them can be applied to OOD algorithms. Authors did not compare with any existing efforts.
4. Lack of interpretatbility. Why do the algorithms in this paper work? Or why do existing algorithms not work?



References:

[Yi et al'2021, ICML] Improved OOD Generalization via Adversarial Training and Pre-training.

[Alhamound'2022, arXiv] Generalizability of Adversarial Robustness Under Distribution Shifts.

**Questions:**

See weakness.

**Limitations:**

See weakness

---

> ### Author Rebuttal · Authors · 2023-08-03
>
> Thank you for your suggestions.
>
> **Comparison with related works.** We are sorry for ignoring the related works, we now provide comparisons and will incorporate them in the revision.
> - [Yi et al'2021, ICML] studies the relationship between adversarial robustness and OOD generalization, it shows that good adversarial robustness implies good OOD performance when the target domain lies in a Wasserstein ball. While we study the OOD adversarial robustness and propose algorithms to improve OOD adversarial robustness.
> - [Alhamound'2022, arXiv] empirically analyzes the transferability of models' adversarial/certified robustness under distributional shifts. It shows that adversarially trained models do not generalize better without fine-tuning and that the accuracy-robustness trade-off generalizes to the unseen domain. Its results for adversarial robustness can also be found in our experimental results. Moreover, we find an exception that the robustness and accuracy can be reconcilable on MNIST-based datasets. We think it is due to the simplicity of MNIST-based datasets. Moreover, we provide theoretical analyses.
>
>
> **The novelty of the algorithms**
> - Firstly, the algorithms are theory-driven, we just propose algorithms to optimize the upper bounds of the target domain risk. We believe that good algorithms needn't be complicated or tricky, they just need to align well with the theory and perform well in practice. The experiments show that our methods perform well in practice.
> - Secondly, RDANN is not intuitive. A trivial method is running DANN [1] with adversarial examples, however, we have tried it but the training is unstable and the performance is bad. DANN is popular in domain adaptation **(DA)** and is based on the theory of [2]. [2] defines $d _\mathcal{H}(S,T)=\sup _h|P _S[h(x)=1]-{P} _T[h(x)=1]|$ and computes the discrepancy by training a domain classifier (Lemma 2), which is used by DANN. However, **in multi-class case, no theory shows that we can compute the discrepancy in this way. So there is a gap between DANN and its theory in the multi-class case**. To address this issue, we define a new discrepancy in terms of labels (the target labels are not available in DA, but source labels are provided in the OOD setting), perturbation set, and loss function, where $d _{\ell(\mathcal{H})}^\mathcal{B}(S,T)=\sup _h|E _{(x,y)\sim S} [\sup _{x^\prime\in\mathcal{B}(x)}\ell(h(x^\prime),y)]-E _{(x,y)\sim T} [\sup _{x^\prime\in\mathcal{B}(x)}\ell(h(x^\prime),y)]|$ and it **can characterize the multi-class adversarial robustness better**. RDANN optimizes the upper bound in Theorem 2.3 and calculates the discrepancy $d _{\ell(\mathcal{H})}^\mathcal{B}$ according to its definition, which **aligns well with the theory** and makes RDANN more stable and perform better.
>
> **The presentation of the theoretical part**
> - **Logical line and the motivations**. Motivated by the failure of existing OOD generalization methods, we theoretically study the OOD adversarial robustness and derive methods to improve models' OOD adversarial robustness. We first analyze the natural case where the domains are drawn from some unknown distribution $p$, since after we deploy the system, we can naturally expect the input examples to be of different styles (i.e., from different domains). Then we get Theorem 2.1 where the convergence rate is $O(t^{-0.5})$. However, the source domain number $t$ is limited in practice, so we derive a target error bound with a faster rate $O(n^{-0.5})$ in Theorem 2.3, where $n\gg t$. However, the term $d_ {\ell(\mathcal{H})}^\mathcal{B}(\mathcal{T},\mathcal{T} _P)$ in Theorem 2.3 is uncontrollable since it depends on the target domain. We regard $d _{\ell(\mathcal{H})}^\mathcal{B}(\mathcal{T},\mathcal{T} _P)$ as a quantity that describes the difficulty of the task and provide Example 1 and Theorem 2.4 to support this point.
> - **The novelty and contributions of our theory**. Our definition of the discrepancy is different from that of [2] and our theory is not based on [2]. Please refer to the second point of "The novelty of the algorithms" part, which indicates that **the theory of [2] only works for binary case, but our theory works for the multi-class case and describes the adversarial robustness rather than accuracy**. Furthermore, Example 1 and Theorem 2.4 show that the target adversarial robustness is related to the task difficulty measured by the $d_ {\ell(\mathcal{H})}^\mathcal{B}(\mathcal{T},\mathcal{T} _P)$. And the experimental results align well with Theorem 2.4.
>
> **The experiments part**
> 1. Our experiments are based on a popular benchmark DomainBed, where ResNet50 is the largest network. DomainBed chooses the best results in 20 runs with different random seeds, which is time-consuming, especially when we use adversarial examples. Moreover, ViT is larger than ResNet50, we cannot finish it due to time limitations but are glad to report the results in the revision.
> 2. Due to time limitations, we conduct additional experiments on PACS and OfficeHome, the results are in **Table 1** in the "reply to all reviewers" part. The additional results are consistent with that in our paper.
> 3. We claim that our methods are orthogonal to the adversarial methods. Moreover, AERM can be viewed as a variant of basic adversarial training, and it can be adapted to other adversarial training methods.
> 4. Our algorithms are driven by our theory. The algorithms both try to minimize the upper bound of the target adversarial error, which is the reason that our algorithms work. Moreover, RDANN works better than AERM, which is also consistent with our theory. Note that the convergence rate in Theorem 2.1 is $O(t^{-0.5})$ and that in Theorem 2.3 is $O(n^{-0.5})$. $n\gg t$ in practice, so it is expected that RDANN works better than AERM.
>
> **References**
>
> [1] Domain-Adversarial Training of Neural Networks. Ganin et al'2016, JMLR.
>
> [2] A theory of learning from different domains. Ben-David et al'2010, ML.

---

> > ### Comment · Reviewer_rjrX · 2023-08-18
> > **Thanks for your response**
> >
> > I would like to thank the authors for your response. However, I still think this paper lacks technical novelty. So I will not change my rating and lean towards rejection.

---

> > > ### Author Response · Authors · 2023-08-18
> > >
> > > Thank you for your response.

---

### Official Review · Reviewer_YSCG · 2023-07-06

**Soundness:** 3 good
**Presentation:** 2 fair
**Contribution:** 2 fair
**Rating:** 6
**Confidence:** 4

**Summary:**

The authors use statistical learning theory tools and the HdH divergence to motivate methods to improve the performance of a domain generalizing model dealing with adversarially perturbed, out-of-distribution domains. The authors evaluate two methodologies in that setting, and show how this can indeed defend models.

**Strengths:**

1. While OoD generalization is mostly concerned with improving performance on out-of-distribution domains, there is not much work on evaluating the performance of OoD-trained models on adversarially attacking an out-of-distribution domain.
2. The two approaches studied are motivated with theory, and are shown to work well in that setting.


**Weaknesses:**

Fixing the following issues would lead me to accept the paper. They are related to toning down some claims and missing references.
1. Missing references and comparisons. First, the relevant work of “Adversarial Feature Desensitization” by Bashivan et al. 2021. E.g., Bashivan et al. already appear to apply HdH theory to estimate adversarial generalization error bounds, etc. Your setup has some differences in that you’re using pairwise HdH distances on perturbed domains, but it still needs to mention previous HdH approaches to adversarial robustness (and potentially factor that in when making the claims). Moreover, RDANN also echoes “Generalizing to unseen domains via distribution matching” by Albuquerque et al. 2019, e.g., they train on non-perturbed domains while you do, but the logic is quite similar; RDANN would be applying Albuquerque et al.’s approach to adversarially train the model.
2. Please correct me if I am wrong, but AERM appears to just be adversarial training as done by Madry et al. 2018 (and Goodfellow et al. 2014 before). Adversarial training is done over all samples seen during training; it’s orthogonal to having multiple domains or a single one. I value that the authors provide both theoretical motivation for it and experimental results in the particular setting of adversarial robustness of a model trained for OoD generalization. I would however avoid calling the method itself “your” contribution, or giving it a new name; it’s just adversarial training = training on the adversarially perturbed training dataset (agnostically to whether it corresponds to one or multiple domains).
3. The benchmark/discussion of results/claims re: performance are quite strange, as they effectively compare undefended models with two defended models, so seeing an improvement in robustness is quite trivial. I suggest framing the results a bit better to avoid making it about improving over existing methods (of course it does, you’re using adversarial training), but rather discussing how adversarial training (your AERM) compares with the RDANN approach, and how on their own (instead of compared to undefended models), they provide robustness on OoD datasets that are adversarially perturbed. Interesting things with your results are that on RotatedMNIST and ColoredMNIST you do not suffer much from the usual drop in clean (= unperturbed) accuracy as you defend a model, compared to undefended models.


**Typos and suggestions:**
1. I believe the phrase “OOD adversarial robustness” is more indicative/suggestive of robustness against diverse perturbations, which is an existing problem, than “adversarial robustness of models trained for OoD generalization” (in fact, a google/bing search appears to confirm this view, with e.g. “Towards Out-of-Distribution Adversarial Robustness” Ibrahim et al. 2022). For minimal effort, I suggest rephrasing to “adversarial robustness of OoD” if you want to keep it short, to clear up any confusion.
2. I suggest adding equation numbers.
3. The equation below L250, $c_{ij}$ are introduced. I would write in the text what they correspond to (classifiers for pairs of distributions of features of perturbed domains).


**Questions:**

No.

**Limitations:**

Yes.

---

> ### Author Rebuttal · Authors · 2023-08-02
>
> Thank you for your valuable suggestions.
>
> **For weakness 1:** We are sorry for missing the references and comparisons.
> - "Adversarial Feature Desensitization" focuses on improving the adversarial robustness of the models by regarding the adversarial distribution as the target domain and then using the domain adaptation methods, while we focus on improving the model's adversarial robustness on the OOD distribution. Furthermore, due to the availability of the labels for source domains, we do not use $d_ {\mathcal{H} \Delta \mathcal{H}}$ (the HdH distance) but use $d_ {\mathcal{H}}$, which takes better advantages of the label information.
> - For "Generalizing to unseen domains via distribution matching" by Albuquerque et al. 2019, e.g, we claim that RDANN does not echo the method of Albuquerque et al. 2019.
>   - Firstly, in "Generalizing to unseen domains via distribution matching", the authors use $d _\mathcal{H}(S,T)=\underset{h\in\mathcal{H}}{\sup}|P _S [h(x)=1]-{P} _T [h(x)=1]|$ to measure the discrepancy between the domains. Their algorithm computes the discrepancy by training a domain classifier, which is proven to be valid for binary case (Lemma 2, [1]). However, **in multi-class case, no theory shows that we can compute the discrepancy in this way. So in their paper, there is a gap between the theory and algorithm in the multi-class case**. While our discrepancy is defined as $d _{\ell(\mathcal{H})}^\mathcal{B}(S,T)=\sup _h|E _{(x,y)\sim S}[\sup _{x^\prime\in\mathcal{B}(x)}\ell(h(x^\prime),y)]-E _{(x,y)\sim T} [\sup _{x^\prime\in\mathcal{B}(x)}\ell(h(x^\prime),y)]|$, which takes the loss function and labels into consideration. Moreover, our algorithm RDANN optimizes the upper bound in Theorem 2.3 and calculates the discrepancy $d _{\ell(\mathcal{H})}^\mathcal{B}$ according to its definition, which is **consistent with our theory**.
>   - Secondly, we have tried applying Albuquerque et al.’s approach to adversarially train the model, but the training is extremely unstable and the performance is quite bad. This means that directly combining adversarial training with Albuquerque et al.’s approach is not a good choice, while our theory-driven RDANN is stable and works well.
>
> Thank you for your kind suggestion, we will incorporate the references and comparisons in the revision.
>
> **For weakness 2:** Yes, the AERM algorithm is equivalent to conducting adversarial training on multiple source domains. In our paper, we investigate the model's OOD adversarial robustness in two different situations, and our theory-driven AERM algorithm occasionally coincides with the adversarial training method for multiple domains (we are not just trivially applying AT in OOD generalization, but the algorithms are theory-driven).
>
> **For weakness 3:** Thank you for your meaningful suggestions. We will reframe our results to mainly compare and discuss the performance of the AERM and RDANN algorithms. The comparison and discussion about the performance of AEMR and RDANN are as follows(We will incorporate them in the revision):
> - **Comparison**. According to the experimental results in **Table 1** in the "reply to all reviewers" part, we can see that in the small datasets RotatedMNIST and ColoredMNIST, AERM is comparable to RDANN, while on the larger datasets VLCS, PACS, and OfficeHome, RDANN achieves better performance.
> - **Discussions**. As the paper suggests, the risk convergence rate for AERM is of order $\mathcal{O}(t^{-0.5})$ (Theorem 2.1), while the risk convergence rate for RDANN is of order $\mathcal{O}(n^{-0.5})$ (Theorem 2.3). Here, $t$ stands for the number of source domains while $n$ is the number of examples in each source domain. In our datasets, the number of domains is small, so the convergence rate of RADANN is faster than that of AERM, which is consistent with the superiority of RDANN on VLCS, PACS, and OfficeHome. For the comparable results on RotatedMNIST and ColoredMNIST, we think it is because the dataset is too small to distinguish which one of the two is better.
> - **Discussions for not dropping the clean accuracy on RotatedMNIST and ColoredMNIST**. In the community of adversarial robustness, it is a well-known phenomenon that there is a trade-off between clean accuracy and adversarial robustness. There are many papers trying to explain it theoretically[2] and empirically[3], so it is intuitive that there is still a trade-off between the OOD clean accuracy and OOD adversarial robustness in the OOD setting. While the experiments in [4] show that in the supervised learning setting (no distributional shift), conducting adversarial training on the MNIST dataset does not drop the clean accuracy (Table 4 and Table 5 in [4]), which is consistent with our experiment results. For the reason that the clean OOD accuracies on MNIST datasets do not drop, we think it is that the MNIST dataset is too simple.
>
> **For the typos and suggestions:**
> 1. Thank you for your useful suggestions, we will correct them to clear up the confusion in the revision.
> 2. We are sorry for bringing the inconvenience during the review period due to missing equation numbers, we agree that it is a good choice to add the equation numbers, and we will add them in the revision.
> 3. Thank you for your reminder. In fact, in line 252, we state that $c_ {ij}$ refers to the discriminator for domain $i$ and domain $j$.
>
>
> **References**
>
> [1] A theory of learning from different domains. Ben-David et al'2010, ML.
>
> [2] Precise tradeoffs in adversarial training for linear regression. Javanmard et al'2020, COLT.
>
> [3] Towards deep learning models resistant to adversarial attacks. Madry et al'2017. arXiv.
>
> [4] Implicit bias of gradient descent based adversarial training on separable data. Li, Yan, et al'2020, ICLR.

---

> > ### Comment · Reviewer_YSCG · 2023-08-14
> >
> > I would like to thank the authors for their reply.
> >
> > > Weakness 1 [...] Thank you for your kind suggestion, we will incorporate the references and comparisons in the revision.
> >
> > My apologies for mentioning HdH and not H in your case. In the cases of either paper, as stated, my point mostly was that the approach is related enough that those papers should be mentioned. It is good to hear that you will, and even better that one of the methods was empirically evaluated.
> >
> > > For weakness 2: Yes, the AERM algorithm is equivalent to conducting adversarial training on multiple source domains. In our paper, we investigate the model's OOD adversarial robustness in two different situations, and our theory-driven AERM algorithm **occasionally** coincides with the adversarial training method for multiple domains (we are not just trivially applying AT in OOD generalization, but the algorithms are theory-driven).
> >
> > Could you give details as to how/when the AERM loss in 3.1 differs from AT ?
> >
> > > For the reason that the clean OOD accuracies on MNIST datasets do not drop, we think it is that the MNIST dataset is too simple.
> >
> > I agree and suggest having such explanations in the paper. I could not find any Table in [4] btw in any of that paper's versions. Are you sure [4] is the correct reference ? Thanks !

---

> > > ### Author Response · Authors · 2023-08-15
> > >
> > > Thank you for the reply.
> > >
> > > **Q1: Could you give details as to how/when the AERM loss in 3.1 differs from AT?**
> > >
> > > In fact, our AERM loss is the same as AT loss, for the classification problem, we use the frequently-used adversarial cross-entropy loss. We use the word **occasionally** in our rebuttal to explain that we are not adapting the AT method to the OOD setting, but design the AERM algorithm according to our theory and the resulting algorithm can be regard as a multi-domain AT algorithm. Here, we emphasize that in addition to AERM, we also introduce RDANN, which is our main contribution and works better.
> > >
> > > **Q2: I agree and suggest having such explanations in the paper. I could not find any Table in [4] btw in any of that paper's versions. Are you sure [4] is the correct reference ? Thanks!**
> > >
> > > We are sorry for giving the incorrect reference and confusing you. The correct reference is:
> > >
> > > [4] Theoretically Principled Trade-off between Robustness and Accuracy. Zhang et al'2019, Arxiv. https://arxiv.org/pdf/1901.08573.pdf

---

> > > ### Author Response · Authors · 2023-08-17
> > >
> > > Dear Reviewer,
> > >
> > > ​	We wonder whether we have solved your questions and concerns. If you have further questions, we are glad to follow up.

---

> > > > ### Comment · Reviewer_YSCG · 2023-08-17
> > > >
> > > > > We are sorry for giving the incorrect reference and confusing you. The correct reference is:
> > > > [4] Theoretically Principled Trade-off between Robustness and Accuracy. Zhang et al'2019, Arxiv. https://arxiv.org/pdf/1901.08573.pdf
> > > >
> > > > Thank you ! Note that their table still shows that adversarial training reduces the clean accuracy (the more the TRADES penalty term is weighted, the [marginally] lower the clean accuracy, even on MNIST). I do believe that the OoD case may very well be different and perhaps adversarial training acts as a form of beneficial augmentation even for non-adversarial distributional shifts.
> > > >
> > > > > In fact, our AERM loss is the same as AT loss, for the classification problem, we use the frequently-used adversarial cross-entropy loss. We use the word occasionally in our rebuttal to explain that we are not adapting the AT method to the OOD setting, but design the AERM algorithm according to our theory and the resulting algorithm can be regard as a multi-domain AT algorithm. Here, we emphasize that in addition to AERM, we also introduce RDANN, which is our main contribution and works better.
> > > >
> > > > My last remaining concern has to do with "AERM" instead of using "AT" since it's really what it is. I believe the paper should be well contextualised within the area of adversarial machine learning by using what is perhaps the most common terminology aside from "adversarial example": "adversarial training". This is orthogonal to whether they rederive it from theory; it is still adversarial training. If the authors want, they could perhaps reframe "deriving an algorithm, AERM, from our theory" as "we show how adversarial training can be motivated/derived from our analysis in the OoD setting". They can also rename it "AT-OoD" or something if they want, but I believe AT is way too common to not be referred to as such in a paper on adversarial robustness. If the authors agree to make that change, I will be happy to recommend the paper for acceptance.

---

> > > > > ### Author Response · Authors · 2023-08-17
> > > > >
> > > > > Dear Reviewer,
> > > > >
> > > > > Thank you for your valuable suggestions and being willing to recommend our paper for acceptance.
> > > > >
> > > > > For the name of AERM, we promise that we will change it according to your comments in the revision.

---

> > > > > > ### Comment · Reviewer_YSCG · 2023-08-17
> > > > > >
> > > > > > Thank you. I believe that most of my concerns have been addressed and I am happy to raise my score (4->6).

---

> > > > > > > ### Author Response · Authors · 2023-08-17
> > > > > > >
> > > > > > > Dear Reviewer,
> > > > > > >
> > > > > > > We are glad that we have addressed most of your concerns.
> > > > > > >
> > > > > > > Thank you very much for reviewing our paper.

---

### Author Rebuttal · Authors · 2023-08-02

# The reply to all reviewers

We would like to thank all the reviewers for all your valuable suggestions, which help us improve the quality of our paper.

According to the suggestion of reviewers, we conduct extra experiments on the datasets PACS and OfficeHome. We present the new results in Table 1.

**Table 1**: The OOD adversarial robustness of the models trained by the baselines and our algorithms (%). Results are presented in the form of **a/b**, where **a** is the OOD adversarial accuracy under PGD-20 attack; **b** is the OOD adversarial accuracy under AutoAttack. The other settings are the same as the Tables in our paper.
| Algorithm | RoatatedMNIST | ColoredMNIST | VLCS | PACS | OfficeHome |
| :----: | :----: | :----: | :----: | :----: | :----: |
| ERM | 0.6 / 0.0 | 5.8 / X | 0.0 / 0.0 | 0.3 / 0.6 | 0.4 / 0.0 |
| MLDG | 0.2 / 0.0 | 4.8 / X | 0.0 / 0.0 | 0.1 / 0.3 | 0.6 / 0.1 |
| CDANN | 0.9 / 0.0 | 8.2 / X | 3.0 / 0.0 | 1.5 / 0.3 |0.1 / 0.0 |
| VREx | 0.2 / 0.0 | 6.4 / X | 0.0 / 0.0 | 0.0 / 0.3 | 0.4 / 0.1 |
| RSC | 1.0 / 0.0 | 3.9 / X | 0.0 / 0.0 | 0.1 / 0.4 | 0.7 / 0.0 |
| MAT | X / X | 10.7 / X | 0.0 / 0.0 | 0.7 / 1.4 | 0.8 / 0.1 |
| LDAT | X / X | 7.9 / X | 0.0 / 0.0 | 0.1 / 0.3 | 0.4 / 0.1 |
| **AERM (ours)** | 93.4 / **93.3** | **51.6** / X | 42.6 / 41.8 | **48.1** / 47.6 | 28.6 / 27.4 |
| **RDANN (ours)** | **93.5** / **93.3** | 51.1 / X | **44.9** / **43.9** | **48.1** / **48.0** | **30.2** / **29.8** |


**Table 1** shows that: On the datasets PACS and OfficeHome, the baselines achieve **no more than $2\\%$** average adversarial accuracy under the attack PGD-20 and AutoAttack. Again, our proposed methods can significantly improve the average adversarial accuracy, which is **consistent** with the results of the experiments in our paper.

---

### Decision · Program_Chairs · 2023-09-21

**Decision:**

Accept (poster)

**Comment:**

The paper received mixed reviews. On the positive side, the reviewers found the paper to be well written, strong theoretical motivation, and good empirical results to support them. On the negative side, one reviewer found the paper to be lacking novelty. However, the other reviewers found the paper to be novel and interesting. The AC agrees the paper has interesting results.